**A new version of the CABLE land surface model (Subversion revision r4546), incorporating land use and land cover change, woody vegetation demography and a novel optimisation-based approach to plant coordination of photosynthesis.**

Vanessa Haverd[1], Benjamin Smith[1,2], Lars Nieradzik[3], Peter R. Briggs[1], William Woodgate[3], Cathy M. Trudinger[4], Josep G. Canadell[1], Matthias Cuntz[6]

[1] CSIRO Oceans and Atmosphere, Canberra, 2601, Australia
[2] Dept of Physical Geography and Ecosystem Science, Lund University, Sölvegatan 12, 22362, Lund, Sweden
[3] Centre for Environmental and Climate Research (CEC), Lund University Sölvegatan 37, 22362 Lund, Sweden
[4] CSIRO Land & Water, Canberra, 2601, Australia
[5] CSIRO Oceans and Atmosphere, Melbourne, 3195, Australia

[6] INRA, Université de Lorraine, AgroParisTech, UMR Silva, 54000 Nancy, France

*Correspondence to*:  Vanessa Haverd (Vanessa.haverd@csiro.au)

**Keywords: LULCC, carbon-climate feedback, tree-demography, vegetation structure, phenology, photosynthesis,**

**coordination, CABLE**

**Abstract.**

The Community Atmosphere-Biosphere Land Exchange model (CABLE) is a land surface model (LSM) that can be applied stand-alone, as well as providing the land surface-atmosphere exchange within the Australian Community Climate and Earth System Simulator (ACCESS). We describe new developments that extend the applicability of CABLE for regional and global carbon-climate simulations, accounting for vegetation responses to biophysical and anthropogenic forcings. A land-use and land-cover change module, driven by gross land-use transitions and wood harvest area was implemented, tailored to the needs of the Coupled Model Intercomparison Project-6 (CMIP6). Novel aspects include the treatment of secondary woody vegetation, which benefits from a tight coupling between the land-use module and the Population Orders Physiology (POP) module for woody demography and disturbance-mediated landscape heterogeneity. Land-use transitions and harvest associated with secondary forest tiles modify the annually-resolved patch age distribution within secondary-vegetated tiles, in turn affecting biomass accumulation and turnover rates and hence the magnitude of the secondary forest sink. Additionally, we implemented a novel approach to constrain modelled GPP consistent with the Co-ordination Hypothesis, predicted by evolutionary theory, which suggests that electron transport and Rubisco-limited rates adjust seasonally and across biomes to be co-limiting. We show that the default prior assumption – common to CABLE and other LSMs – of a fixed ratio of electron transport to carboxylation capacity at standard temperature ($J_{max,0}/V_{cmax,0}$) is at odds with this hypothesis; we implement an alternative algorithm for dynamic optimisation of this ratio, such that co-ordination is achieved as an outcome of fitness maximisation. Results have significant implications for the magnitude of the simulated $CO_2$ fertilisation effect on photosynthesis in comparison to alternative estimates and observational proxies.

These new developments enhance CABLE's capability for use within an Earth System Model, and in stand-alone applications to attribute trends and variability in the terrestrial carbon cycle to regions, processes and drivers. Model evaluation shows that the new model version satisfies several key observational constraints, including (i) trend and interannual variations in the global land carbon sink, including sensitivities of interannual variations to global precipitation and temperature anomalies; (ii) centennial trends in global GPP; (iii) co-ordination of Rubisco-limited and electron transport-limited photosynthesis; (iv) spatial distributions of global ET, GPP, biomass and soil carbon; and (v) age-dependent rates of biomass accumulation in boreal, temperate and tropical secondary forests.

CABLE simulations agree with recent independent assessments of the global land-atmosphere flux partition that use a combination of atmospheric inversions and bottom-up constraints. In particular, there is agreement that the strong $CO_2$-driven sink in the tropics is largely cancelled by net deforestation and forest degradation emissions, leaving the Northern Hemisphere (NH) extra-tropics as the dominant contributor to the net land sink.

# 1 Introduction

The Community Atmosphere-Biosphere Land Exchange model (CABLE) is a land surface model (LSM) that can be applied in stand-alone applications and also provides the land surface-atmosphere exchange within the Australian Community Climate and Earth System Simulator (ACCESS) (Kowalczyk et al., 2013; Law et al., 2017; Ziehn et al., 2017). In its stand-alone configuration, CABLE was used in the IPCC 5[th] Assessment report (Ciais et al., 2013), and is one of an ensemble of ecosystem and land-surface models contributing to the Global Carbon Project's annual update of the global carbon budget (Le Quéré et al., 2016; Le Quéré et al., 2018). The current paper describes updates to CABLE targeting two key areas that have been identified as limitations in the applicability and utility of the existing generation of LSMs: (i) land-use and land-cover change (LULCC, hereafter abbreviated to 'LUC') and (ii) adaptation of photosynthesis to changing enviromental conditions.

Additional model updates based on existing parameterisations from the literature include: (i) drought and summer-green phenology (Sitch et al., 2003; Sykes et al., 1996); (ii) low-temperature reductions in photosynthetic rates in boreal forests (Bergh et al., 1998); (iii) photo-inhibition of leaf day-respiration (Clark et al., 2011); and (iv) acclimation of autotrophic respiration (Atkin et al., 2016). These are described in Appendix 2.

## Land-Use and Land-Cover Change

The CABLE version that precedes developments described here (hereafter "Prior CABLE") assumes fixed present-day or pre-industrial vegetation cover in the absence of land management. Capturing the impact of human LUC on the terrestrial carbon and water cycles, and on land-atmosphere coupling, is a key application of LSMs and associated Earth system models (ESMs), and a pre-requisite for evaluation of the models against observation-based datasets.

For the CMIP6 climate model inter-comparison process, the globally gridded Harmonised Land Use Dataset (LUH2) (Hurtt et al., 2016; Hurtt et al., 2011) specifies a matrix of transitions between land use classes (e.g. primary forest, secondary forest, pasture, cropland) through time (Lawrence et al., 2016). In traditional LSMs, these transitions must be translated into annual land-cover maps that specify the fraction of the land surface occupied by each plant functional type (PFT) (Lawrence et al., 2012). This approach reduces the transition matrix to a set of net transitions, thereby discarding information about the gross transitions leading to land-cover change. Simulations driven by gross land use transitions produce emissions that are 15-40% higher than the net transitions alone (Hansis et al., 2015; Stocker et al., 2014; Wilkenskjeld et al., 2014).

Traditional LSMs are also unable to simulate realistic dynamics resulting from the accumulation of carbon in forests following harvest and agricultural abandonment – the so-called secondary forest sink – that is an important contributor to the extant global terrestrial carbon sink (Shevliakova et al., 2009), second only to $CO_2$ fertilisation. This is because traditional LSMs lack representation of woody demography that is required to simulate age-effects on growth and mortality that lead to very high biomass accumulation rates in young forests compared to old-growth stands (e.g. Poorter et al., 2016; Purves and Pacala, 2008; Wolf et al., 2011).

In contrast to traditional LSMs, demography-enabled Dynamic Vegetation Models (DVMs) can implement gross transitions directly and provide realistic representation of the secondary forest sink by explicitly simulating biomass removal and subsequent recovery following a land use event (e.g. Shevliakova et al., 2009). However, keeping track of a representative distribution of landscape elements (patches) of different time since disturbance can be computationally difficult as repeated land use events can lead to a very high number of such elements in a grid-cell.

In this work, we develop a novel LUC scheme for CABLE that is driven by LUH2 gross transitions, and represents age effects on biomass dynamics in all tiles with woody vegetation, including those occupied by secondary forest. This is achieved via coupling with the POP module for woody demography and disturbance-mediated heterogeneity (Haverd et

al., 2013b). The key simplification in the POP approach, compared with other demography-enabled DVMs, is to compute physiological processes such as photosynthesis at the scale of a land-cover tile ("grid-scale"), but to partition the grid-scale biomass increment amongst sub grid-scale patches, each subject to its own dynamics, and distinguished by time since last disturbance. This makes tracking biomass in a large number of patch ages (as arise through both natural disturbance and human land-cover change) easy, and circumvents the computational difficulties of tracking land-cover classes in DVMs.

**Coordination of Photosynthesis**

Almost all global LSMs use the photosynthesis model of Farquhar et al. (1980), or a related scheme derived from this model. Different implementations result in divergent estimates of the response of photosynthesis to environmental drivers in large scale models (e.g. Friend et al., 2014). One reason for this may be that global LSMs have mostly neglected the constraint imposed by the evolutionary-ecological assumption that plants optimise productivity in their environment through relative investment in electron transport and Rubisco-limited steps in the photosynthesis chain, that adjust seasonally and across biomes to be co-limiting. This so-called Co-ordination Hypothesis was originally proposed by Chen et al. (1993) and has been verified experimentally by Maire et al. (2012). Its advantages as an approach to modelling photosynthetic dynamics using limited data constraints was pointed out by Wang et al. (2017), while Ali et al. (2016) have incorporated it into a global mechanistic model of photosynthetic capacity, based on the optimal nitrogen allocation model of Xu et al. (2012). In this work, we will show that the assumption of a temporally invariant ratio of Rubisco and electron-transport capacities (at standard temperature), adopted in Prior CABLE and typically in other LSMs, is not only inconsistent with the Co-ordination Hypothesis, but introduces large uncertainty in simulated sensitivity of GPP to atmospheric $CO_2$ concentration. We solve this problem by developing an algorithm for dynamic optimisation of this ratio, such that co-ordination is achieved as an outcome of fitness maximisation.

**Paper Structure**

The paper is structured as follows. In Section 2 we review the basic structure of CABLE. In Section 3 we describe the model developments that are the focus of this work: firstly, updates to the POP module for woody demography and disturbance; secondly, the new land-use and land-cover change module; thirdly, the dynamic optimisation of plant photosynthesis. In Section 4, we describe the modelling protocol that is used to deliver simulations for evaluating the new model version, and assessing terrestrial carbon-cycle implications of changing climate, $CO_2$, land-use and land-cover over the historical period (1860-2016). In Section 5, we present results of these simulations. Section 5.1 evaluates predictions of present-day spatial distributions of evapotranspiration, gross primary production, biomass and soil carbon. Section 5.2 evaluates predictions of biomass accumulation rates in re-growing forests. Section 5.3 illustrates the capability and behaviour of the land use implementation, showing examples of land-atmosphere carbon exchange at four locations with contrasting LUC histories. Section 5.4 shows the implications of $CO_2$, climate and LUC on historical global and regional land-atmosphere exchange. Sections 5.5 and 5.6 address the implications of simulated photosynthesis co-ordination for the sensitivity of photosynthesis to $CO_2$ and for the $CO_2$ fertilisation of global photosynthesis. Section 5.7 evaluates the new model's prediction of the annual time series of the net land carbon sink by comparison with the equivalent quantity derived from atmospheric mass balance (atmospheric growth rate + ocean sink − fossil fuel emissions). Priorities for future development are summarised in Section 6.

## 2 Model Description

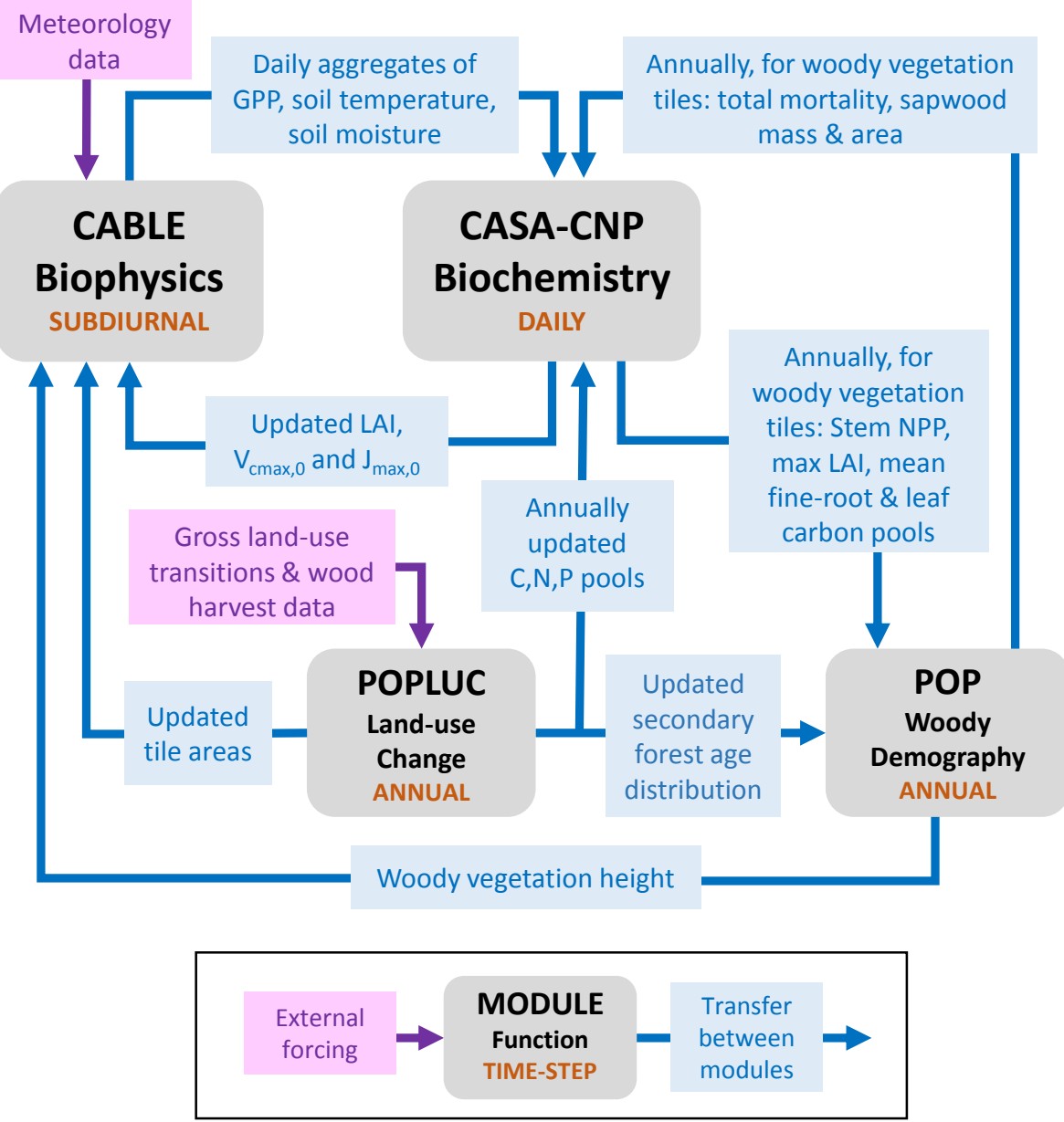

Figure 1: Major sub-models of CABLE (revision 4546), showing forcing data, characteristic time steps, and information flows between modules, which include fluxes, store updates, and changes to vegetation characteristics and their spatial extent (tile areas) within grid cells. Data from faster modules are aggregated before passing to slower modules. Faster modules are updated with data from slower modules at the rate of the slower time step.

Figure 1 summarises the content of CABLE and how the components interact. Further details are presented in Figure A1 (Appendix 1), as pseudo code for each component, and Tables A1-3 (Appendix 3) which document parameter values and temperature response functions of photosynthesis used in this work. CABLE consists of a Biophysics core (Haverd et al., 2016a; Kowalczyk et al., 2013; Wang et al., 2011), the CASA-CNP 'biogeochemistry' module (Wang et al. 2010), the POP module for woody demography and disturbance-mediated landscape heterogeneity (Haverd et al., 2013c; Haverd et al., 2014), and a completely new module for land-use and land management (POPLUC)..

The Biophysics core (sub-diurnal time-step) consists of four components: (1) the radiation module describes radiation transfer and absorption by sunlit and shaded leaves (Goudriaan and van Laar, 1994); (2) the canopy micrometeorology module describes the surface roughness length, zero-plane displacement height, and aerodynamic conductance from the reference height to the air within canopy or to the soil surface (Raupach, 1994); (3) the canopy module includes the coupled energy balance, transpiration, stomatal conductance and photosynthesis and respiration of sunlit and shaded leaves (Wang and Leuning, 1998); (4) the soil module describes heat and water fluxes within soil (6 vertical layers) and snow (up to 3 vertical layers) and at their respective surfaces. The CASA-CNP biogeochemistry module (daily time-step) inherits daily net photosynthesis from the biophysical code, calculates autotrophic respiration, allocates the resulting net primary production (NPP) to leaves, stems and fine roots, and transfers carbon, nitrogen and phosphorous between plant, litter and soil pools, accounting for losses of each to the atmosphere and by leaching. POP (annual time-step) inherits annual stem NPP from CASA-CNP, and simulates woody ecosystem stand dynamics, demography and disturbance-mediated heterogeneity, returning the emergent rate of biomass turnover to CASA-CNP.

The biophysics core of CABLE has been benchmarked using prescribed meteorology (e.g. Best et al., 2015; Zhang et al., 2013; Zhou et al., 2012) and its performance evaluated as part of the Australian Community Climate and Earth System Simulator climate model (Kowalczyk et al., 2013). The CASA-CNP module was developed and tested as a stand-alone module (Wang et al., 2010), and basic performance demonstrated as part of ACCESS (Law et al., 2017; Ziehn et al., 2017). POP (coupled to CABLE) has been evaluated against savanna data (Haverd et al., 2013b; Haverd et al., 2016b), and boreal and temperate forest data (Haverd et al., 2014).

## 3 Model Developments

### 3.1 The Population Orders Physiology (POP) module for woody demography

In previous work, POP has been coupled to both the CABLE and HAVANA land surface schemes and demonstrated to successfully replicate the effects of rainfall and fire disturbance gradients on vegetation structure along a rainfall gradient in Australian savannah – the Northern Australian Tropical Transect (Haverd et al., 2013c; Haverd et al., 2016b), and leaf-stem allometric relationships derived from global forest data. For the latter, it may be argued to reflect the simultaneous development of trees in closed forest stands in terms of structural and functional (productivity) attributes (Haverd et al., 2014). The summary below is reproduced from these papers, which describe POP in detail and with full equations. To enable the extension of CABLE to simulate dynamic land use and implications for forest carbon uptake, we used the most recent version of POP's representation of growth partitioning amongst age/size classes (cohorts) of trees established in the same year; that accounts for both cohort-dependent light interception and sapwood respiration. This contrasts with the original growth partitioning which assumed that individuals capture resources in varying proportion to their size.

POP is designed to be modular, deterministic, computationally efficient, and based on defensible ecological principles. POP simulates allometric growth of cohorts of trees that compete for light and soil resources within a patch. Parameterisations of tree growth and allometry, recruitment and mortality are broadly based on the approach of the LPJ-GUESS Dynamic Vegetation Model (Smith et al., 2001). The time step is one year.

Input variables to POP are annual grid-scale stem biomass increment and mean return times for two classes of disturbance: (i) "catastrophic" disturbance, which kills all individuals (cohorts) and removes all biomass in a given patch; (ii) "partial" disturbances, such as fire, which result in the loss of a size-dependent fraction of individuals and biomass, preferentially affecting smaller (younger) cohorts. For the present study, we adopt a mean catastrophic

disturbance return time of 100 years, and neglect partial disturbance, such as damage caused by wildfires. Stem biomass increment is provided by the host land surface model (LSM), here CABLE.

State variables are the density of tree stems partitioned among cohorts of trees and representative patches of different age-since-last-disturbance across a simulated landscape or grid-cell. Each patch has a number of cohorts. Trees in each cohort are the same age and size because they are established simultaneously and share the same growth rate. Patches are not spatially explicit. Their areal representation in the landscape is given by the patch age distribution.

In the current implementation of POP, the annual stem biomass increment is partitioned among cohorts and patches in proportion to current net primary production of the given cohort (Haverd et al., 2016b). For this purpose, gross primary production and autotrophic respiration for each woody tile are passed from CABLE to POP, and each is partitioned amongst patches and cohorts. Gross resource uptake is partitioned amongst cohorts and patches in proportion to light interception. which is evaluated for each cohort as the difference between downward-looking gap probabilities above and below each cohort. Gap probabilities are calculated using the geometric approach of Haverd et al. (2012). This requires estimates of cohort-specific crown cross-sectional area (related allometrically to DBH) and LAI, computed using the CABLE maximum leaf area, distributed amongst patches and cohorts in proportion to sapwood area. For autotrophic respiration: leaf, fine-root and sapwood respiration components are also partitioned amongst cohorts and patches, according to the size of each biomass component. Cohort-specific sapwood is prognosed by assuming sapwood conversion to heartwood at a rate $0.05 \text{ y}^{-1}$. Cohort-specific leaf and root carbon pools are estimated by partitioning the aggregate values for each woody tile in proportion to leaf area index (LAI). Net resource uptake for each patch and cohort is evaluated as its gross primary production minus autotrophic respiration.

Cohort stem density is initialised as recruitment density, and is episodically reset when the patch experiences disturbance. Mortality, parameterized as the sum of cohort-specific resource-limitation and crowding components, reduces the stem density in the intervening period. Resource-limitation mortality, a function of growth efficiency (GE i.e. growth rate relative to biomass), is described by a logistic curve with an inflection point representing a critical GE level at which plants experience a steep increase in mortality risk due to a shortage of resources to deploy in response to stress or biotic damage (Haverd et al., 2013c). The crowding mortality component (Haverd et al., 2014) allows for self-thinning in forest canopies.

Additional mortality occurs as a result of disturbances. Patches representing stands of differing age since-last-disturbance are simulated for each grid-cell. It is assumed that each grid-cell is large enough to accommodate a landscape in which the frequency of patches of different ages follows a negative exponential distribution with an expectation related to the current disturbance interval. This assumption is valid if grid-cells are large relative to the average area affected by a single disturbance event and disturbances are a Poisson process, occurring randomly with the same expectation at any point across the landscape, independent of previous disturbance events. To account for disturbances and the resulting landscape structure, state variables of patches of different ages are linearly interpolated between ages, and weighted by probability intervals from the negative exponential distribution. The resultant weighted average of, for example, total stem biomass or annual stem biomass turnover, is taken to be representative for the grid-cell as a whole.

In earlier applications, CABLE-POP coupling consisted of just two exchanges: (i) stem NPP passed from the host LSM to POP; (ii) woody biomass turnover returned from POP to the host LSM. To convert between stem biomass (POP) and tree biomass (CABLE), we assume a ratio of 0.7, a representative average for forest and woodland ecosystems globally (Poorter et al., 2012). The POP biomass lost by mortality is applied as an annual decrease in the CASA-CNP tree biomass pool, and replaces the default fixed biomass turnover rate. In the current work, the coupling also includes the return of sapwood area and sapwood biomass to the CASA-CNP biogeochemical module of CABLE, where these

variables respectively influence C-allocation to leaves and autotrophic respiration. Combined allocation to leaves and wood is partitioned following the Pipe Model (Shinozaki et al., 1964), such that a target ratio of leaf area to sapwood area (a global value of 5000 is assumed) is maintained. Sapwood replaces stem-wood biomass in the CASA-CNP calculation of stem respiration. These feedbacks of POP structural variables on leaf area and autotrophic respiration

result in net primary production (NPP) that reflect the area-average sapwood area and mass of each woody tile.

**Advantages and Limitations of the POP approach to simulating large-scale biomass dynamics**

POP is not a replacement for a full-featured Dynamic Vegetaion Model (DVM), but does overcome key limitations of prior CABLE and many DVMs adopted by most Earth System Models (Arora et al., 2013) for which biomass turnover is often represented as a first-order decay process, expressed as the product of grid-cell biomass and a bulk rate parameter.

This "big wood" approximation does not resolve underlying population and community processes such as recruitment, mortality and competition between individuals for limiting resources (e.g. Sitch et al., 2003), and has been demonstrated to lead to an inaccurate trajectory of biomass accumulation with stand age (Wolf et al. 2011; Haverd et al. 2014) . Big wood models are additionally unable to directly exploit the wealth of information on forest stand structure and dynamics available from forest inventories. By discriminating individual and population growth and explicitly representing

asymmetric competition among age/size classes of trees co-occurring within forest stands, POP overcomes the limitation of the big wood approach and has proved able to reproduce allometric relationships reflecting linkages between productivity, biomass and density in widely distributed forests (Haverd et al., 2014). This is achieved without a marked increase in model complexity or computational demand, thanks to a modular design that separates the role of the parent land surface model (prognosing whole-ecosystem production) and the population dynamics model (partitioning the

production among cohorts, computing mortality for each and returning the stand-level integral as whole-ecosystem biomass turnover to the parent model) (Fig. 1).

A draw-back of this modular approach is that age effects on leaf area and NPP are not accounted for explicitly at the scale of the individual, because these variables are computed for each woody-tile and in-turn distributed amongst POP patches and cohorts. Feedbacks of stand-structure on leaf area and NPP thus reflect the area-averaged structural properties

(sapwood area and sapwood mass) of each woody tile.

POP does not represent competitive interactions among PFTs, that provide an important explanation for global biome distributions and may modulate the responses of vegetation to future climate and $CO_2$ forcing (Smith et al., 2014). We plan to introduce PFTs and to distinguish canopy and understorey strata in a later development of the approach.

**3.2 POPLUC Land-use and land-cover change module**

This development enables the simulation of the effect of LUC on land-cover fractions and associated carbon flows into and out of soil, litter, vegetation and product pools.
Three land-use tile types are considered: primary woody vegetation (p); secondary woody vegetation (s) and open grassy vegetation (g), the latter encompassing natural grassland, rangeland, pasture and cropland. Forcing data comprising four possible annual gross transition rates are used to drive the annual LUC-induced changes to land-use area fractions. These

transition rates are: (i) primary clearing (p→g), (ii) secondary clearing (s→g), (iii) primary harvest (p→s), (iv) abandonment (g→s). In addition, secondary forest harvest area is used to drive changes in the secondary forest age distribution. Further, cropland and pasture area fractions are diagnosed from transitions to and from pasture and cropland, and used to estimate carbon cycle consequences of crop harvest, tillage and grazing.

## 5 Mapping land-use tile types to CABLE plant functional types

Potential vegetation cover is prescribed using BIOME1 (Prentice et al., 1992), a semi-mechanistic climate-envelope approach, to construct global spatial distribution of biomes according to CABLE's own climate drivers, which are accumulated from 30 years (1901-1930) of meteorological inputs (Figure 2).

Biomes (combinations of dominant plant types (Prentice et al., 1992)) are mapped to a single CABLE plant functional
10  type (PFT), or in some cases to two CABLE PFTs (one woody and one herbaceous) with fixed relative areal proportions (Table 1). We make use of five woody vegetation types (Evergreen Needleleaf , Evergreen Broadleaf, Deciduous Needleleaf, Deciduous Broadleaf, Shrub), and six non-woody types ($C_3$ grass, $C_4$ grass, Tundra, Wetland, Barren, Ice). All woody vegetation tiles are represented by POP, and secondary woody vegetation tiles are assumed to be occupied by the woody PFT of the primary woody vegetation tile in the same grid-cell.

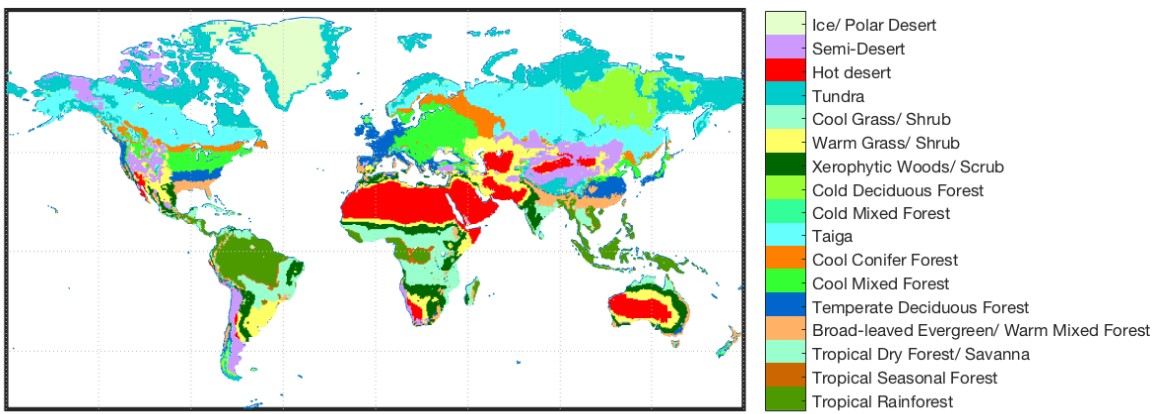

Figure 2: Spatial distribution of BIOME1 biomes (Table 1) that determines the type of primary vegetation cover

**Table 1: CABLE primary vegetation: mapping of BIOME1 biomes to CABLE Plant Functional Types**

| BIOME1 biome | CABLE PFT | Fraction grass* |
|---|---|---|
| Tropical Rainforest | Evergreen Broadleaf | 0 |
| Tropical Seasonal forest | Evergreen Broadleaf | 0 |
| Tropical Dry Forest/Savanna | Evergreen Broadleaf | 0.6 |
| Broad-leaved Evergreen/Warm Mixed-Forest | Evergreen Broadleaf | 0 |
| Temperate Deciduous Forest | Deciduous Broadleaf | 0.3 |
| Cool Mixed Forest | Deciduous Broadleaf | 0.3 |
| Cool Conifer Forest | Deciduous Needleleaf | 0.2 |
| Taiga | Evergreen Needleaf | 0.2 |
| Cold Mixed Forest | Evergreen Needleaf | 0.2 |

| Cold Deciduous Forest | Deciduous Needleleaf | 0.2 |
|---|---|---|
| Xerophytic woods/scrub | Shrub | 0.6 |
| Warm grass/shrub | Shrub | 0.8 |
| Cool grass/shrub | Shrub | 0.8 |
| Tundra | Tundra | 0 |
| Hot Desert | Barren | 0 |
| Semi-Desert | Shrub | 0.8 |
| Ice/ polar Desert | Ice | 0 |

* Grass is specified as $C_3$ where monthly minimum temperature is less than 15.5°C, and $C_4$ elsewhere.

**Tracking land-use area fractions and secondary forest age-distribution**

Each land-use tile has an associated areal fraction, representing its fractional area cover of the grid-cell. Land transition area rates augment and deplete land-use area fractions, subject to land availability. In secondary forest tiles, the areal
fraction of each integral age class (0-400 y) is also tracked: a transition to secondary forest (p→s or g→s) augments the 0 age-class by the same amount. A transition from secondary forest to open land (s→g) depletes the areas of youngest age classes first, starting from 10-y. If the clearing area exceeds the area covered by age classes older than 10-y, clearing is applied uniformly across all age classes. A secondary harvest event sequentially depletes the areas of each age class, starting from the oldest, until all harvest area is satisfied, subject to land availability. Secondary forest tiles are also
subject to natural disturbance, which further modifies the patch age distribution.
The POPLUC code provides the secondary forest patch age distribution to POP. POP tracks biomass in each of a set of patches with different ages,, based on patch-dependent growth and turnover. It then computes biomass for each integral age class represented by the secondary forest tile patch age distribution by interpolating biomass in the simulated patches.
POPLUC represents integral secondary forest ages classes from 0 to 1000 y old inclusive, although many ages may have a weight of zero. The frequency distribution is fully dynamic. In contrast POP represents 60 patches in each woody tile, spanning a distribution of ages from 0 to 1000.

**Re-distribution of carbon stocks following land-use-change**

Changes in pool sizes of biomass, soil and litter carbon in the biogeochemical module are updated to reflect the areal changes from gross land-use transitions. Analogous updates occur for nitrogen pools. The mass balance equation for the carbon density $c_j$ [g m$^{-2}$] in each land-use tile $L$, with area $A_L$ [m$^2$] that accounts for the possibility of more than one gross receiver ($r$) or donor ($d$) transition to or from the tile, is:

$$c_{j,L,0}A_{L,0} - c_{j,L,0}\Delta A_{L,d} + F_{j,L,r}^{transfer} = c_{j,L}\left(A_{L,0} + \Delta A_L\right)$$
(0)

Here $j$=1-9 (referring to carbon in leaf, wood, fine roots, 3 litter pools and 3 soil pools) and $L$ = 1-3 (referring to primary woody, secondary woody, open land-use tiles); subscript $0$ refers to the value of the tile area or carbon density prior to the transitions; $\Delta A_L$ refers to the total (net) change in land-area of the $L_{th}$ tile; $\Delta A_{L,d}$ refers to the absolute change in land area due to donor transitions. In Eq (0), the first term on the LHS is the carbon stock prior to land-use perturbations; the second term is the carbon lost from the tile due to donor transitions (transitions from the $L_{th}$ tile) and the third term is the
carbon gained by receiver transitions (transitions to the $L_{th}$ tile). The term on the RHS is the carbon stock following the perturbations (i.e. the product of the new carbon density and the new tile area).
The carbon gained by receiver transitions is generally:

$$F_{j,L,r}^{transfer} = \sum_{k=1}^{n_{trans}} \Delta A_k c_{j,k} \qquad (0)$$

where the total transfer of carbon is summed over all possible gross transitions ($n_{trans}$ = 4), and each transition contributes carbon to the receiver pool that is equal to the product of the transition area $\Delta A_k$ multiplied by the carbon density of the donor pool $c_{j,k}$. An exception to Equation (0) is the transfer of carbon the coarse woody debris pool and fine structural litter as the result of clearing or wood harvest: woody biomass residue from harvest and clearing augments the coarse woody debris pool, whereas leaf and fine-root residue augment the fine structural litter pool. In the case of secondary forest, harvest and clearing are age-selective, which means that biomass loss and litter increment are affected not only by cleared/harvested secondary forest area, but also by the age distribution of the stems that are removed. Harvested and cleared biomass that is not left as residue is extracted into three product pools with turnover rates of 1 y, 10 y and 100 y. Coefficients for allocation to these product pools, as well as the fractions of harvested and cleared biomass that remain in the landscape as litter are prescribed following Hansis et al. (2015).

Carbon losses by secondary forest harvest and clearing need to be resolved from net biomass loss in secondary forest tiles, which also includes components from natural disturbance and areal expansion. ,POP diagnoses a change in biomass resulting from the aggregate shift in age distribution contributed by natural disturbance, forest expansion, harvest and clearing. The proportional contributions of each of these processes to total biomass change is recorded. The carbon flux implied by this total biomass change is subsequently disaggregated according to the previously recorded proportional contributions of each process.

Carbon removal from the landscape by crop harvest and grazing are treated simply. Crops and pasture are not treated in separate land-use tiles, but are simulated as grass in the open "grassy" tile of each grid-cell. The areal fractions of cropland and pasture in each open tile are tracked via the gross transitions to and from these land-use types. These fractions, combined with assumed respective removals of 90% and 50% of above-ground NPP by crop-harvest and grazing (Lindeskog et al., 2013), are used to prescribe leaf-litter transfer to an agricultural product pool with a turnover time of 1 y. Following Lindeskog et al. (2013), soil carbon loss by tillage is simulated by increasing turnover of soil carbon by 50% in croplands. Where crops and pasture occupy more than 10% of a grass tile, it is assumed that there is no nutrient limitation to growth.

### 3.4 Optimisation-based approach to plant coordination of electron transport and carboxylation capacity-limited photosynthesis in C$_3$ plants

Photosynthesis, as represented by the Farquhar et al. (1980) model, may be limited by the Rubisco-catalysed maximum rate of carboxylation ($V_{cmax}$), or the maximum rate of electron transport ($J_{max}$). Estimates of these parameters based on leaf gas exchange measurements suggest their ratio at standard temperature (25$^o$C) to be conservative around a global mean of $b_{JV} = J_{max,0}/V_{cmax,0} = 1.7\pm0.1(1\sigma)$ (e.g. Walker et al., 2014) which has led to it being widely adopted as a fixed parameter in global terrestrial biosphere models. However, as we will show in Sections 5.5 and 5.6, the assumption of a fixed value of $b_{JV}$ leads to large deviations from the Co-ordination Hypothesis (Chen et al., 1993; Maire et al., 2012) that Rubisco and electron-transport capacity adjust seasonally and across biomes to be co-limiting. An alternative but closely-related assumption is that plants optimise $b_{JV}$ to minimise the nitrogen cost per unit photosynthesis. Here we describe a generic approach to dynamically optimizing $b_{JV}$ based on this assumption.

### Review of model for net photosynthesis

Here we review the equations of the C$_3$ photosynthesis model (Farquhar et al., 1980) as embedded in CABLE. We note here that in CABLE, these equations are coupled to the canopy environment via leaf surface energy balance, and to the

air above the canopy via turbulent transfer processes, which we will not review here (see Kowalczyk et al. (2006) for full description).

Net photosynthesis ($A_n$) is equated with supply of $CO_2$ to the inter-cellular air-spaces:

$$A_n = g_{sc}\left(c_s - c_i\right) \tag{1}$$

where $g_{sc}$ is the stomatal conductance to $CO_2$, $c_s$ is the concentration of $CO_2$ at the leaf surface and $c_i$ is the intercellular $CO_2$ concentration.

Net photosynthesis is also equated with biochemical demand for $CO_2$, i.e. the lesser of Rubisco- and electron transport-limited rates of carboxylation, minus day respiration:

$$A_n = \min\left[A_c, A_e\right] - R_d \tag{1}$$

The two potentially-limiting rates are given by

$$A_c = V_{cmax} \frac{c_i - \Gamma_*}{c_i + K_c\left(1 + c_o / K_o\right)} \tag{1}$$

and

$$A_e = \frac{J}{4} \frac{c_i - \Gamma_*}{c_i + 2\Gamma_*} \tag{1}$$

where $V_{cmax}$ is the maximum catalytic activity of Rubisco in the presence of saturating levels of $RuP_2$ and $CO_2$; $\Gamma_*$ is the $CO_2$ compensation point in the absence of day respiration; $K_c$ and $K_o$ are Michaelis-Menten constants for $CO_2$ and $O_2$ respectively; $c_o$ is concentration of $O_2$; J is the electron transport rate, and is related to absorbed photon irradiance Q by (Farquhar and Wong, 1984):

$$\theta J^2 - \left(\alpha Q + J_{max}\right)J + \alpha Q J_{max} = 0 \tag{1}$$

where $\alpha$ is the quantum yield of electron transport and $\theta$ a curvature parameter. Temperature response functions of $V_{cmax}$, $J_{max}$, $\Gamma_*$, $K_c$ and $K_o$ are given in Table A3 (Appendix 3). The parameterisation of Rd is given by Equation (21) in Appendix 2.

Stomatal conductance is expressed as a linear function of $A_n$:

$$g_{sc} = g_{min} + X A_n \tag{1}$$

Following, (Lin et al., 2015) we set $g_{min}$ to zero, and adopt the following dependence of $X$ on leaf-air vapour pressure deficit ($D_{leaf}$)

$$X = \frac{1}{c_s}\left(1 + \frac{f_{w,soil} g_1}{\sqrt{D_{leaf}}}\right) \tag{1}$$

where $f_{w,soil}$ is related to soil moisture deficit and is parameterised according to Haverd et al. (2016a) and the PFT-dependent $g_1$ parameter is sourced from Lin et al. (2015).

Equations (1), (1) and (1) are solved simultaneously for $A_n$, $c_i$ and $g_{sc}$.

**Dynamic optimisation of $b_{JV}$: assumptions**

The approach to optimisation of $b_{JV}$ is based on four assumptions:

(i) Leaf nitrogen resources may be dynamically re-distributed at a 5-day timescale at no cost, i.e. $b_{JV}$ is optimised, such that net photosynthesis (given total available leaf nitrogen) accumulated over the last 5 days (approximately the time-scale for turnover of Rubisco) would have been maximised.

(ii) Leaf nitrogen resources available for partitioning between Rubisco- and electron-transport capacity are proportional to effective nitrogen content ($N_{eff}$), defined as the sum of prior estimates of $V_{cmax,0}$ and $J_{cmax,0}$, weighted by relative cost $c_{cost,JV}$:

$$N_{eff} = V_{c,max,0}^0 + c_{cost,JV}\frac{J_{max,0}^0}{4} \qquad (1)$$

where superscript $0$ denotes prior estimate; subscript $0$ denotes standard temperature, and

$$J_{max,0}^0 = b_{JV}^0 V_{c,max,0}^0 \qquad (1)$$

$N_{eff}$ is preserved as $b_{JV}$ is adjusted, such that the adjusted (actual) values of $V_{cmax,0}$ and $J_{max,0}$ are :

$$V_{c,max,0} = \frac{N_{eff}}{1+\dfrac{c_{cost,JV}b_{JV}}{4}} \qquad (1)$$

and

$$J_{max,0} = b_{JV}V_{cmax,0} \qquad (1)$$

(iii) The prior values of $V_{cmax,0}$ (related to leaf nitrogen and phosphorous content) and $b_{JV}$ are prescribed according to the synthesis of globally distributed leaf gas exchange measurements by Walker et al. (2014).

(iv) The emerging contributions of electron transport and Rubisco-limited rates contribute approximately equally to total net photosynthesis.(Chen et al., 1993) In practice, this requires a relative cost factor $c_{cost,JV}$ of 2.0 (slightly higher than a prior estimate of 1.6 which is the ratio of the linear-regression slopes relating $J_{max,0}$ and $V_{c,max,0}$ to leaf N (Chen et al., 1993)).

**Dynamic optimization of $b_{JV}$: method**

The method for implementing these assumptions in CABLE is:

(i) Maintain a 5-day history of subdiurnal leaf-level meteorology (absorbed PAR; leaf-air VPD difference; leaf temperature, $c_s$) for sun-lit and shaded leaves, such that $A_{n,5d}$ can be reconstructed for sunlit and shaded leaves. Other subdiurnal variables that are required are $R_d$ (Eq (1)) , $f_{wsoil}$ (Eq (1)) and a scaling parameter that relates leaf-level $J_{max}$, $V_{cmax}$ and $R_d$ to their effective "big-leaf" sunlit and shaded values via integration of these parameters over canopy depth under the assumption that the leaf-level values are proportional to leaf nitrogen which decreases exponentially from canopy top (Wang and Leuning, 1998 (Eqs C6 and C7)).

(ii) Construct a function that calculates leaf nitrogen cost per unit net photosynthesis ($N_{eff}/A_{n,5d}$). Inputs to this function are: (1) current estimate of $b_{JV}$; (2) $N_{eff}$ (Eq (1)) ; (3) 5-day history of subdiurnal leaf-level meteorology.

(iii)     Implement a search algorithm to find $b_{JV}$ that minimises the function above for $N_{eff}/A_{n,5d}$. Here we use the Golden Section Search Algorithm (Press et al., 1993).

(iv)     Insert a call to the optimisation algorithm at the end of each day, at the point in the code where $V_{c,max,0}$ and $J_{max,0}$ are being returned from the CASA-CNP biogeochemistry module to the CABLE biophysics module (Figure A1) In this way, $b_{JV}$, and hence $V_{c,max,0}$ and $J_{max,0}$ for sun-lit and shaded leaves are updated daily, based on the leaf environment of the last five days.

## 4. Modelling Protocol

Global simulations were performed at 0.5° × 0.5° spatial resolution, with time steps of 3h (biophysics); 1d (biogeochemistry) and 1y (woody demography, disturbance, LUC). The nitrogen cycle was enabled, but not the phosphorous cycle. Recently developed parameterisations for drought-response of stomatal conductance and effects of leaf litter on soil evaporation were enabled (Haverd et al., 2016a), but not representations of effects of ground water and sub-grid scale heterogeneity on the water cycle (Decker, 2015). The soil-moisture response of heterotrophic respiration developed by Trudinger et al. (2016) was enabled, and the default Q10 formulation for the temperature response was replaced by that of Lloyd and Taylor (1994). For $C_3$ PFTs, The relationship between $V_{c,max,0}$ and leaf nutrient status was prescribed using the meta-analysis of leaf gas-exchange data by Walker et al. (2014), and $\alpha$ and $\theta$ (Eq (1)) were prescribed to be consistent with this analysis.

### Forcing Data

Simulations were driven by (i) daily CRU-NCEP V7 (1901-2016) (Viovy, 2009), down-scaled to 3-hourly resolution using a weather generator (Haverd et al., 2013a); (ii) $CO_2$ (1-y) resolution (Dlugokencky and Tans, 2017); (iii) gridded nitrogen deposition (10-y resolution) (Lamarque et al., 2011); (iv) gridded gross land-use transitions and harvest (1500-2015) and initial land-use states (1500) from the LUH2 harmonised land-use data set (Hurtt et al., 2016; Hurtt et al., 2011), re-gridded to 0.5° × 0.5° spatial resolution, and aggregated to four transitions associated with the three land-use classes resolved in this study (Section 3.1). In this aggregation, we include all transitions to and from both 'forest' and 'non-forest' components of LUH2 primary and secondary vegetation. Land-use transitions and harvest are only applied in grid-cells where CABLE's primary vegetation includes a woody PFT. For simplicity, we neglect transitions from natural grass land to forest.

### Simulation Scenarios

Simulations were performed to quantify the net land-atmosphere carbon flux, and attribute it to three components: (i) the land-atmosphere exchange that would occur in response to changing climate, $CO_2$ and nitrogen deposition under a scenario of 1860 land-cover ($F_{cc}$); (ii) the land-atmosphere exchange that would occur in response to land-use-change and management under a scenario of 1860 $CO_2$ and Nitrogen deposition and baseline (recycled 1901-1920) climate ($F_{LUC,0}$); (iii) the additional LUC and management emissions arising from the effects of changing climate and $CO_2$, combined with the reduction in sink capacity arising from land-use conversion ($F_{CC \times L}$).

**Table 2: Simulation Scenarios**

| Scenario | climate | $CO_2$ | Nitrogen Deposition | Land-use and land-cover change | Net C flux to atmosphere, including decay of products |
|---|---|---|---|---|---|
| | | | | | |

| (i) | Recycled (1901-1920) | 1860 | 1860 | 1860 | $F_{0,0}$ |
|---|---|---|---|---|---|
| (ii) | 1901-2016 | 1860-2016 | 1860-2016 | 1860 | $F_{CC,0}$ |
| (iii) | Recycled (1901-1920) | 1860 | 1860 | 1500-2016 | $F_{0,L}$ |
| (iv) | Recycled (1901-1920) | 1860 | 1860 | 1500-2016, no wood harvest residue | $F_{0,L,no\_residue}$ |
| (v) | Recycled (1901-1920) | 1860 | 1860 | 1500-2016, no grazing and crop harvest | $F_{0,L,no\_Ag}$ |
| (vi) | 1901-2016 | 1860-2016 | 1860-2016 | 1500-2016 | $F_{CC,L}$ |

This allows the net flux $F_{CC,L}$ (combined response to $CO_2$, climate and LUC) to be partitioned as:

$$F_{CC,L} = F_{LUC,0} + F_{CC} + F_{LUC \times CC} \qquad (2)$$

where

$$F_{LUC,0} = F_{0,L} - F_{0,0}$$

$$F_{cc \times L} = F_{CC,L} - F_{CC,0} - F_{LUC,0} \qquad (3)$$

$$F_{CC} = F_{CC,0}$$

Scenario (iv) is included so that the net ecosystem production (NPP minus heterotrophic respiration) on secondary forest tiles can be partitioned between secondary forest regrowth, and legacy emissions from post-harvest and post-clearing residues, which are zero in Scenario (iv). Note here that $F_{0,L,no\_residue}$ and $F_{0,L}$ slightly different (~0.05 PgCy$^{-1}$ globally, because of soil nitrogen feedbacks on growth and different carbon residence times in product pools vs soil and litter).

However this difference doesn't affect the accuracy of reported net fluxes, since Scenario (iv) is only used for flux partitioning.

Scenario (v) is included to resolve the net LUC emissions associated with grazing and cropland management as the difference $F_{0,L}$ - $F_{0,L,no\_Ag}$.

The loss of additional sink capacity (1860 reference year) $F_{LASC}$ can be resolved as one component of $F_{LLxC}$, using tile-

based fluxes computed in Scenario (ii), and tile area weights computed in Scenaro (vi) as

$$F_{LASC} = \sum_{i=1}^{n} w_{1860}^i F_{CC,0}^i - \sum_{i=1}^{n} w_{actual}^i F_{CC,0}^i \qquad (4)$$

where $w_{1860}$ and $w_{actual}$ are the 1860 and actual grid-cell tile weights respectively, and the sums are over all the tiles in each grid-cell.

The initialization phase of each scenario was designed to establish the dynamic equilibrium between model state

(biomass and soil carbon pools) and the forcing data. All scenarios were initialized from zero biomass (to ensure biomass variables in POP and CASA-CNP start from the same value) and arbitrary soil carbon and nutrient stocks, and brought to equilibrium with 1901-1920 climate by five repetitions of a pair of model runs. This pair comprised a full model run (1901-1920 climate, 1860 land-cover, $CO_2$, Nitrogen deposition), followed by a semi-analytic spin-cycle (Xia et al., 2012), adapted to include calls to the POP demography module, and driven by GPP, soil moisture and temperature fields

from the full model run. Due to the need to account for the legacy effects of past land-use on soil carbon and secondary forest state, an additional initialization of the vegetation and soil carbon pools as influenced by land-use change and land management was performed for 1500-1710, for the scenarios with dynamic land-use. To circumvent high computational

costs of the sub-diurnal solution of carbon and water fluxes, we used the same pre-computed GPP, soil moisture and temperature fields generated for the semi-analytic spin cycle. A final initialization phase consisted of running the full model from 1711 to 1859 with dynamic land-use forcing. The full model was then run for the 1860-2016 analysis period for all scenarios, with 1901-1920 meteorology recycled prior to 1901.

In addition to the above scenarios, we also explored the impact on global GPP of dynamically optimizing $b_{JV}=J_{max,0}/V_{cmax,0}$. Simulations were performed under assumptions of dynamically optimized and fixed $b_{JV}$ (values of 1.6, 1.7, 1.8). For these simulations, static 1860 land-cover was assumed and for computational efficiency, simulations were based on a sample of 1000 randomly distributed grid-cells across the global ice-free land-surface.

## 5 Results

### 5.1 Model evaluation: evapotranspiration, GPP, biomass and soil carbon

Model-data comparisons of spatial distributions of key fluxes and stocks are presented in Figure 3. We choose to evaluate the model against GPP, biomass and soil carbon because these are key quantities that are critical constraints on the global terrestrial carbon cycle and for which global distributions are available. We include evapotranspiration (ET) here as it is a key constraint on GPP, because both ET and GPP are regulated by stomatal conductance.

The mean of evapotranspiration (ET) was obtained from the LandFlux $0.5° \times 0.5°$ data product (Mueller et al., 2013), that merges multiple remote sensing and flux station-based ET products into a single data set. CABLE and the LandFlux latitudinal profile of ET differ by a mean absolute error of 0.12 mm d$^{-1}$. There is an underestimate in the tropics of up to 0.4 mm d$^{-1}$ (although note LandFlux $1\sigma$ uncertainty of ~1 mm d$^{-1}$ in this region), an underestimate that has been noted in previous evaluations of CABLE global ET (De Kauwe et al., 2015; Decker, 2015) and is particularly noticeable in the 20  Amazon.

Observation-based global gross primary production (GPP) was obtained from upscaled FLUXNET eddy-covariance tower measurements (1982-2011) (Jung et al., 2010). CABLE and FLUXNET estimates of the latitudinal distribution of GPP differ by mean absolute error of 147 gCm$^{-2}$y$^{-1}$.. CABLE global GPP sums to 134 PgCy$^{-1}$ for the year 2000, 9% higher than the FLUXNET estimate (123 PgCy$^{-1}$). An over-prediction by CABLE is noted for southern hemisphere (SH) 25  regions south of -30°, a bias that is possibly related to SH temperate Evergreen Broadleaf forests being represented by the same CABLE PFT as tropical Evergreen Broadleaf forests (Table 1), and a fixed global value of the leaf area to sapwood area ratio.

Observation-based above-ground forest biomass at $0.01°\times0.01°$ resolution for the first decade of the 2000s was obtained from the GEOCARBON product (Figure 3(vii)), which is an integration of northern-hemisphere forest biomass (Santoro 30  et al., 2015) with a pan-tropical biomass map (Avitabile et al., 2016), itself a fusion of two existing large-scale biomass maps (Baccini et al., 2012; Saatchi et al., 2011) with local biomass data. The map covers only forest areas, where forests are defined as areas with dominance of tree cover in the GLC2000 map (Bartholomé and Belward, 2005). We also compare CABLE above-ground biomass with the product of Saatchi et al. (2011) (Figure 3(ix)), that is a combination of data from in situ inventory plot data, satellite Lidar samples of forest structure, and optical and microwave imagery to 35  extrapolate over the landscape, also at $0.01°\times0.01°$ resolution. The CABLE and GEOCARBON latitudinal biomass estimates differ by mean absolute error of 0.47 PgCdeg$^{-1}$.Globally, CABLE's estimate for the year 2000 sums to 246 PgC above ground biomass (assumes above ground fraction of 0.7), 15 % higher than the GEOCARBON estimate of 209

PgC. Most of the discrepancy is in China (observational uncertainties of 25-50%), where CABLE over-predicts biomass carbon compared to GEOCARBON, but under-predicts compared to Saatchi et al. (2011).

Soil carbon density in the top 1 m of soil for the year 2000 was obtained from the Harmonized World Soil Database (HWSDA) (version 1.2). (FAO/IIASA/ISRIC/ISSCAS/JRC, 2009). Latitudinal profiles of soil carbon from CABLE
(total soil carbon and litter) differs from the HWSDA product by a mean absolute error of 1.8 PgCdeg$^{-1}$ (Figure 3(xii)), and the CABLE global total of 1426 PgC is 7% higher than the HWSDA estimate of 1329 PgC. However, spatial distributions show large differences, most notably over-prediction by CABLE across much of the taiga and cold deciduous forest biomes. Another region of discrepancy is temperate south-eastern Australia, where CABLE predicts higher soil carbon (35-40 kg C m$^{-2}$) than HWSDA; however CABLE estimates are consistent with regional observation-
based estimates (Viscarra Rossel et al., 2014).

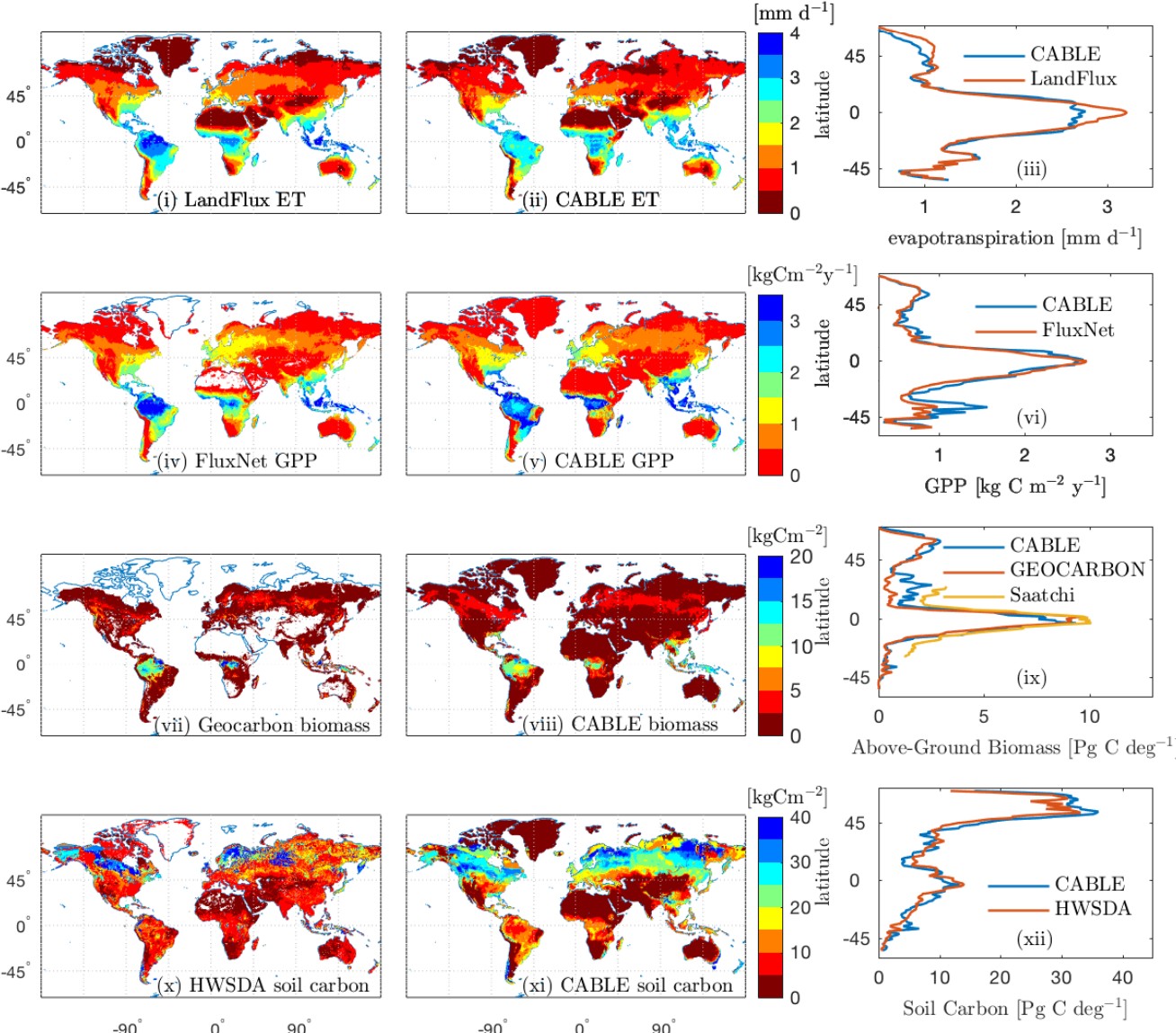

Figure 3: Observation-based (left), CABLE (middle) spatial distributions, and corresponding latitudinal distributions (right) of: (i)-(iii) evapotranspiration; (iv)-(vi) GPP; (vii)-(ix) above-ground biomass (x)-(xi) soil carbon (FAO/IIASA/ISRIC/ISSCAS/JRC, 2009). In the observation-based estimates, white indicates missing values. All
quantities are annual means for the year 2000, except for GEOCARBON biomass (first decade of the 2000s).

## 5.2 Model evaluation: age-dependence of biomass accumulation

### Temperate and Boreal Forests

Forest inventory data for above-ground biomass and age were sourced from the Biomass Compartments Database (Teobaldelli, 2008). This database contains data from around 5790 plots and represents a harmonized collection of Cannell (1982) and Usoltsev (2001) datasets, covering the temperate and boreal forest region globally. In earlier work we used the database to construct biomass-density plots for the purpose of calibrating the crowding mortality component of POP and to evaluate CABLE leaf-stem allometry plots relating foliage and stem biomass per tree (Haverd et al., 2014). Here we directly evaluate CABLE predictions of above-ground stem biomass for 1990 (approximate median year for the observational data) (Figure 4) for a wide range of stand ages (2-200 y). Despite significant scatter, predictions show low bias (Figures 4(i) and (ii)) and biomass-age relationships that accord with the data (Figures 4(iii) and (iv)): [DBL, n= 1476; $r^2 = 0.35$; bias error = 0.4 kgCm$^{-2}$; root mean squared error = 2.6 kgCm$^{-2}$], [ENL, n=931; $r^2 = 0.46$; bias error = -0.9 kgCm$^{-2}$; root mean squared error = 3.7 kgCm$^{-2}$].

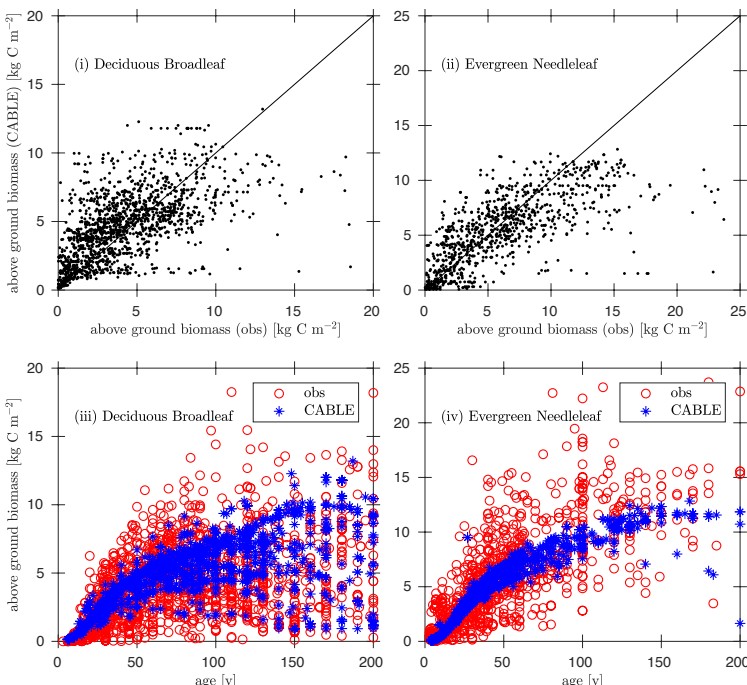

Figure 4: Evaluation of CABLE (1990) above-ground biomass predictions against Biomass Compartments Data (Teobaldelli, 2008), separated into Deciduous Broadleaf and Evergreen Needleleaf classes: (i),(ii) above-ground biomass predictions versus observations (solid line represents 1:1 line); (iii)-(iv) predictions and observations of above-ground biomass versus age.

### Tropical Forests

CABLE regrowth rates of secondary forests in the Tropical Rainforest, Tropical Seasonal Forest and Tropical Dry Forest/Savanna biomes (Figure 2) in South America compare well with observation-based estimates by Poorter et al. (2016). This database has 1500 forest plots at 45 sites spanning the major environmental gradients across the Neotropics (Figure 5), where mean annual rainfall is the strongest environmental predictor of biomass accumulation after 20 y (Poorter et al., 2016).

In this region, CABLE predicts that secondary forest biomass recovers to 41±6 (1σ) % of its undisturbed value after 20 yeras of recovery, in good agreement with observations 54±16 (1σ) % (Poorter et al., 2016). Poorter et al. (2016) emphasise high average secondary forest biomass accumulation rates in the first 20 years of regrowth compared with uptake rate of old growth forests. CABLE captures this distinction: mean above-ground biomass accumulation rates in the first 20 years of regrowth of 0.26±0.06 (1σ) kgC m$^{-2}$ y$^{-1}$, compare well with the mean of the observations of 0.31±0.13 (1σ) kgC m$^{-2}$ y$^{-1}$ (Poorter et al., 2016), while simulated old growth forest rates 0.05±0.01(1σ) kgC m$^{-2}$ y$^{-1}$ (1990-2010, Tropical Rainforest and Tropical Seasonal Forest biomes in South America) compare well with estimates of 0.03-0.05 kgC m$^{-2}$ y$^{-1}$ from the Amazon RAINFOR plot network for this period (Brienen et al., 2015).

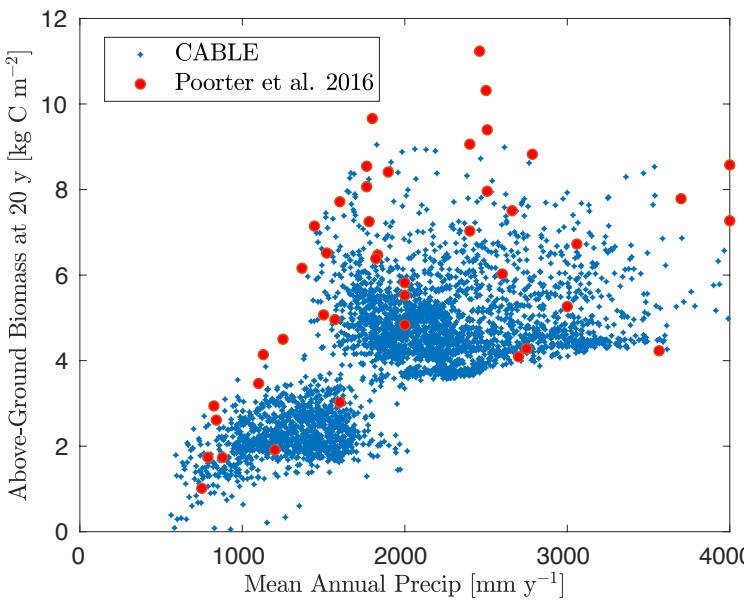

Figure 5: CABLE and observation-based estimates (Poorter et al., 2016) of Neotropical secondary forest biomass after 20 years of regrowth versus mean annual precipitation. CABLE estimates are extracted from secondary-forest tiles in Tropical Rainforest, Tropical Seasonal Forest and Tropical Dry Forest/Savanna biomes (Figure 2) in South America. The lower distinct cloud of CABLE simulated values corresponds to the Tropical Dry Forest/Savanna biomes.

**5.3 Land-use change and forest change: illustrative examples.**

Four examples of contrasting regional land-use histories (0.5$^{o}$ x 0.5$^{o}$ grid cells) are presented to illustrate carbon pool changes and the rate of land-atmosphere carbon flux from 1860-present (Figure 6). The landscape-scale responses reveal details that are obscured in the subsequent aggregation to regional and global scale (Section 5.4), but are important for demonstrating the functionality of the model at the spatial scale at which it is applied.

Each column in Figure 6 corresponds to one site, and the four rows show: (1) land-use transition rates: clearing (p→g + s→g), abandonment (g→s), primary forest harvest (p→s) and secondary forest harvest; (2) land-use area fractions: partitioned into primary-woody, secondary-woody and open land. Open land is further partitioned into cropland, pasture, and the remainder comprising rangeland and "natural grass" meaning all other non-woody vegetation; (3) carbon stocks associated with soil and vegetation for each land-use type and in product pools; (4) land carbon flux to the atmosphere split into gross emissions (positive terms) and gross sinks (negative terms).

**Brazil (first column).** The land-use history for this grid-cell is dominated by clearing of primary forest with peak clearing events in 1940 and 1960 corresponding to respective conversion to rangeland and cropland. The 1860 carbon stocks are partitioned approximately equally between soil and vegetation. Cumulative carbon loss of 30 kgCm$^{-2}$ is dominated by the vegetation carbon stock lost, with additional loss of soil carbon following conversion of forest to cropland, and is only marginally offset by net carbon gains due to differences in the effects of climate and CO$_2$ drivers on

the actual versus baseline land use. The land-atmosphere flux components indicate that the interaction flux (dominated by the loss of additional sink capacity) largely cancels $F_{CC}$ when all forest has been cleared. As such, the net flux ($F_{CC,L}$) closely tracks $F_{LUC,0}$.

**Papua New Guinea (second column).** The land-use history is dominated by shifting cultivation (s→c): secondary forest clearing and abandonment track each other closely for the whole time-series. There is also additional non-s→c clearing and harvest post 1950. This leads to land-area fractions that are largely constant, except for a small decrease in primary forest area post-1950 and associated expansion of crop-land and secondary forest area. Similar to the Brazil example, 1860 carbon stocks are partitioned approximately equally between soil and vegetation. The total carbon stock, and particularly carbon in primary forest vegetation, increases over the time-series because of cumulative carbon uptake in response to the combined effects of $CO_2$ and climate. Land-use change emissions from shifting cultivation are close to zero since emissions from s-c clearing are approximately balanced by regrowth. As such the net flux $F_{CC,L}$ closely tracks $F_{CC}$, with small additional contributions from agricultural management and wood harvest.

**France (third column).** There is no primary forest. Land-use activity is dominated by secondary forest harvest pre-1920, and abandonment of pasture. The cessation of harvest leads to significant carbon accumulation in secondary forest vegetation post-1920. Of the total carbon accumulation since 1860 (7 kg C m$^{-2}$), 4 kgCm$^{-2}$ is attributable directly to LUC (first from forest regrowth post-harvest (pre-1940) and then from regrowth post-abandonment (post-1940)), and the remainder to $CO_2$-climate effects.

**Poland (fourth column).** This is a landscape dominated by agricultural activity. All secondary forest is cleared by 1900, however abandonment of cropland post-1945 leads to an expansion of secondary forest land. Carbon stocks in vegetation are very low because of secondary forest harvest. Soil carbon in open land is depleted because of cropland management (tillage and removal of biomass). The cumulative carbon loss from 1860 is 4 kgCm$^{-2}$, and this is dominated by the direct effect of LUC. At the beginning of the time-series, emissions to the atmosphere are dominated by contributions from cropland management and forest clearing (including legacy effects). From 1980, the land is a sink because carbon uptake by forest regrowth post-harvest and $CO_2$-climate effects outweigh gross LUC emissions.

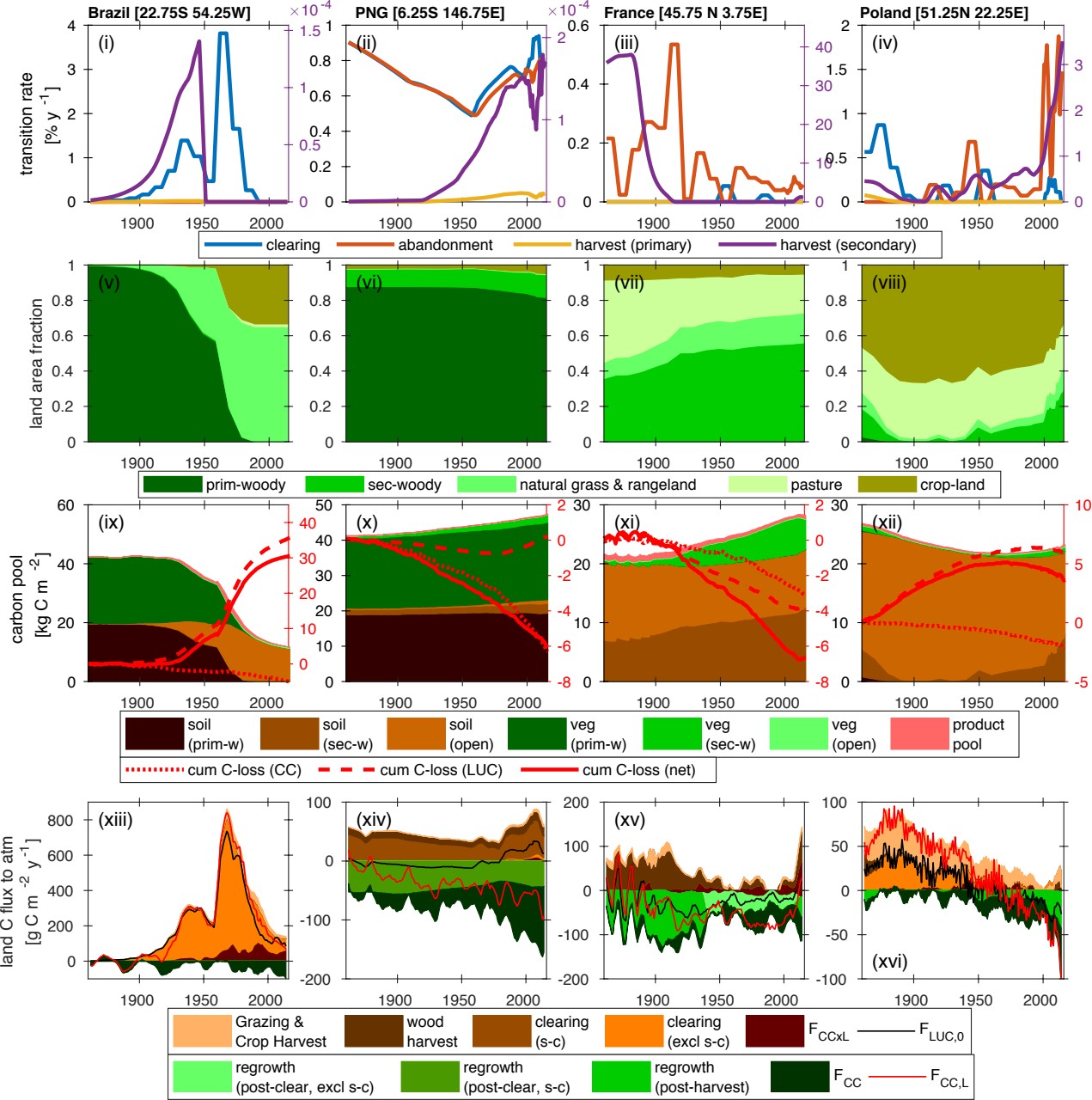

Figure 6: Contrasting land-use and land-management for sample $0.5^{o} \times 0.5^{o}$ grid-cells in Brazil, Papua New Guinea (PNG), France and Poland. (i)-(iv) land area transition rates: clearing (p→g + s→g), abandonment (g→s), primary forest harvest (p→s) and secondary forest harvest; (v)-(viii) land-cover fractions; (ix)-(xii) vegetation stocks in soil (including litter), vegetation and product pools, and cumulative total carbon loss to the atmosphere from combined climate-$CO_2$ (CC) effects, land-use change and land management, and the net effect of all drivers together; (xiii)-(xvi) net land-atmosphere carbon flux ($F_{CC,L}$), and its components. Positive components are the contributions from land-use change and land-management. "Grazing & Crop Harvest" refers to the net carbon flux associated with these activities, as derived by subtraction of net fluxes simulated with and without grazing and crop management. Wood harvest and clearing include legacy emissions and decay of product pools. Clearing and regrowth fluxes associated with shifting cultivation (s-c) are resolved from fluxes not associated with s-c. Regrowth on secondary forest land is resolved from legacy effects of past land use by differencing simulated net ecosystem production on secondary forest tiles, as simulated under scenarios of harvest and clearing residues being extracted to product pools versus residues left as litter. Five year smoothing is applied for clarity.

**5.4 Land-use-change and forest-change: global implications.**

Figure 7 shows the combined impacts of changing climate and $CO_2$, and land-use change on the global terrestrial carbon cycle, and three broad latitudinal bands: tropics (-30° – 30°); extra-tropics of the northern hemisphere (NH) (>30°); and extra-tropics of the southern hemisphere (SH) (<30°) in which land-use activities have affected the trajectory of the net land carbon sink very differently.

**Tropics (first column).** The net effect of clearing and abandonment has been a decline in forest area $6.7 \times 10^6$ km$^2$ since 1860, with clearing emissions peaking in 1954. Forest harvest (degradation) has also been a feature since 1950, and has accelerated steeply in recent decades. Cumulative sources and sinks are approximately equal, yielding negligible change in carbon stocks since 1860. Shifting-cultivation (s→c) is a key feature of land-use: it is useful to resolve the s→c components of the clearing and abandonment fluxes since these approximately cancel each other. The interaction flux $F_{CC \times L}$, which is dominated by loss of additional sink capacity, contributes 30 PgC to the total cumulative loss of carbon by land-use change (176 Pg C since 1860).

**Extra-tropics NH (second column).** Forest area has declined by $3.3 \times 10^6$ km$^2$ since 1860. Although the loss of primary forest areas ($9.3 \times 10^6$ km$^2$) is similar to tropical primary forest loss ($9.6 \times 10^6$ km$^2$), cumulative carbon loss from LUC is much less (92.4 Pg C) because primary vegetation carbon stocks are smaller, and those lost have been largely replaced by regrowth. Net emissions became negative (i.e., net carbon sink) in 1954, and the increasing sink trend is dominated by effects of $CO_2$ fertilisation and lengthening growing season, with net LUC emissions approximately constant and very close to zero in recent decades.

**Extra-tropics SH (third column).** This region has been subject to particularly aggressive deforestation, with $1.0 \times 10^6$ km$^2$ (or one third) of primary forest lost since 1860. Deforestation peaked and declined rapidly in 1953, and was succeeded by a period of increasing forest harvest. In contrast to the other regions, cumulative carbon loss since 1860 (7 PgC) is a significant fraction (8%) of the 1860 carbon stocks. The region has been a sink in recent decades due to the combined effects of $CO_2$-fertilisation and agricultural abandonment.

**The Globe (fourth column).** Global primary forest area has decreased by $20.0 \times 10^6$ km$^2$, while secondary forest area has increased by $9.3 \times 10^6$ km$^2$ since 1860. Cumulative LUC emissions are 287 PgC since 1860 (243 PgC in the absence of interactions between $CO_2$-climate and LUC drivers), and have been counteracted by a cumulative $CO_2$-climate-driven sink of 305 PgC. Cumulative LUC emissions in the absence of interactions between $CO_2$-climate and LUC drivers are 243 PgC, and this is comparable with the BLUE book-keeping model (261 Pg C, 1850-2005) (Hansis et al., 2015) and is within the range of recent estimates (171-295 PgC) by other models that account for gross land-use transitions, as compiled by Hansis et al. (2015).

LUC emissions have been declining steadily since 1960 (albeit with a slight upturn since 2005), while the $CO_2$-climate-driven sink is increasing rapidly and dominates the trend in the net flux. The simulated present day (2012-2016 mean) global land-atmosphere flux of -2.2 PgCy$^{-1}$ is the balance between sources (4.1 PgCy$^{-1}$) and sinks (-6.3 PgCy$^{-1}$). Sources comprise: $F_{CC \times L}$ (0.80), including loss of additional sink capacity $F_{LASC}$ (0.51); clearing excluding s→c (1.12); clearing s→c (0.59); wood harvest (1.20); crop and pasture management (0.40). Sinks comprise: post-clearing regrowth excluding s→c (-0.38); post-clearing regrowth (s→c) (-0.55); post-harvest regrowth (-0.87); $F_{CC}$ (-4.52).

While the $F_{CC}$ term dominates the sink, no sink or source tem is negligible, and the $F_{CC \times L}$ term (itself dominated by the loss of additional sink capacity) is large, pointing to the need to model the effects of land-use, climate and $CO_2$ on terrestrial carbon stocks explicitly and simultaneously, as we have done here.

Table 3 shows that CABLE's partitioning of the net land-carbon sink between the tropics and NH extra-tropics accords well with a recent synthesis by Schimel et al. (2015), which utilised atmospheric inversion data (selected according to

assessment against aircraft vertical profile observations), biomass inventory data, and an ensemble of model estimates of global land carbon uptake in response to rising $CO_2$. Both estimates agree that the strong $CO_2$-driven sink in the tropics is largely cancelled by net deforestation emissions, leaving the NH extra-tropics as the region contributing most to the net land sink, a result also supported by top-down estimates from CarbonTracker Europe (van der Laan-Luijkx et al.,

5    2017). Note however a stronger tropical $CO_2$ fertilisation effect in CABLE than estimated by Schimel et al. (2015). CABLE's high simulated $CO_2$ fertilisation effect in tropical forests is consistent with growth rates in mature forests in Amazonia (Brienen et al., 2015) (See also Section 5.2).

**Table 3: The net land carbon sink [Pg C y$^{-1}$] (1990-2007) and its partitioning, as estimated by CABLE, and a**
10   **synthesis using a combination of top-down and bottom-up constraints (Schimel et al., 2015)**

|  | This work (CABLE) | Schimel et al. 2015 |
|---|---|---|
| Tropical gross deforestation (including harvest) | 2.6 | 2.9 ± 0.5 |
| Tropical regrowth | 1.3 | 1.6 ± 0.5 |
| Net tropical deforestation (including harvest) | 1.3 | 1.3 ± 0.7 |
| Northern extra-tropical uptake | 0.8 | 1.2 ± 0.1 |
| Tropics + SH net uptake (excluding net tropical deforestation) | 1.8 | 1.4 ± 0.4 |
| Net global land uptake | 1.3 | 1.3 ± 0.8 |

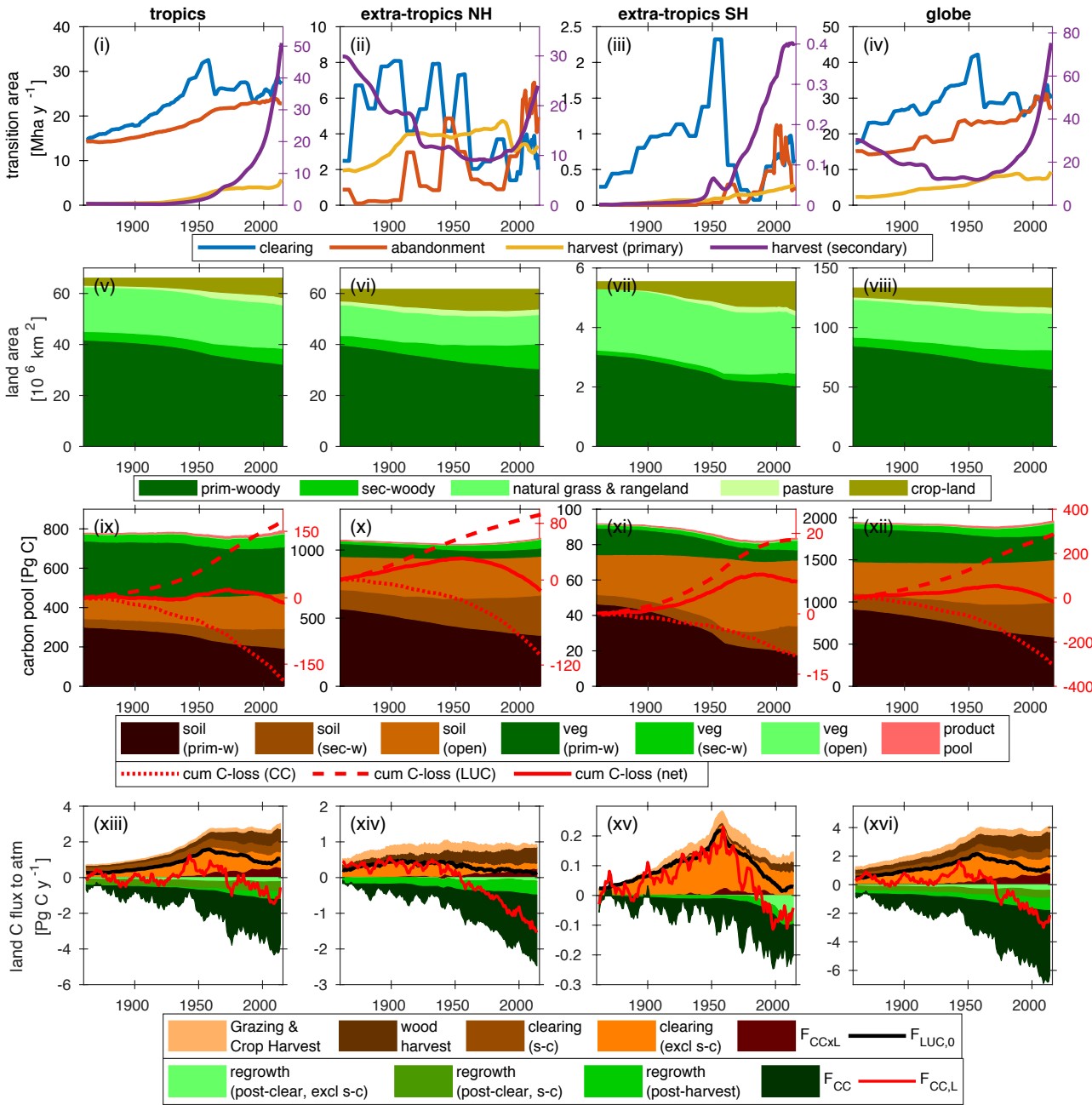

Figure 7: The global terrestrial carbon balance (1860-2016) and its partitioning, as influenced by LUC, land management, $CO_2$ and climate. Columns refer to (a) tropics (-30° – 30°); (b) extra-tropics of the northern hemisphere (NH) (>30°); (c) extra-tropics of the southern hemisphere (SH) (<30°); (d) Globe, excluding permanent ice. Rows refer to the same quantities as in Figure 6. Five year smoothing is applied for clarity.

## 5.5 Coordination of Leaf Photosynthesis: illustrative examples

The effect of dynamically optimising the ratio of $J_{max}$ to $V_{cmax}$ ($b_{JV}$), compared with a fixed value of $b_{JV}$ =1.7 (Walker et al., 2014), over the course of one year for shaded leaves in two contrasting biomes: tropical forest and tundra, is presented in Figure 8. While optimising $b_{JV}$ only slightly increases net-photosynthesis, it significantly reduces variability in the fraction of Rubisco-limitation, compared with the assumption of fixed $b_{JV}$. Periods of near-exclusive electron transport-limitation (fractional Rubisco-limitation close to zero) are avoided when $b_{JV}$ is optimized. Critical to the $CO_2$ fertilisation effect on photosynthesis, this affects the sensitivity of net photosynthesis with respect to $c_s$ because the

electron transport-limited rate is less sensitive to $c_s$ than the Rubisco-limited rate. The proportional change in $A_n$ per proportional change in $c_s$ is demonstrated using the dimensionless elasticity variable $\eta$ (Figure 8(iii) and 8(vii)):

$$\eta = \frac{\partial A_n}{\partial c_s} \frac{c_s}{A_n} \qquad (5)$$

5 Low values of elasticity occur when electron-transport limitation dominates.

In the tropics, the dynamic values of $b_{JV}$ reflect higher investment of nitrogen in $V_{cmax}$ in the dry season (around days 200-300) when absorbed irradiance is higher, whereas in the Tundra, higher investment in $J_{max}$ occurs at the height of the growing season because of the different temperature responses of $J_{max}$ and $V_{cmax}$. Overall, the effect of dynamically optimising $b_{JV}$ is to make electron transport- and Rubisco-limited rates approximately co-limiting, in agreement with

10 experimental evidence (Maire et al., 2012). The effect of increasing $c_s$ is to increase allocation of leaf nitrogen to $J_{max}$, resulting in reduced $V_{cmax}$. At constant $N_{eff}$, the magnitude of the reduction is 10.4% (Tropics) and 12.9% (Tundra) for an increase in $c_s$ from 366 ppm to 567 ppm, in good agreement with CO2-acclimation effects on $V_{cmax}$ inferred from Free Air $CO_2$ Enrichment studies (~10% reduction for an increase in $c_a$ from 366 ppm to 567 ppm) (Ainsworth and Rogers, 2007).

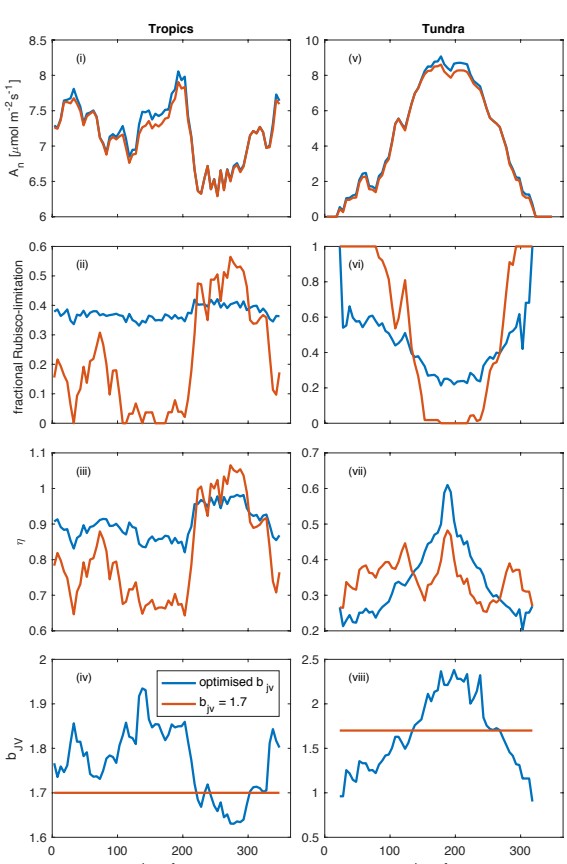

Figure 8: Illustrative simulations of net photosynthesis, fractional Rubisco-limitation, elasticity of net photosynthesis with respect to surface [CO2] and $b_{JV} = J_{max,0}/V_{cmax,0}$ for shaded leaves in a tropical forest environment [2.25°S, 63.2°E] (i)-(iv) and a tundra environment [61.75°N, 75.75°W] (v)-(viii), aggregated over 5-day periods. Simulations were

20 performed under assumptions of dynamically optimized and fixed $b_{JV}$ for a 365-day period (1990 meteorology, 400 ppm CO2.).

**5.6 Dynamic optimization of $b_{JV}$: implications for centennial trend in global photosynthesis**

The impacts of optimising $b_{JV}$ on fractional Rubisco-limitation and centennial increase in global GPP are shown in Figure 9. Simulations using a fixed value of $b_{JV} = 1.7$ (solid blue line), and $b_{JV} = 1.7 \pm 0.1$ (limits of dark shading) and $b_{JV} = 1.7 \pm 0.2$ (limits of light shading) reveal that a static value of $b_{JV}$ translates to highly unpredictable fractional Rubisco-limitation with possible values covering almost the full range from 0 to 1 at every latitude. In contrast, the fractional Rubisco-limitation that is simulated when $b_{JV}$ is dynamically optimized has a value that is approximately 0.5 (corresponding to co-limitation) at all latitudes. Poor prediction of fraction Rubisco-limitation under the assumption of fixed $b_{JV}$ translates to a wide range of GPP increase (1900-2015) relative to values in 1900, with simulated relative increases spanning a range of ~0.2 at most latitudes. Dynamic optimization of $b_{JV}$ results in predictions of centennial increase in GPP that are in good agreement with a recent estimate that uses atmospheric carbonyl sulfide (COS) (Campbell et al., 2017) as a constraint.

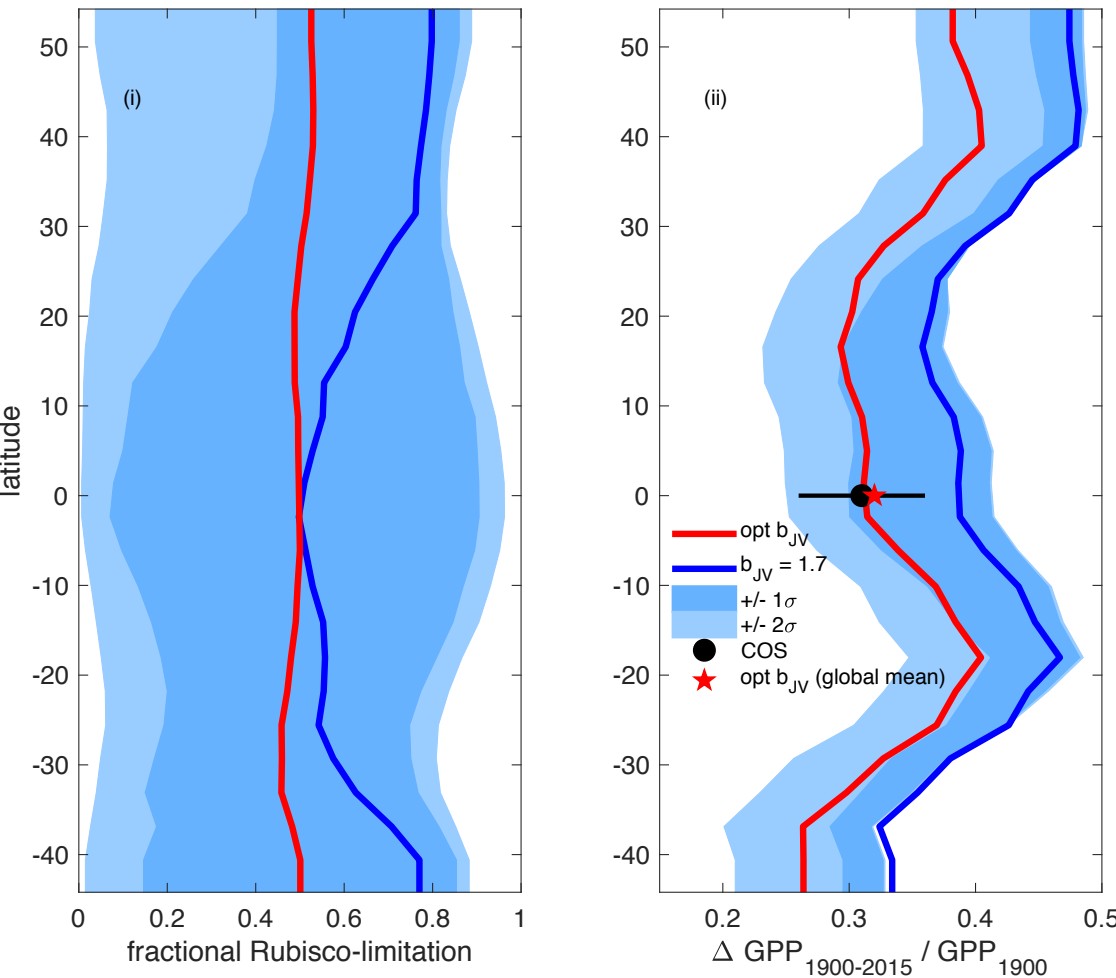

Figure 9. Latitudinal profiles of (i) fractional Rubisco-limited photosynthesis (1980-2015) and (ii) the increase in gross primary production (GPP), relative to 1900 values. Simulations were performed under assumptions of dynamically optimized (red) and fixed $b_{JV} = 1.7$ (blue). For computational efficiency, profiles are based on a sample of 1000 randomly distributed grid-cells across the global ice-free land-surface. The limits of the blue shaded areas represent the results of simulations performed with fixed $b_{JV} = 1.7 \pm 0.1$ (dark shading) and $b_{JV} = 1.7 \pm 0.2$ (light shading), with lower limits corresponding to lower fractional Rubisco-limitation and lower increase in GPP. In panel (ii), the 'COS' value represents the trend in global GPP inferred from the carbonyl sulfide tracer (Campbell et al., 2017)..

**5.7 The global net land carbon sink**

Key functions of global terrestrial biosphere models such as CABLE are the attribution and projection of the global net land carbon sink. Therefore we assess CABLE predictions against observation-based estimates of this important quantity. Figure 10 depicts simulated annual times series of the global land carbon sink from CABLE and the corresponding Global Carbon Project (GCP) estimate, diagnosed as the sum of atmosphere and ocean sinks, minus fossil fuel emissions (Le Quéré et al., 2016). Of the 14 land models represented in the GCP's 2016 assessment of the global carbon budget (Le Quéré et al., 2016), the five contributing simulations of the net land carbon sink (as opposed to the residual land sink, equivalent to the net land sink plus net LUC emissions, represented by all land models) are also shown in Figure 10. For each model, correlation of annual values with GCP estimates (1959-2015), trend (1980-2015) and magnitude (2006-2015) are quantified in Table 4. Uncertainty on the GCP estimates is 0.4 Pg C$y^{-1}$ (Le Quéré et al., 2016). CABLE captures 57% of the variance in the annual sink, simulates a trend that is very similar to the GCP estimate (0.067 Pg C $y^{-2}$ vs 0.061 Pg C $y^{-2}$) and simulates a mean sink for the (2006-2015) period that is 0.5 PgC$y^{-1}$ higher than GCP (2.7 Pg C $y^{-1}$ vs 2.2 Pg C $y^{-1}$). One contribution to this discrepancy could be that the area of tropical forest degradation (p→s or secondary forest harvest) may be under-estimated in the LUH2 forcing data-set. In particular, CABLE simulations for the present day (2012-2016) indicate that forest degradation (secondary harvest) contributes 33% to gross carbon losses from harvest and clearing tropical forests (Figure 8(iii)), compared with 69% (including forest disturbances such as fire) suggested by a recent remote sensing-based estimate by Baccini et al. (2017).

CABLE captures a high proportion of the variance in the GCP estimate, relative to the other models in Table 4. This is in part attributable to its relatively good representation of the 1973-1974 and 1975-1976 positive anomalies corresponding to very strong La Niña events. Moisture sensitivities of both productivity and decomposition are important for capturing the response of the net flux to such events: in particular the high temporal correlation of heterotrophic respiration with NPP in water-limited environments reduces the response of the net flux compared with the response of NPP (Haverd et al., 2016c).

In contrast, CABLE under-predicts large negative anomalies corresponding to 1987-1988 and 1997-1998 El Niño events. Possible explanations are that wildfire is not represented, and the simulated drought response of tropical forests may be too weak.

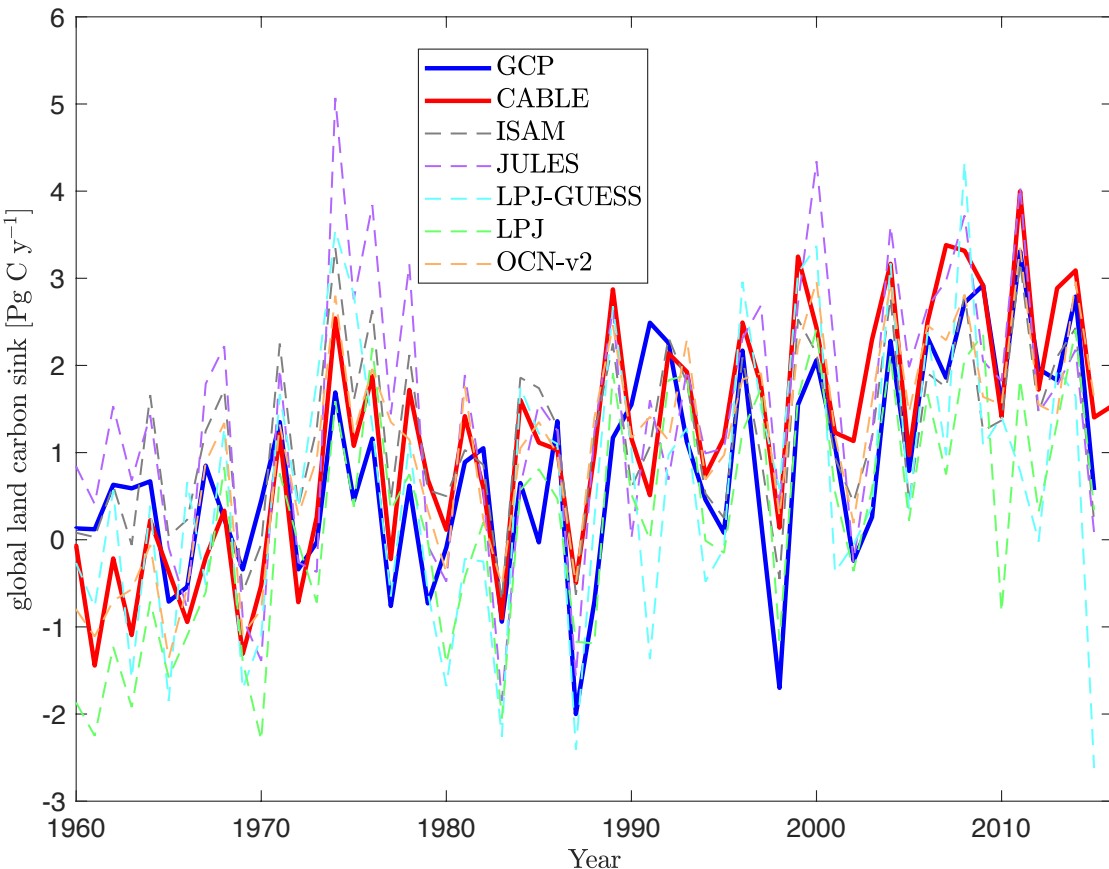

Figure 10. Global land carbon sink, as predicted by CABLE and five terrestrial biosphere models contributing to TRENDY-v5 (Le Quéré et al., 2016), and the Global Carbon Project (GCP) estimate, as the sum of atmosphere and ocean sinks, minus fossil fuel emissions (Le Quéré et al., 2016).

**Table 4. Simulated annual time-series of the global land carbon sink: correlation with GCP, linear trend and mean sink.**

|  | Correlation with GCP ($R^2$) 1959-2015 | Linear trend [Pg C $y^{-2}$] 1980-2015 | Mean Sink [Pg C$y^{-1}$] 2006-2015 |
|---|---|---|---|
| GCP | - | 0.061±0.02 | 2.18 |
| CABLE | 0.57 | 0.067±0.01 | 2.66 |
| JULES | 0.46 | 0.063±0.02 | 2.28 |
| ISAM | 0.56 | 0.031±0.01 | 1.85 |
| LPJ-GUESS | 0.32 | 0.044±0.03 | 1.19 |
| LPJ | 0.48 | 0.052±0.02 | 1.22 |
| OCN-v2 | 0.49 | 0.051±0.01 | 2.17 |

**Carbon-Climate Sensitivity**

10  We evaluate the global land carbon-climate sensitivity, following the analysis by Piao et al. (2013) of 10 terrestrial biosphere models. A linear model relating anomalies in the annual detrended land carbon sink ($y_{sink}$) to anomalies in annual detrended temperature ($x_T$) and precipitation ($x_P$) and an error term $\varepsilon$:

$$y_{\text{sink}} = \gamma_{IAV}^{T} x_{T} + \gamma_{IAV}^{P} x_{P} + \varepsilon \qquad (6)$$

Equation (6) was fitted to CABLE-simulated annual anomalies in net carbon uptake. Results are given in Table 5, and show good agreement with analysis of the Residual Land Sink by Piao et al. (2013). Note the Residual Land Sink (equivalent to the net land sink plus net LUC emissions) is expected to have very similar interannual variations to the net land sink.

Table 5: Interannual global carbon-climate sensitivities, as defined by Equation (6)

| | $\gamma_{IAV}^{T}$ [Pg C y$^{-1}$ K$^{-1}$] | $\gamma_{IAV}^{P}$ [Pg C y$^{-1}$ per 100 mm] | reference |
|---|---|---|---|
| CABLE net land sink (1980-2009) | -3.0 ± 0.5 | 0.7 ± 0.3 | This work |
| Residual Land Sink (1980-2009) | -3.9 ± 1.1 | 0.8 ± 1.1 | Piao et al. (2013) |
| Multi-model range (1980-2009) | -5.1 to -1.0 | 0.4 to 6.0 | Piao et al. (2013) |

## 7. Conclusion and Future Directions

We have presented CABLE model developments that improve its applicability as a terrestrial biosphere model for use within an Earth System Model, and in stand-alone applications to attribute trends and variability in the terrestrial carbon cycle to regions, processes and drivers. Model evaluation has shown that the new model version satisfies several key observational constraints, including (i) trend and interannual variations in the global land carbon sink, including sensitivities of interannual variations to global precipitation and temperature anomalies; (ii) centennial trends in global GPP; (iii) co-ordination of Rubisco-limited and electron transport-limited photosynthesis; (iv) spatial distributions of global ET, GPP, biomass and soil carbon; and (v) secondary forest rates of biomass accumulation in boreal, temperate and tropical forests.

Model evaluation highlighted a few discrepancies that warrant further investigation: (i) under-prediction of ET in tropical forests in Amazonia; (ii) Over-prediction of GPP in SH temperate evergreen broadleaf forests; (iii) under-prediction of large negative anomalies in the global land carbon sink, corresponding to 1987-1988 and 1997-1998 El Niño events.

Further work on the model configuration presented here should include formal benchmarking in the International Land Model Benchmarking Project framework (Hoffman et al., 2017) and model-data fusion (Trudinger et al., 2016). The latter would aim to quantify data constraints on the regional and process attribution the global land carbon sink using multiple parameters sets that are consistent with the observations, in the same way that Trudinger et al. (2016) did for the Australian region. Data for this task would comprise observation-based constraints presented in this work, extended for example to include remotely-sensed vegetation cover.

Priorities for further process enhancement are (i) wildfire impacts on vegetation and related emissions; (ii) explicit cropland management; (iii) dynamic biogeography and PFT-interactions; and (iv) dynamic allocation of carbon that optimises plant fitness.

**Code Availability**

The source code can be accessed after registration at https://trac.nci.org.au/trac/cable. Simulations in this work used Revision Number 4546.

5 **Competing interests.**

The authors declare that they have no conflict of interest.

**Acknowledgements**

VH, CMT, PRB and JGC acknowledge support from the Earth Systems and Climate Change Hub, funded by the Australian Government's National Environmental Science Program. BS contributions to this work were supported by the 10 Strategic Research Area MERGE and by a CSIRO Distinguished Visiting Science grant. We thank Graham Farquhar for helpful discussions about the optimisation-based approach to simulating plant coordination of photosynthesis.

**Appendix 1: CABLE Pseudo-Code**

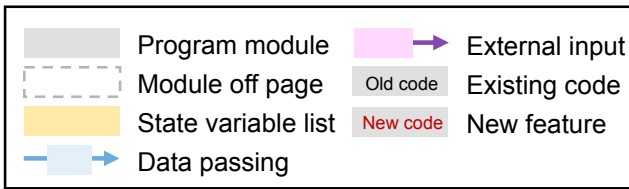

**CABLE Biophysics**

**State Variables:** Soil moisture and temperature in 6 vertical layers; snow water equivalent (up to 3 layers); canopy interception store.

Initialise parameter and state variables
**Main time step loop (sub-diurnal)**
- Read subdiurnal meteorology ← Meteorology data
- Compute surface roughness characteristics
- Compute albedo of canopy and back-ground
- Canopy radiation transfer: Compute canopy extinction coefficients for beam and diffuse radiation; canopy reflectances; fractions beam and diffuse incoming radiation; short-wave radiation absorption by shaded and sunlit leaves and background (soil or snow); iso-thermal longwave radiation absorption by background and vegetation
- Update canopy water storage and fraction wet canopy and compute throughfall

**Loop over Monin-Obukov atmospheric stability parameter**
- Compute aerodynamic properties: friction velocity and turbulent resistances required to compute the dispersion matrix (Localised Near Field Theory)
- Compute forced convection boundary layer conductance at leaf surface

**Loop over (dry) leaf temperature** (solves coupled leaf energy balance, stomatal conductance, net photosynthesis)
- Compute free convection boundary layer conductance at leaf surface
- Compute T-dependent $V_{cmax}$ and $J_{max}$ for shaded and sunlit leaves, accounting for extinction through canopy (leaf-to-canopy scaling).
- Compute T-dependent Michaelis Menten constants for Rubisco
- Compute leaf respiration
    - Fixed fraction of $V_{cmax}$ (default)
    - Alternative: temperature acclimation function multiplied by instantaneous T-response
    - Option: modify for photo-inhibition
- Solve coupled equations for net photosynthesis and stomatal conductance
- Compute root-water extraction and update soil-moisture modifier to stomatal conductance
**Check for convergence**

- Update dry leaf surface energy balance
- Compute leaf wet leaf energy balance, including wet leaf temperature
- Update canopy energy balance
- Compute soil surface energy balance (long-wave component depends on canopy energy balance above)
- Compute dispersion matrix, and update in-canopy temperature and humidity
- Recompute Monin-Obukhov stability parameter
**Next stability iteration**

- Soil physics: update vertical distribution of heat and water content in soil and snow and compute surface runoff and deep soil drainage
- Update climate history variables as required for phenology, acclimation of respiration, optimization of $J_{max}/V_{cmax}$
- Update daily aggregates of GPP, soil temperature and moisture for use in biogeochemistry
- If end of day: Call driver for CASA-CNP Biochemistry
- If end of year: Call drivers for POPLUC (land-use change) and POP (woody demography)
**Next sub-diurnal time step**

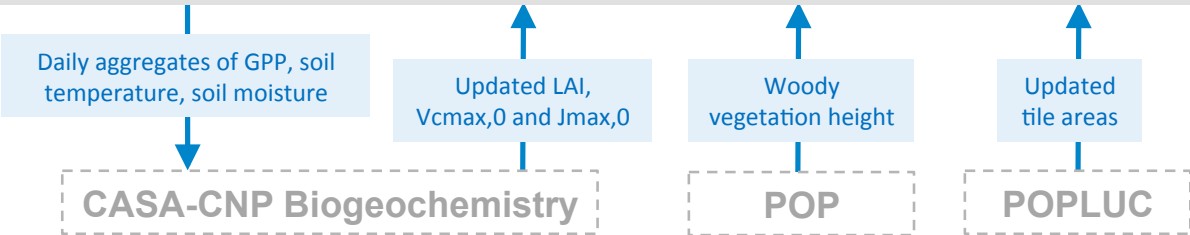

Figure A1a: CABLE Biophysics

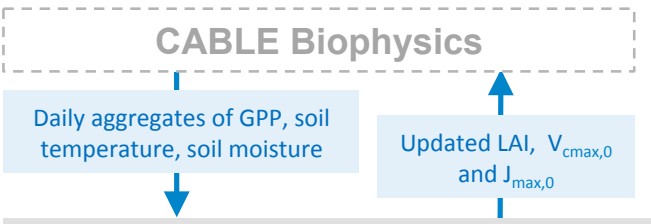

## CABLE Biophysics

Daily aggregates of GPP, soil temperature, soil moisture

Updated LAI, $V_{cmax,0}$ and $J_{max,0}$

### CASA-CNP Biochemistry

**State variables:** C, N, P pools in each of 3 plant compartments (leaves, fine roots, wood); 3 litter compartments (metabolic litter, fine structural litter, coarse woody debris); 3 soil compartments differing by turnover time (fast, slow, passive); soil mineral N and P pools; soil occluded P pool; labile C pool.

**Main time step loop (daily)**
- Get leaf phenology phase for deciduous pfts based on remote-sensing climatology or climate history
- Construct root-weighted soil temperature and moisture variables from vertical profiles.
- Evaluate autotrophic growth and maintenance respiration fluxes for leaves, stems (sapwood only) and fine-roots based on tissue nitrogen content. Assumed Lloyd and Taylor (1994) T-dependence. Option for acclimation based on temperature of warmest quarter, similar to acclimation of leaf respiration.
- Compute modifier to leaf base turnover rate based on cold and/or drought stress. For deciduous pfts, reduce or accelerate leaf turnover based on phenological phase.
- Calculate turnover rates of plant pools and fraction of plant turnover entering litter pool. For woody pfts, wood turnover rate is inherited from POP demography module.
- Check if soil nutrient supply can meet the plant uptake demand: otherwise reduce NPP
- Set allocation coefficients to partition NPP between leaves fine roots and wood. For woody pfts, relative leaf and woody allocation coefficients are based on leaf-area to sapwood-area ratio, with sapwood area inherited from POP demography module.
- Compute temperature- and moisture-modifiers to base turnover rates of soil and litter carbon. New options to use Trudinger et al. (2016) moisture response and Lloyd and Taylor (1994) temperature response.
- Calculate turnover rates of plant, soil and litter carbon pools and the transfer coefficients between different pools
- Computing the reduction in litter and SOM decomposition when decomposition rate is N-limiting
- Compute N and P uptake by plants and allocation of each to plant compartments
- Update C, N and P stores according to turnover rates, NPP, allocation coefficients and transfer coefficients computed above.
- Augment annual aggregates of carbon allocated to stems; maximum LAI, mean fine-root and leaf carbon pools for use in POP.
- Compute LAI (from leaf carbon store) and Vcmax,0 from leaf N and P stores. Option to use global synthesis (Walker et al. 2014) to relate $V_{cmax,0}$ to leaf N and P. $J_{max,0}$ set to constant (1.7) times $V_{cmax,0}$.
- Adjust prior $V_{cmax,0}$ and $J_{max,0}$ using **OptJV algorithm** to minimize nitrogen cost of net photosynthesis, based on conditions for the last 5 days.
- Return updated LAI, $V_{cmax,0}$ and $J_{max,0}$ to CABLE biophysics

**Next daily time step**

Annual , for woody vegetation tiles: total mortality, sapwood mass & area

Annual, for woody vegetation tiles: Stem NPP, max LAI, mean fine-root & leaf carbon pools

Annually updated C,N,P pools

**OptJV algorithm for optimizing ratio $V_{cmax,0}$/ $J_{max,0}$**

- Define leaf nitrogen available for re-distribution, based on prior estimates of $V_{cmax,0}$ and $b_{JV}=J_{max,0}/V_{cmax,0}$.
- Find the value of $b_{JV}$ that minimizes leaf nitrogen cost per unit net photosynthesis (aggregated over the last 5 days) for each of sunlit and shaded leaves.
- Return to CABLE biophysics the next day's $V_{cmax,0}$ and $J_{max,0}$ for sunlit and shaded leaves, based on updated value of $b_{jv}$.

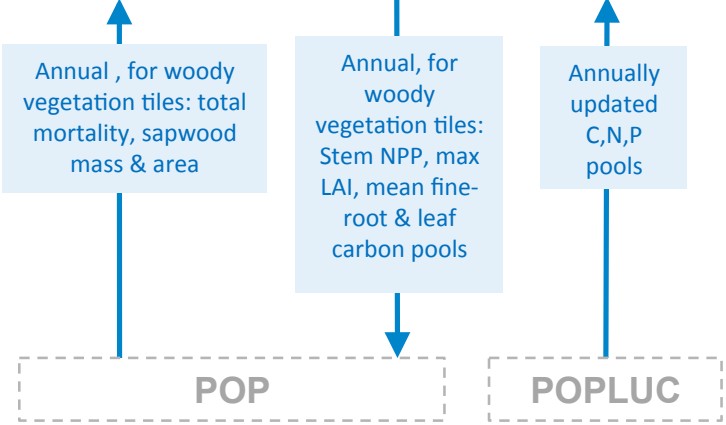

POP

POPLUC

Figure A1b: CASA-CNP Bogeochemistry

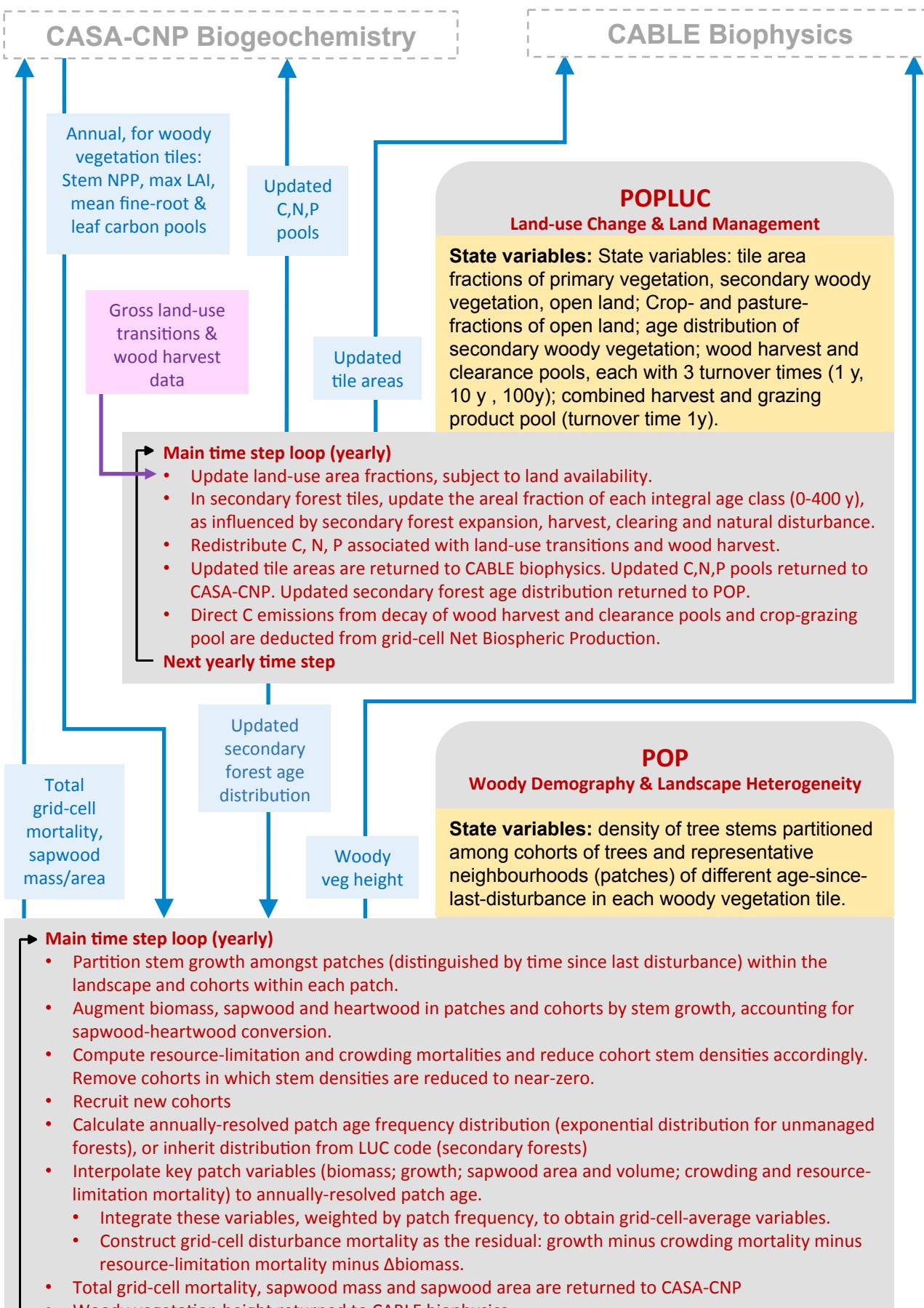

Figure A1c: POP and POPLUC components of CABLE

## Appendix 2: Additional Model Updates

Additional model updates include: (i) drought and summer-green phenology (Sitch et al., 2003; Sykes et al., 1996); (ii) low-temperature reductions in photosynthetic rates in boreal forests (Bergh et al., 1998); (iii) photo-inhibition of leaf day-respiration (Clark et al., 2011); and (iv) acclimation of autotrophic respiration (Atkin et al., 2016). These are described below.

### Drought and summer-green phenology

Prior CABLE predicts phenology based on an annual climatology of remotely-sensed vegetation cover. This precludes simulating the effects of interannual variations and trends in phenology on the terrestrial carbon and water cycles, and land-atmosphere exchange. We addressed this deficiency by implementing drought and summer-green phenology following the LPJ model (Sitch et al., 2003), with extensions to account for chilling requirements of bud-burst (Sykes et al., 1996).

Summer-green phenology applies to deciduous forest types (Decidous Needle-Leaf and Deciduous Broad-Leaf, Table 1) and $C_3$ grass where its growth is temperature-limited. Leaf onset occurs when growing degree days referenced to $5^o$C (GDD) exceed growing degree days to budburst ($GDD^0$). $GDD^0$ is assumed to decline exponentially with the length of chilling period (number of days with mean temperature between $0^o$C and $5^o$C). This relationship represents an adaptation to weather variability: green-up is delayed long enough to minimise the risk that emerging buds will be damaged by frost. The green-up phase ends when $GDD–GDD^0$ exceeds a threshold (set to 200 degree days). The onset of senescence occurs after a fixed period (200 d) of growth.

Rain-green phenology applies to $C_3$ and $C_4$ grass where they are water-limited. No rain-green woody PFTs are represented in CABLE. We define "growing moisture days" (GMD) as the number of consecutive days when an indicator of plant-available soil moisture ($f_{w,soil}$, Eq (1)) exceeds a threshold (set to 0.3). The green-up phase begins when GMD is greater than zero and ends when GMD exceeds a threshold (set to 21 days). Senescence begins when GMD becomes zero.

For both summer-green and rain-green phenology, green-up translates to high allocation of NPP to leaves. Leaf turnover rate is set to zero outside of the senescence period, when turnover time is set to 4 weeks.

### Low-temperature effects on boreal forest photosynthesis

Three processes that contribute to low-temperature reduction of photosynthesis in boreal conifer forests are: (i) reduction caused by frozen soils; (ii) incomplete recovery of photosynthetic capacity during spring; (iii) frost-induced autumn decline. The first effect is largely accounted for in Prior CABLE, because soil moisture limitation on stomatal conductance (Eq (1)) depends on liquid water content, meaning that soil freezing induces soil moisture limitation. Our treatment of the other two processes follows that of Bergh et al. (1998). Rate of post-winter recovery of $V_{cmax,0}$ is held proportional to a degree-day sum referenced to $0^o$C. Recovery is suspended for two days followng a frost event, while a severe frost ($\leq$ -3 $^o$C) also reduces $V_{cmax,0}$. Autumn decline of $V_{cmax,0}$ is simulated by assuming that severe frost nights reduce it progressively and irreversibly until it reaches a 'dormancy' level, where it remains until the onset of spring recovery.

### Photo-inhibition of leaf day respiration

In Prior CABLE, the rate of leaf respiration at standard temperature is assumed the same day and night. However many studies have shown that, at a given temperature, the rate of leaf respiration in daylight is less than that in darkness (Brooks and Farquhar (1985), Hoefnagel et al. (1998), Atkin et al. (1998, 2000)). To account for this, we implement the

inhibition of leaf respiration by light, as demonstrated by Brooks and Farquhar (1985), implemented by Lloyd et al. (1995) and successfully tested in the JULES land surface model for an Amazonian rainforest site by Mercado et al. (2007), and globally by Clark et al. (2011). The light-dependent non-photo-respiratory leaf respiration ($R_l$) is thus:

$$R_l = R_d \qquad\qquad\qquad ; 0 < I_0 < 10\ \mu\text{mol quanta m}^{-2}\ \text{s}^{-1}$$
$$R_l = \left[0.5 - 0.05\ln(I_0)\right] R_d \qquad ; I_0 > 10\ \mu\text{mol quanta m}^{-2}\ \text{s}^{-1}$$

(6)

where $I_0$ is the flux of incoming radiation at the top of the canopy (μmol quanta m$^{-2}$ s$^{-1}$) and $R_d$ is the dark leaf respiration rate.

**Acclimation of Autotrophic Respiration**

Prior CABLE assumes a fixed PFT-dependent value of leaf respiration at standard temperature (25$^{\text{o}}$C), an assumption which may lead to exaggerated latitudinal gradients in leaf respiration rates, as well as exaggerated trends in leaf
respiration as global warming occurs. This is because, with sustained changes in the prevailing ambient growth temperature, leaf dark respiration ($R_d$ [μmol m$^{-2}$ s$^{-1}$]) acclimates to the new conditions, resulting in higher rates of $R_d$ in cold-acclimated plants (Atkin et al., 2016 and references therein). To capture such acclimation effects, we utilise the synthesis of leaf dark respiration rates by Atkin et al. (2016) to parameterise the temperature dependence of leaf respiration at a standard temperature of 25$^{\text{o}}$C ($R_{d,25}$). We then apply the same temperature-acclimation response to root
and stem maintenance respiration rates. Specifically, we use the linear model relating $R_{d,25}$ to $V_{c,max,25}$ and temperature of the warmest quarter ($T_{WQ}$), here the mean temperature of the warmest three-month period during the preceding calendar year.

$$R_{d,25} = c_4 \left[ c_1 + c_2 V_{c\max,0} + c_3 T_{WQ} \right]$$

(6)

In Equation (6), $c_1$-c3 are taken from Atkin et al. (2016, Table S4) and $c_4$ is an additional scaling parameter of order 1,
introduced in this work, with values 0.9 (Evergreen Broadleaf); 1.0 (Deciduous Broadleaf); 1.0 (Evergreen Needleleaf); 1.0 (Deciduous Needleleaf); 0.8 (C3 grass); 0.7 (other).

For consistency with Atkin et al. (2016), we adopt the "variable Q10" instantaneous temperature response of $R_d$ (Tjoelker et al. 2001):

$$R_d = R_{d,25} \left[ 3.09 - 0.043 (T + 25.0)/2.0 \right]^{\frac{T - 25.0}{10.0}}$$

(6)

In the absence of data to inform a general formulation of the temperature acclimation responses of sapwood and fine root maintenance respiration, we formulate them to be consistent with leaf temperature acclimation, but proportional to nitrogen content of the respective compartment:

$$R_{m,sapwood,25} = c_5 N_{sapwood} \left[ c_1 + c_2 V'_{c,\max,0} + c_3 T_{WQ} \right]$$

(6)

$$R_{m,root,25} = c_5 N_{root} \left[ c_1 + c_2 V'_{c,\max,0} + c_3 T_{WQ} \right]$$

(6)

where $c_5$ is a PFT-dependent scaling factor, and $V'_{c,\max,0}$ is the value of $V_{c\max,0}$ obtained with maximum values of leaf N/C and P/C , such that variations in leaf stoichiometry do not affect sapwood and root respiration. As in Prior CABLE, the instantaneous temperature response of Lloyd and Taylor (1994) is assumed.

Appendix A3: CABLE parameters and temperature response functions for photosynthesis used in this work

Table A1: CABLE Biophysics and CASA-CNP Biogeochemistry Parameters for Evergreen Needle-leaf (ENL); Evergreen Broadleaf (EBL); Deciduous Needle-leaf (DNL); Deciduous Broad-leaf (DBL); shrub; C3 grass; C4 grass and Tundra plant functional types.

| Parameter | Description | ENL | EBL | DNL | DBL | shrub | C3 grass | C4 grass | Tundra |
|---|---|---|---|---|---|---|---|---|---|
| $\Gamma_0$ [μmol mol$^{-1}$] | $CO_2$ compensation point | 34.6 | 34.6 | 34.6 | 34.6 | 34.6 | 34.6 | 34.6 | 34.6 |
| $K_{C,0}$ [μbar] | M-M Constant of Rubisco ($CO_2$) | 405 | 405 | 405 | 405 | 405 | 405 | 302 | 405 |
| $K_{O,0}$ [mbar] | M-M Constant of Rubisco ($O_2$) | 278 | 278 | 278 | 278 | 278 | 278 | 256 | 278 |
| $E_{k,C}$ [J mol$^{-1}$] | Activation energy for $K_C$ | 59430 | 59430 | 59430 | 59430 | 59430 | 59430 | 59430 | 59430 |
| $E_{k,O}$ [J mol$^{-1}$] | Activation energy for $K_O$ | 36000 | 36000 | 36000 | 36000 | 36000 | 36000 | 36000 | 36000 |
| $T_{ref}$ [K] | Ref temp for photosynthesis | 298 | 298 | 298 | 298 | 298 | 298 | 298 | 298 |
| $\alpha^{\ddagger}$ [mol mol$^{-1}$] | Quantum yield | 0.28 | 0.28 | 0.28 | 0.28 | 0.28 | 0.28 | 0.04 | 0.28 |
| $\theta$ | Curvature of response of electron transport rate to absorbed photon irradiance | 0.85 | 0.85 | 0.85 | 0.85 | 0.85 | 0.85 | 0.8 | 0.85 |
| $k_N$ | Canopy extinction coefficient for nitrogen | 0.7 | 0.7 | 0.7 | 0.7 | 0.7 | 0.7 | 0.01 | 0.7 |
| $V_{cmax,0}$ scale factor ¶ | maximum catalytic activity of Rubisco | 1.25 | 1.10 | 1.25 | 1.25 | 1.25 | 1.25 | N/A§ | 1.25 |
| $g_1$ | Parameter in response of stomatal conductance to leaf-air vapour pressure deficit | 2.35 | 3.34 (6.0) † | 2.35 | 4.45 | 4.22 | 4.5 | 1.6 | 2.2 |
| Y | Parameter controlling response of stomatal conductance to soil water deficit | 0.03 | 0.03 | 0.03 | 0.03 | 0.03 | 0.03 | 0.03 | 0.03 |
| $z_r$ [m] | Maximum rooting depth | 1.8 | 4.0 | 2.0 | 2.0 | 2.5 | 0.5 | 1.0 | 1.5 |
| $c_{LITT}$ [tC ha$^{-1}$] | Parameter controlling litter-resistance to heat transfer | 20.0 | 3.0 | 10.0 | 13.0 | 2.0 | 2.0 | 0.3 | 0.3 |
| $\chi$ | Parameter controlling leaf angle distribution | 0.01 | 0.1 | 0.01 | 0.01 | 0.25 | 0.01 | -0.30 | -0.30 |
| Leaf dimension [m] | Affects leaf boundary layer conductance | 0.05 | 0.11 | 0.04 | 0.17 | 0.11 | 0.3 | 0.3 | 0.3 |
| Leaf scattering coefficient (PAR). | Sum of leaf reflectance and transmittance. | 0.18 | 0.18 | 0.15 | 0.18 | 0.18 | 0.22 | 0.22 | 0.15 |
| Leaf scattering | Sum of leaf reflectance | 0.60 | 0.58 | 0.60 | 0.58 | 0.60 | 0.64 | 0.64 | 0.64 |

| | | | | | | | | | |
|---|---|---|---|---|---|---|---|---|---|
| coefficient (NIR) | and transmittance. | | | | | | | | |
| Autotrophic respiration scale factor (leaf) | Coefficient ($c_4$) in expression for $R_{d,25}$ (Eq 20) | 1.0 | 0.9 | 1.0 | 1.0 | 1.0 | 0.8 | 0.7 | 0.7 |
| Maintenance respiration scale factor (sapwood and fine roots) | Coefficient ($c_5$) in expression for $R_{m,sapwood,25}$ and $R_{m,root,25}$ (Eqs 22 & 23) | 24 | 6 | 12 | 9 | 12 | 12 | 12 | 12 |
| LA:SA | Leaf area to sapwood area ratio | 5000 | 5000 | 5000 | 5000 | 5000 | N/A | N/A | N/A |
| $\tau_L$ [y] | Leaf turnover time # | 3.7 | 2.1 | N/A | N/A | 1.4 | 0.5 | 0.5 | 0.9 |
| $\tau_R$ [y] | Fine-root turnover time | 0.7 | 0.7 | 0.7 | 0.7 | 0.7 | 0.3 | 0.2 | 0.5 |
| $\tau_{Litt,met}$ [y] | Base turnover time: metabolic litter | 0.01 | 0.01 | 0.01 | 0.01 | 0.01 | 0.01 | 0.01 | 0.01 |
| $\tau_{Litt,str}$ [y] | Base turnover time: fine structural litter | 0.20 | 0.20 | 0.20 | 0.20 | 0.20 | 0.20 | 0.20 | 0.20 |
| $\tau_{Litt,CWD}$ [y] | Base turnover time: coarse woody debris | 1.0 | 5.0 | 1.0 | 1.0 | 5.0 | N/A | N/A | N/A |
| $\tau_{soil,mic}$ [y] | Base turnover time: microbial soil C | 0.13 | 0.13 | 0.13 | 0.13 | 0.13 | 0.13 | 0.13 | 0.7 |
| $\tau_{soil,slow}$ [y] | Base turnover time: slow soil C | 15 | 15 | 15 | 15 | 15 | 15 | 15 | 5 |
| $\tau_{soil,pass}$[y] | Base turnover time: passive soil C | 250 | 250 | 250 | 250 | 250 | 250 | 250 | 250 |
| $\alpha_R$ | Fraction NPP allocation to fine roots. ## | 0.4 | 0.4 | 0.2 | 0.3 | 0.37 | 0.65 | 0.65 | 0.65 |
| $L_{C,L}$ | Fraction of structural C that is in lignin (leaves) | 0.1 | 0.1 | 0.1 | 0.1 | 0.1 | 0.1 | 0.1 | 0.1 |
| $L_{C,wood}$ | Fraction of structural C that is in lignin (wood) | 0.4 | 0.4 | 0.4 | 0.4 | 0.4 | 0.4 | 0.4 | 0.4 |
| $L_{C,R}$ | Fraction of structural C that is in lignin (fine roots) | 0.1 | 0.1 | 0.1 | 0.1 | 0.1 | 0.1 | 0.1 | 0.1 |
| $LAI_{min}$ | Minimum LAI | 0 | 0.5 | 0.5 | 0.35 | 0.35 | 0.1 | 0.1 | 0.3 |
| $LAI_{max}$ | Maximum LAI* | 10 | 10 | 10 | 10 | 10 | 4 | 4 | 3 |
| N:C$_{leaf,min}$ | Minimum N:C ratio (leaf) | 46 | 29 | 23 | 48 | 33 | 23 | 30 | 21 |
| N:C$_{leaf,max}$ | Maximum N:C ratio (leaf) | 70 | 40 | 55 | 48 | 70 | 55 | 30 | 50 |
| N:C$_{wood,min}$ | Minimum N:C ratio (wood) | 476 | 270 | 488 | 312 | 284 | N/A | N/A | N/A |
| N:C$_{wood,max}$ | Maximum N:C ratio | 476 | 270 | 488 | 312 | 284 | N/A | N/A | N/A |

| | | | | | | | | | |
|---|---|---|---|---|---|---|---|---|---|
| | (wood) | | | | | | | | |
| N:C$_{froot,min}$ | Minimum N:C ratio (fine roots) | 120 | 112 | 120 | 120 | 120 | 120 | 63 | 120 |
| N:C$_{froot,max}$ | Maximum N:C ratio (fine roots) | 120 | 112 | 120 | 120 | 120 | 120 | 63 | 120 |
| N:C$_{mic,min}$ | Minimum N:C ratio (soil microbial pool) | 5.4 | 7.7 | 7.1 | 6.7 | 6.2 | 5.7 | 6.0 | 8.0 |
| N:C$_{mic,max}$ | Maximum N:C ratio (soil microbial pool) | 8.0 | 8.0 | 8.0 | 8.0 | 8.0 | 8.0 | 8.0 | 8.0 |
| N:C$_{slo,min}$ | Minimum N:C ratio (slow soil pool) | 26.9 | 13.5 | 26.9 | 16.2 | 16.6 | 11.4 | 13.3 | 20.9 |
| N:C$_{slo,max}$ | Maximum N:C ratio (slow soil pool) | 30 | 30 | 30 | 30 | 20 | 20 | 30 | 30 |
| N:C$_{pass,min}$ | Minimum N:C ratio (passive soil pool) | 26.9 | 13.5 | 26.9 | 16.2 | 16.6 | 11.4 | 13.3 | 20.9 |
| N:C$_{pass,max}$ | Maximum N:C ratio (passive soil pool) | 30 | 30 | 30 | 30 | 20 | 20 | 30 | 30 |

‡ quantum yield for electron transport (C3 plants) or carboxylation (C4 plants)

§ C4 photosynthesis follows Collatz et al (1994). Vcmax,0 set to 10 umolm-2s-1; Jmax,0/Vc,max,0 = 2.0;

† Higher value for tropical evergreen broad-leaf

¶Applied to relationship between $V_{c,max}$, leaf N and leaf P (Walker et al. 2014, Table 3, Model 1).

5   # For both summer-green and rain-green phenology, leaf turnover rate is set to zero outside of the senescence period, when turnover time is set to 4 weeks.

## For both summer-green and rain-green phenology, green-up translates to high (0.95) allocation of NPP to leaves, with allocation to roots correspondingly reduced.

✳ Set arbitrarily high for woody PFTs to allow LAI to be contolled by Pipe Model allocation constraint (to maintain

10   prescribed leaf area to sapwood area ratio).

Table A2: POP Parameters

| Parameter | Description | Value |
|---|---|---|
| α$_{Fulton}$ | Shape parameter in function relating recruitment density to conditions of growth suppression | 3.5 |
| N$_{max}$ [m$^{-2}$] | Maximum density of individuals within a cohort | 0.2 |
| N$_{min}$ [m$^{-2}$] | Minimum density of individuals within a cohort | 10$^{-9}$ |
| K$_{biometric}$ | Constant in height-diameter relationship | 50.0 |
| ρ$_{wood}$ [kg C m$^{-3}$] | Wood density | 300.0 |
| GE$_{min}$ | Growth efficiency threshold below which mortality increases markedly | 0.012 |

| | | |
|---|---|---|
| $P_{mort}$ | Exponent in resource-limitation moratlity formulation | 5.0 |
| $k_{allom}$ | constant in crown area relation to tree diameter | 200 |
| $k_{rp}$ | Power in crown area relation to tree diameter | 1.67 |
| $k_{sapwood}$ [$y^{-1}$] | rate constant for conversion of sapwood to heartwood | 0.05 |
| Ncohort_max | Maximum number of cohorts | 20 |
| Npatch | Number of patches | 60 |
| $\lambda$ | Mean disturbance interval | 100 |

Table A3: Temperature response functions for photosynthesis

| variable | Response function | Response function parameters |
|---|---|---|
| $V_{c,max}$ (C3 plants) ‡ | $V_{c,max} = V_{c,max,0}\left[1+\exp\left[\left(S_v T_{ref} - H_d\right)/\left(RT_{ref}\right)\right]\right]\dfrac{\exp\left[\left(H_a/RT_{ref}\right)\left(1-T_{ref}/T\right)\right]}{1+\exp\left[\left(S_v T - H_d\right)/\left(RT\right)\right]}$ | R = 8.314 $Jmol^{-1}K^{-1}$ <br> $H_a$ =73647 $Jmol^{-1}$ <br> $H_d$ = 149252 J $mol^{-1}$ <br> $S_v$ = 486 $Jmol^{-1}$ |
| $J_{max}$ (C3 plants) ‡ | $J_{max} = J_{max,0}\left[1+\exp\left[\left(S_v T_{ref} - H_d\right)/\left(RT_{ref}\right)\right]\right]\dfrac{\exp\left[\left(H_a/RT_{ref}\right)\left(1-T_{ref}/T\right)\right]}{1+\exp\left[\left(S_v T - H_d\right)/\left(RT\right)\right]}$ | $H_a$ =50300 $Jmol^{-1}$ <br> $H_d$ = 152044 J $mol^{-1}$ <br> $S_v$ = 495 $Jmol^{-1}$ |
| $V_{c,max}$ (C4 plants) § | $V_{c,max} = \dfrac{V_{c,max,0} Q_{10}^{(T-298)/10}}{\left(1+\exp\left[c_1\left(c_2-(T-273)\right)\right]\right)\left(1+\exp\left[c_3\left(T-273-c_4\right)\right]\right)}$ | $Q_{10}$=2.0 <br> $c_1$= 0.3 <br> $c_2$ = 13.0 <br> $c_3$ = 0.2 <br> $c_4$ = 38 |
| $\Gamma_*$ † | $\Gamma_* = \Gamma_{*,0}\left[1+\gamma_1\left(T-T_{ref}\right)+\gamma_2\left(T-T_{ref}\right)^2\right]$ | $Y_1$ = 0.0509 <br> $Y_2$=0.001 |
| M-M Constant of Rubisco ($CO_2$) † | $K_c = K_{C,0}e^{\left(E_{Kc}/\left(RT_{ref}\right)\right)\left(1-T_{ref}/T\right)}$ | |

| M-M Constant of Rubisco ($O_2$) † | $K_O = K_{O,0}e^{\left(E_{KO}/\left(RT_{ref}\right)\right)\left(1-T_{ref}/T\right)}$ | |

‡ (Leuning, 2002)

§ Collatz et al. (1992), modified to match (Massad et al., 2007)

† (Leuning, 1990)

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
