# Peer review of "A new version of the CABLE land surface model (Subversion revision r4546), incorporating land use and land cover change, woody vegetation demography and a novel optimisation-based approach to plant coordination of photosynthesis."

_Geoscientific Model Development, 2017_

## Referee Comment (RC1) · Anonymous Referee #1 · 4 Jan 2018

Haverd et al present a set of updates to the CABLE model, including the "Populations Order Physiology" (POP) model representation of land use, an algorithm depicting photosynthetic optimality principles, and several other updates presented as appendices.

Numerous integrations of the model with different land use and climate drivers are presented, along with a comprehensive model evaluation exercise.

While this is a substantial paper that should almost certainly ultimately be published, and while it includes many interesting a novel benchmarking approaches that the land surface modeling community would do well to take notice of any repeat with other models, I find this version of the manuscript in need of considerable work in terms of the model description presented and in terms of discussion of the uncertainties inherent in both the POP approach and the other updates.

Firstly, the authors rather over-zealous 'selling' of the POP concept in the manuscript strikes me as not particularly objective and thus quite unconvincing. Further, given the lack of critical discussion of the approach, I am left unsure in which circumstances POP might act as an appropriate simplification, and those in which it would not. For example, there is no discussion of how PFT competition might be represented in this framework, nor of how it would respond to the implementation of partial disturbance processes. The somewhat heuristic and undocumented disaggregation of grid cell fluxes into patches and cohorts (which is the critical central assumption) is also presented without any consideration of whether it is realistic or appropriate. I realize that it is imperative to illustrate in some way the basic competence of an LSM, in order to allow the following experiments to be seriously analyzed, but this must be balanced with some humility about how much can really be read into the conclusions, given the vast difficulties of parameterization and appropriate validation of these models.

Secondly, the model description is inadequate and confusing throughout much of the methods section. I have detailed specific instances of this below, but in general, the description is vague, not accompanied with technical equations nor any accompanying documentation, and is not up the standards that are found within a typical GMD article. I suggest that this section needs completely re-writing with transparency and provenance tracking in mind. In my view it requires a full separate technical note to allow proper assessment of the methods employed, which again, would be normal practice

within GMD.

Thirdly, the manuscript focuses in great detail on the POP land use and the photosynthetic optimization modifications, then almost ignores the other myriad of modifications that have been made to the model. Why are these two modifications selected for special treatment? Maybe there is a good reason, but it needs to be made clearer.

Lastly, the paper is essentially presents a new version of CABLE with many updates, but the performance of this new version in contrast to any previous versions is not considered and the impact of the implementation of the different model features is in general ignored, nor is the performance compared to any other LSM. Thus, the skill of this model version is presented in isolation, and is quite difficult to assess other than broadly stating that is performs reasonably well.

Specific Comments P1 L5: Critical for what?

P1 L15: This theory has been proposed previously (Xu et al. 2012), so is not novel

P1 L21: "state of the art" is jargon and should be replaced by a statement with some clear scientific meaning.

P1 L25: I wasn't aware that we had any credible estimates of global GPP, let alone centennial trends therein.

P3 L8: These two developments seems quite arbitrary and distinct from one another. Why are they the joint focus of this one paper?

P3 L10: Now there is a new list of developments. Why is this list different from the last list?

P3 L33: "second generation" in what sense?

P3 L36: Many current DGVM models using some sort of similarity clustering to deal with the problem of expanding numbers of disturbances classes to track. (see all implementations of ED...) This might be difficult, but it is nonetheless the 'state of the art',

if we are going to use that sort of terminology.

P4 L5: POP has some advantages in terms of computational time, but the rules used to disaggregate the big leaf fluxes into size classes of vegetation are necessarily arbitrary. There has not been, as far as I know, any attempt to investigate the uncertainty introduced by not resolving vertical light partitioning in POP. Thus, it is not clear to me that is all that useful of an idea.

P4 L 12: As above, this idea was also proposal by Xu et al. (2012) and it's global implementation presented by Ali et al. (2016)

P5 L11: Define what is meant by 'offline' in this context?

P5 L27: This sentence is really just hype and doesn't add anything of scientific value to the paper.

P5 L30: If the timestep is one year, how is the growth of leaf tissue, and disturbance events from individual fires resolved with appropriate fidelity?

P5 L31: Input variables to POP, not CABLE, I assume.

P5 L35: Surely neglecting partial disturbance from fires and mortality will introduce a large bias? How can this decision be justified?

P5 L37: Need to define the nature of a 'cohort' here. Are they all of similar height, age, dbh? Similarly, are the 'patches' spatially explicit or implicit? To what does the term 'neighborhoods' refer?

P6 L1: Does this just mean that stem biomass is a fixed fraction of NPP?

P6 L2: GPP and Ra at the grid-scale level?

P6 L3-6: This is not an adequate description of the disaggregation process, which is the most important assumption in this POP system. How are gap probabilities evaluated? How is light interception of the different cohorts and patches evaluated? With what

set of assumptions? I think the authors would do well, if they genuinely wish this approach to become more broadly accepted, to apply some more critical thinking to this particular aspect of the model and to be much more transparent with the limitations and strengths of the assumptions here. Maybe it is defensible to assume that NPP is directly proportional to light interception, or maybe it isn't, but the absence of discussion and questioning of this topic is frustrating. I was a referee on the original POP paper too, and continue to find this to be a limiting aspect of this exercise.

P6 L11: I thought there was only catastrophic disturbance?

P6 L13: There should, at the very least, be a reference to the place where one can find an actual description of this mortality function. Growth efficiency is often also defined as NPP/LAI (in LPJ, for example). Hence this needs more careful definition. There is no description at all of how the crowding mortality works.

P6 L 17: In what sense are the patches 'replicates'?

P6 L24: How are the state variables interpolated? This sentence doesn't make sense to me, nor does the one that follows. Is this a new feature of POP? In which case, it needs much, much clearer documentation.

P6 L30: "The resulting tree biomass turnover" : resulting from what? The combination of the mortality rates discussed above?

P6 L32: Thus far the distinction of how CASA-CNP and CABLE interact has not been made clear.

P6 L35: Is this a feature of POP, or of CASA-CNP?

P6 L36: I thought NPP, GPP and Ra were all calculated at the grid scale level, so how can NPP thus be depdendant on stand age?

P7 L 12: What is a biome in this context? Is each grid cell really only populated by one or two PFTs?

P8 L5: How many age classes are there? Is this dynamic or fixed?

P8 L10: I don't think the description in section 3.1 was sufficient to let the reader understand how this interacts with the age structure tracking in secondary forests.

P8 L14: typo in 'pools'

P9 L5: Again, this seems like a huge shift in focus from land use and demography to fast timescale photosynthesis. Further, a very similar method was suggested by the studies of Xu, and Ali.

P11 L23: So, shouldn't the BGC model be CASA-CN, not CASA-CNP?

P11 L31: Which version of CRU-NCEP are you using here?

P12 L9: None of these scenarios explores the impact of the photosynthetic optimization approach that you just documented in considerable detail? This happens later, but that is mixing of methods and results and is confusing.

P15 L4: From where are these successional data taken? Surely there is massive geographical/climatic variance in these rates? Is the model sampled to make sure that it has the same climatic regimes as the dataset?

P16 L27 : Papua New Guinea??

P26 L10: Arguably, if one is going to say 'state of the art', this model should already include fire effects on vegetation, croplands and dynamic biogeography and PFT interactions already, since those are things that are included in many other models. Saying a model is the 'state of the art' is a bold statement, given both the complexity and the wide range of approaches within this field. Further, it is not really necessary for the purpose of model documentation. All LSMs have strengths and weaknesses in different areas. Progress can only be made be careful and objective analysis of the uncertainties inherent in different types of structural assumption, parameters and boundary conditions. I found this paper somewhat lacking in any thoughtful discussion of these

things, whether the model is 'state of the art' or not.

References

Ali AA, Xu C, Rogers A, Fisher RA, Wullschleger SD, Massoud EC, Vrugt JA, Muss JD, McDowell NG, Fisher JB, Reich PB. A global scale mechanistic model of photosynthetic capacity (LUNA V1. 0). Geoscientific Model Development. 2016 Feb 12;9(2):587-606

Xu C, Fisher R, Wullschleger SD, Wilson CJ, Cai M, McDowell NG. Toward a mechanistic modeling of nitrogen limitation on vegetation dynamics. PloS one. 2012 May 23;7(5):e37914.

---

## Referee Comment (RC2) · Anonymous Referee #2 · 17 Jan 2018

Haverd and collaborators present and evaluate the latest developments for the land surface model CABLE. The manuscript contains the information that is expected in such a study and the structure of the manuscript is good. The figures and tables

support the text but the text itself is often too concise which hampers the readability of the manuscript. I made several specific comments to underpin my opinion but I would like to encourage the authors to carefully go through the manuscript and check every paragraph, even those not mentioned in my comments.

Contrary to the text itself, the title is too wordy. A shorter alternative could be "Incorporating gross land cover change, tree-demography and a novel optimization-based photosynthesis in CABLE land surface model (revision 4546)". The words that will no longer be in the title could be moved to the keywords. Words already in the title should not be repeated as keywords as they will result in new hits from search engines.

P4, L16. "is inconsistent with the Co-ordination Hypothesis". Rephrase or better explain. The logic of this sentence appears twisted. As I read this sentence it says that the co-ordination hypothesis differs from the hypothesis that the ratio between Vmax and Jmax is constant, which seems trivial given that the Co-ordination hypothesis was established as an alternative for the fixed-hypothesis. It is more relevant for the reader to be informed whether the co-ordination hypothesis is or isn't at odds with the data.

P4, Section 2. The explanation of the structure of the model would likely benefit from a adding a simplified flowchart-type of figure showing the main dependencies. The actual approaches are often missing and should be added to the text. How is, for example, the radiation transfer through the canopy simulated? Describe the approach in a few words (i.e. "Lambert-Beer extinction relationship"), try to add some of the key assumptions (i.e. "single-layer energy budget combining the energy budget of the soil and vegetation" to help other land surface modelers to get a rough idea of the core of CABLE.

P5, Section 3.1. A schematic of POP (along the lines of fig 2 in doi:10.1002/grl.50972) could help the reader to better understanding of what this module does without having to consult the original publication in Biogeosciences.

P7, L 16, e.f. Acronyms for the PFTs are introduced here. These acronyms are only

used a couple of times throughout the text but not enough to accommodate the reader to their meaning. Omit the acronyms and write in full (also in Table 1) for the sake of readability.

P8, L21. Reword. Despite being familiar with modeling land cover changes I don't understand which process is described here.

P9, Section 3.4. The style and information content of this section is very different from the previous paragraphs under section 3. Sections 3.1 to 3.3 are descriptive and do not present any of the equations. Section 3.4 lists the equations with little description. If sections 3.1 to 3.3 are a summary of model developments that have already been published and section 3.4 is a complete new model approach the change in style may be justified. This should be made explicit. I read section 3.4 twice but I could not figure out how this new approach was implemented (in other words, the description would be of little help to write a working code). Another schematic combined with more explanations may help.

P11, L37-38. I read this sentence as if it is impossible for a natural grassland to become a forest. I agree this is probably not the most common land cover change but I was a bit surprised to see this transition being excluded.

P13, Section 5 e.f. Most of the results are descriptive. The authors often claim that the match between simulations and observations is "good" or "acceptable". I still need to meet the first modeler who would claim otherwise. All subjective statements should be removed unless the authors can establish an objective scale of "poor", "acceptable", "good", "very well". It is worth to have a look at the method proposed by Murphy et al 2004 (doi:10.1038/nature02771). Have a look at the performance index outlined in their supplementary material. Murphy et al claim that the method gives the chance that the simulations and the observations come from the same population. Isn't that what we want to know?

P13, L18. Many of the sections start with a single sentence paragraph. This hampers

the readability of the text. This sentence often simply rephrased the caption. It would improve the text flow to use the first paragraph to explain/remind the reader to the significance of the analysis. Why are we, for example, looking at evapotranspiration rather than sensible heat? If the model does a good job in simulating evapotranspiration, which applications could the model be used for?

P13, L28. Write EBL in full. This kind of acronyms hamper readability.

P16, Section 5.3. This section is in the validation section. It is not a validation as the result is not compared to observational products or other simulations. The title is correct in stating it is an illustrative example. Add a single sentence explaining why you show these examples. How do they help to understand the next analysis?

P16, L28. Write PNG in full. I assumed it is Papua New Guinea.

P18, Figure. The coordinates of the sites could go into the text.

P19, L20. It looks like the number preceding 106 km2 is missing. If not, please, write 1.0 x 106 km2 for consistency.

P19, L38. It is stated that the FccxL is large. Is this confirmed by observations? I assume the evidence to look for would be observations showing increasingly faster regrowth of secondary forest.

P23, L18. See comment for P13, L18.

P24, L1. Delete "a" from "a simulates".

P26, L8-9. Please, expand your thoughts and be more specific. Which variable should be benchmarked, which data streams do you intent to use for model-data fusion?

---

## Author Comment (AC2) · 6 Apr 2018

New figures attached (omitted from initial Response)

[Figure]

**Fig. 1.**

Meteorology data

Daily aggregates of GPP, soil temperature, soil moisture

Annually, for woody vegetation tiles: total mortality, sapwood mass & area

**CABLE Biophysics** SUBDIURNAL

**CASA-CNP Biochemistry** DAILY

Updated LAI, $V_{cmax,0}$ and $J_{max,0}$

Gross land-use transitions & wood harvest data

Annually updated C,N,P pools

Annually, for woody vegetation tiles: Stem NPP, max LAI, mean fine-root & leaf carbon pools

Updated tile areas

**POPLUC** Land-use Change ANNUAL

Updated secondary forest age distribution

**POP** Woody Demography ANNUAL

Woody vegetation height

[Figure]

[Figure]

**Fig. 2.**

**CABLE Biophysics**

Daily aggregates of GPP, soil temperature, soil moisture

Updated LAI, $V_{cmax,0}$ and $J_{max,0}$

**CASA-CNP Biochemistry**

**State variables:** C, N, P pools in each of 3 plant compartments (leaves, fine roots, wood); 3 litter compartments (metabolic litter, fine structural litter, coarse woody debris); 3 soil compartments differing by turnover time (fast, slow, passive); soil mineral N and P pools; soil occluded P pool; labile

**Main time step loop (daily)**
- Get leaf phenology phase for deciduous pfts based on remote-sensing climatology or climate history
- Construct root-weighted soil temperature and moisture variables from vertical profiles.
- Evaluate autotrophic growth and maintenance respiration fluxes for leaves, stems (sapwood only) and fine-roots based on tissue nitrogen content. Assumed Lloyd and Taylor (1994) T-dependence. Option for acclimation based on temperature of warmest quarter, similar to acclimation of leaf respiration.
- Compute modifier to leaf base turnover rate based on cold and/or drought stress. For deciduous pfts, reduce or accelerate leaf turnover based on phenological phase.
- Calculate turnover rates of plant pools and fraction of plant turnover entering litter pool. For woody pfts, wood turnover rate is inherited from POP demography module.
- Check if soil nutrient supply can meet the plant uptake demand: otherwise reduce NPP
- Set allocation coefficients to partition NPP between leaves fine roots and wood. For woody pfts, relative leaf and woody allocation coefficients are based on leaf-area to sapwood-area ratio, with sapwood area inherited from POP demography module.
- Compute temperature- and moisture-modifiers to base turnover rates of soil and litter carbon. New options to use Trudinger et al. (2016) moisture response and Lloyd and Taylor (1994) temperature response.
- Calculate turnover rates of plant, soil and litter carbon pools and the transfer coefficients between different pools
- Computing the reduction in litter and SOM decomposition when decomposition rate is N-limiting
- Compute N and P uptake by plants and allocation of each to plant compartments
- Update C, N and P stores according to turnover rates, NPP, allocation coefficients and transfer coefficients computed above.
- Augment annual aggregates of carbon allocated to stems; maximum LAI, mean fine-root and leaf carbon pools for use in POP.
- Compute LAI (from leaf carbon store) and Vcmax,0 from leaf N and P stores. Option to use global synthesis (Walker et al. 2014) to relate $V_{cmax,0}$ to leaf N and P. $J_{max,0}$ set to constant (1.7) times $V_{cmax,0}$.
- Adjust prior $V_{cmax,0}$ and $J_{max,0}$ using **OptJV algorithm** to minimize nitrogen cost of net photosynthesis, based on conditions for the last 5 days.
- Return updated LAI, $V_{cmax,0}$ and $J_{max,0}$ to CABLE biophysics

**Next daily time step**

Annual , for woody vegetation tiles: total mortality, sapwood mass & area

Annual, for woody vegetation tiles: Stem NPP, max LAI, mean fine-root & leaf carbon pools

Annually updated C,N,P pools

**OptJV algorithm for optimizing ratio $V_{cmax,0}$/ $J_{max,0}$**

- Define leaf nitrogen available for re-distribution, based on prior estimates of $V_{cmax,0}$ and $b_{JV} = J_{max,0}/V_{cmax,0}$
- Find the value of $b_{JV}$ that minimizes leaf nitrogen cost per unit net photosynthesis (aggregated over the last 5 days) for each of sunlit and shaded leaves.
- Return to CABLE biophysics the next day's $V_{cmax,0}$ and $J_{max,0}$ for sunlit and shaded leaves, based on updated value of $b_{JV}$.

**POP**

**POPLUC**

**Fig. 3.**

[Figure]

**Fig. 4.**

---

## Author Response (AR1)

**Authors' Response to Referee Comments: gmd-2017-265**

**Reviewer 2**

**Comment 1.1**.

5 Haverd and collaborators present and evaluate the latest developments for the land surface model CABLE. The manuscript contains the information that is expected in such a study and the structure of the manuscript is good. The figures and tables support the text but the text itself is often too concise which hampers the readability of the manuscript. I made several specific comments to underpin my opinion but I would like to encourage the authors to carefully go through the manuscript and check every paragraph, even those not mentioned in my comments.

10 Contrary to the text itself, the title is too wordy. A shorter alternative could be "Incorporating gross land cover change, tree-demography and a novel optimization-based photosynthesis in CABLE land surface model (revision 4546)". The words that will no longer be in the title could be moved to the keywords. Words already in the title should not be repeated as keywords as they will result in new hits from search engines.

**Response 1.1**.

15 We thank Reviewer 1 for the positive general comments and have addressed the conciseness of the text in subsequent comments (1.2, 1.3, 1.5 1.7, 1.10, 1.12, 1.19, 2.24, 2.28, 2.34, 2.35, 2.40).

We have shortened the title to:

"A new version of the CABLE land surface model (Subversion revision r4546), incorporating land use and land cover change, woody vegetation demography and a novel optimisation-based approach to plant coordination of photosynthesis."

**Comment 1.2.**

P4, L16. "is inconsistent with the Co-ordination Hypothesis". Rephrase or better explain. The logic of this sentence appears twisted. As I read this sentence it says that the co-ordination hypothesis differs from the hypothesis that the ratio between Vmax and Jmax is constant, which seems trivial given that the Co-ordination hypothesis was established as an

25 alternative for the fixed-hypothesis. It is more relevant for the reader to be informed whether the co-ordination hypothesis is or isn't at odds with the data.

**Response 1.2**

In the line above, we stated that data confirm the Co-ordination Hypothesis: "This so-called Co-ordination Hypothesis was originally proposed by Chen et al. (1993) and has been verified experimentally by Maire et al. (2012)." We have modified

30 the sentence in question to emphasise poorly appreciated impact of neglecting co-ordination on simulated sensitivity of GPP to CO2 as follows:

"In this work, we will show that the assumption of a temporally invariant ratio of Rubisco and electron-transport capacities (at standard temperature), adopted in Prior CABLE and typically in other LSMs, is not only inconsistent with the Co-ordination Hypothesis, but introduces large uncertainty in simulated sensitivity of GPP to atmospheric $CO_2$

35 concentration."

**Comment 1.3**.

The explanation of the structure of the model would likely benefit from a adding a simplified flowchart-type of figure showing the main dependencies. The actual approaches are often missing and should be added to the text. How is, for

40 example, the radiation transfer through the canopy simulated? Describe the approach in a few words (i.e. "Lambert-Beer extinction relationship"), try to add some of the key assumptions (i.e. "single-layer energy budget combining the energy

budget of the soil and vegetation" to help other land surface modelers to get a rough idea of the core of CABLE.

**Response 1.3**

We appreciate the reviewer's suggestion and further note that comprehensive documentation of the processes encoded in CABLE is restricted to papers that separately describe CABLE biophysics or CABLE biogeochemistry or CABLE
5  population dynamics. To address this we now provide Figure 1 which illustrates how the different model components interact and Figure S1 which gives pseudo code for all the key processes, as a useful reference for what is actually in the CABLE code, as well as highlighting the new developments in this work and how they interact with pre-existing components.

10  **Comment 1.4.**

P5, Section 3.1. A schematic of POP (along the lines of fig 2 in doi:10.1002/grl.50972) could help the reader to better understanding of what this module does without having to consult the original publication in Biogeosciences.

**Response 1.4**

We agree and note that the requested information in now in Figures 1 and S1. (See Response 1.3).

**Comment 1.5**.

P7, L 16, e.f. Acronyms for the PFTs are introduced here. These acronyms are only used a couple of times throughout the text but not enough to accommodate the reader to their meaning. Omit the acronyms and write in full (also in Table 1) for the sake of readability.

20  **Response 1.5:**

We appreciate the helpful comment and have removed the acronyms.

**Comment 1.6**.

P8, L21. Reword. Despite being familiar with modeling land cover changes I don't
25  understand which process is described here.

**Response 1.6**

We have extended the sentence to be very clear that the net biomass loss in secondary forest also has contributions from natural disturbance and expansion:

"Carbon losses by secondary forest harvest and clearing need to be resolved from net biomass loss in secondary forest
30  tiles, which also includes components from natural disturbance and areal expansion."

**Comment 1.7**

P9, Section 3.4. The style and information content of this section is very different from the previous paragraphs under section 3. Sections 3.1 to 3.3 are descriptive and do not present any of the equations. Section 3.4 lists the equations with
35  little description. If sections 3.1 to 3.3 are a summary of model developments that have already been published and section 3.4 is a complete new model approach the change in style may be justified. This should be made explicit. I read section 3.4 twice but I could not figure out how this new approach was implemented (in other words, the description would be of little help to write a working code). Another schematic combined with more explanations may help.

**Response 1.7**

40  Regarding the change in style between the POP and POP-LUC descriptions and the OptJV description, we (i) emphasised that POP equations have already been published and (ii) inserted equations for POPLUC.

In Section 3.1 we now emphasise:

"The summary below is reproduced from these papers, which describe POP in detail and with full equations."

The section of the LUC code that is enhanced by equations is that describing the redistribution of carbon (Section 3.2)
5  This has been modified as follows:

**"Re-distribution of carbon stocks following land-use-change**

[revised manuscript text omitted]

**Comment 1.8**

P11, L37-38. I read this sentence as if it is impossible for a natural grassland to become a forest. I agree this is probably not the most common land cover change but I was a bit surprised to see this transition being excluded.

**Response 1.8**

That is correct. It is an assumption of our parsimonious approach which is now explicit in the text :

"For simplicity, we neglect transitions from natural grass land to forest."

**Comment 1.9**

P13, Section 5 e.f. Most of the results are descriptive. The authors often claim that the match between simulations and observations is "good" or "acceptable". I still need to meet the first modeler who would claim otherwise. All subjective statements should be removed unless the authors can establish an objective scale of "poor", "acceptable", "good", "very well". It is worth to have a look at the method proposed by Murphy et al 2004 (doi:10.1038/nature02771). Have a look at the performance index outlined in their supplementary material. Murphy et al claim that the method gives the chance that the simulations and the observations come from the same population. Isn't that what we want to know?.

**Response 1.9**

We have removed qualitative references in this section and replaced with absolute differences between modeled and obs-based latitudinal profiles:

"CABLE and the LandFlux latitudinal profile of ET differ by a mean absolute error of 0.12 mm d$^{-1}$"

"CABLE and FLUXNET estimates of the latitudinal distribution of GPP differ by mean absolute error of 147 gCm$^{-2}$y$^{-1}$."

"The CABLE and GEOCARBON latitudinal biomass estimates differ by mean absolute error of 0.47 PgCdeg$^{-1}$."

"Latitudinal profiles of soil carbon from CABLE (total soil carbon and litter) differs from the HWSDA product by a mean absolute error of 1.8 PgCdeg$^{-1}$ (Figure 2(xii)), and the CABLE global total of 1426 PgC is 7% higher than the HWSDA estimate of 1329 PgC."

**Comment 1.10**

P13, L18. Many of the sections start with a single sentence paragraph. This hampers the readability of the text. This sentence often simply rephrased the caption. It would improve the text flow to use the first paragraph to explain/remind the reader to the significance of the analysis. Why are we, for example, looking at evapotranspiration rather than sensible heat? If the model does a good job in simulating evapotranspiration, which applications could the model be used for?

Response 1.10

We have extended the opening paragraph of Section 5.1 as follows:

"Model-data comparisons of spatial distributions of key fluxes and stocks are presented in Figure 3. We choose to evaluate the model against GPP, biomass and soil carbon because these are key quantities that are critical constraints on the global terrestrial carbon cycle and for which global distributions are available. We include evapotranspiration (ET) here as it is a key constraint on GPP, because both ET and GPP are regulated by stomatal conductance."

**Comment 1.11**

P13, L28. Write EBL in full. This kind of acronyms hamper readability.

Response 1.11

Done

**Comment 1.12**

P16, Section 5.3. This section is in the validation section. It is not a validation as the result is not compared to observational products or other simulations. The title is correct in stating it is an illustrative example. Add a single sentence explaining why you show these examples. How do they help to understand the next analysis?

Response 1.12

We appreciate the suggestion, and have extended the opening paragraph of Section 5.3 as follows:

"Four examples of contrasting regional land-use histories ($0.5^o$ x $0.5^o$ grid cells) are presented to illustrate carbon pool changes and the rate of land-atmosphere carbon flux from 1860-present (Figure 5). The landscape-scale responses reveal details that are obscured in the subsequent aggregation to regional and global scale (Section 5.4), but are important for demonstrating the functionality of the model at the spatial scale at which it is applied."

P16, L28. Write PNG in full. I assumed it is Papua New Guinea.

**Comment 1.13**

P16, L28. Write PNG in full. I assumed it is Papua New Guinea.

**Response 1.13**

Done

**Comment 1.14**

P18, Figure. The coordinates of the sites could go into the text.

**Response 1.14**

We chose to retain the coordinates in the figure.

**Comment 1.15**

P19, L20. It looks like the number preceding 106 km2 is missing. If not, please, write 1.0 x 106 km2 for consistency.

**Response 1.15**

1.0  has been inserted.

**Comment 1.16**

P19, L38. It is stated that the FccxL is large. Is this confirmed by observations? I assume the evidence to look for would be observations showing increasingly faster regrowth of secondary forest.

**Response 1.16**

We have modified this sentence to emphasise that this term is dominated by the loss of additional sink capacity, which is not observable:

"While the $F_{CC}$ term dominates the sink, no sink or source tem is negligible, and the $F_{CC \times L}$ term (itself dominated by the loss of additional sink capacity) is large, pointing to the need to model the effects of land-use, climate and $CO_2$ on terrestrial carbon stocks explicitly and simultaneously, as we have done here."

**Comment 1.17**

P23, L18. See comment for P13, L18.

**Response 1.17**

We now open Section 5.7 with these sentences to enhance readability:

"Key functions of global terrestrial biosphere models such as CABLE attribution and projection of the global net land carbon sink. Therefore we assess CABLE predictions against observation-based estimates of this important quantity."

**Comment 1.18**

P24, L1. Delete "a" from "a simulates".

**Response 1.18**

Done

**Comment 1.19**

P26, L8-9. Please, expand your thoughts and be more specific. Which variable should be benchmarked, which data streams do you intent to use for model-data fusion?

**Response 1.19**

We have extended this paragraph as follows:

Further work on the model configuration presented here should include formal benchmarking in the International Land Model Benchmarking Project framework (Hoffman et al., 2017) and model-data fusion (Trudinger et al., 2016). The latter would aim to quantify data constraints on the regional and process attribution the global land carbon sink using multiple parameter sets that are consistent with the observations, in the same way that Trudinger et al. (2016) did for the Australian region. Data for this task would comprise observation-based constraints presented in this work, extended for example to include remotely-sensed vegetation cover.

**Reviewer 1**

**Comment 2.1**

Haverd et al present a set of updates to the CABLE model, including the "Populations Order Physiology" (POP) model
representation of land use, an algorithm depicting photosynthetic optimality principles, and several other updates
presented as appendices.

Numerous integrations of the model with different land use and climate drivers are presented, along with a comprehensive
model evaluation exercise. While this is a substantial paper that should almost certainly ultimately be published, and while
it includes many interesting a novel benchmarking approaches that the land surface modeling community would do well to
take notice of any repeat with other models, I find this version of the manuscript in need of considerable work in terms of
the model description presented and in terms of discussion of the uncertainties inherent in both the POP approach and the
other updates.

**Response 2.1**

We thank the reviewer for the positive comments. We trust that our responses to the previous review and the comments
that follow satisfy the request for considerable work.

**Comment 2.2**

Firstly, the authors rather over-zealous 'selling' of the POP concept in the manuscript strikes me as not particularly
objective and thus quite unconvincing. Further, given the lack of critical discussion of the approach, I am left unsure in
which circumstances POP might act as an appropriate simplification, and those in which it would not. For example, there
is no discussion of how PFT competition might be represented in this framework, nor of how it would respond to the
implementation of partial disturbance processes. The somewhat heuristic and undocumented disaggregation of grid cell
fluxes into patches and cohorts (which is the critical central assumption) is also presented without any consideration of
whether it is realistic or appropriate. I realize that it is imperative to illustrate in some way the basic competence of an
LSM, in order to allow the following experiments to be seriously analyzed, but this must be balanced with some humility
about how much can really be read into the conclusions, given the vast difficulties of parameterization and appropriate
validation of these models.

**Response 2.2**

The POP approach has been extensively described and evaluated in three earlier manuscripts, and its limitations noted. We
make it clear that the description here is a summary of what is in those earlier papers. We consider the text in Section 3.1
to be an objective summary of how POP works.

**Comment 2.3.**

Secondly, the model description is inadequate and confusing throughout much of the methods section. I have detailed
specific instances of this below, but in general, the description is vague, not accompanied with technical equations nor any
accompanying documentation, and is not up the standards that are found within a typical GMD article. I suggest that this
section needs completely re-writing with transparency and provenance tracking in mind. In my view it requires a full
separate technical note to allow proper assessment of the methods employed, which again, would be normal practice
within GMD.

**Response 2.3.**

We made substantial changes to the methods to clarify the model description and provenance of model developments in the context of previous work. Please see responses 1.3 and 1.7 above for more details, and the new Figure 1 concisely summarizing model components and developments.

**Comment 2.4.**

Thirdly, the manuscript focuses in great detail on the POP land use and the photosynthetic optimization modifications, then almost ignores the other myriad of modifications that have been made to the model. Why are these two modifications selected for special treatment? Maybe there is a good reason, but it needs to be made clearer.

**Response 2.4**.

There was an intention to split the model development description into two sections to clearly distinguish: (i) implementation of existing parameterisations from the literature (i.e. those described in the Appendix) to those that (ii) required a higher degree of originality. We have made this clearer in the introduction:

"Additional model updates based on existing parameterisations from the literature include: (i) drought and summer-green phenology (Sitch et al., 2003; Sykes et al., 1996); (ii) low-temperature reductions in photosynthetic rates in boreal forests (Bergh et al., 1998); (iii) photo-inhibition of leaf day-respiration (Clark et al., 2011); and (iv) acclimation of autotrophic respiration (Atkin et al., 2016). These are described in Appendix 1."

**Comment 2.5**

Lastly, the paper is essentially presents a new version of CABLE with many updates, but the performance of this new version in contrast to any previous versions is not considered and the impact of the implementation of the different model features is in general ignored, nor is the performance compared to any other LSM. Thus, the skill of this model version is presented in isolation, and is quite difficult to assess other than broadly stating that is performs reasonably well.

**Response 2.5**

We consider that comparison of model simulations with observation-based data carries more weight than with other models or model versions. The reviewer notes (Comment 2.1) that this paper "includes many interesting and novel benchmarking approaches that the land surface modeling community would do well to take notice of any repeat with other models."

We could not meaningfully compare earlier versions of CABLE with global biomass or soil carbon because these stocks are heavily dependent on land-use change which was not represented in earlier versions of CABLE. Also, earlier versions of CABLE could not make use of data pertaining to age-dependent biomass accumulation (Section 5.2) because they lacked tree demography.

Isolating every change and assessing its impact was deemed out of scope for this work, and would not be a productive exercise. For example, it would be of limited use to assess CABLE with LUC switched on and off, since many other studies have already demonstrated that LUC is responsible for huge perturbations to the historic carbon cycle. In the case of the Jmax/Vcmax optimization, we do indeed show results with and without the optimization (Figures 8 and 9).

In the case of the many other changes introduced (Appendix 1), we are relying on established algorithms which have been tested in isolation by their developers (although not of course in CABLE).

It is the combined impact of all the changes that is important for this paper, the purpose of which is to document this new version of the model.

**Comment 2.6**

P1 L5: Critical for what?

**Response 2.6**

**We have deleted "critical".**

**Comment 2.7**

P1 L15: This theory has been proposed previously (Xu et al. 2012), so is not novel

**Response 2.7**

The approach of Xu et al. (2012) is different from our dynamic optimization approach: that of Xu et al. "equalizes" Wc and Wj (although the timescale of this equalization is not obvious), whereas we dynamically minimize the N-cost of photosynthesis, resulting in approximately equal contributions of Wc and Wj to net photosynthesis.

We now reference Ali et al. (2016) and Xu et al. (2012) in the introduction:

"Its advantages as an approach to modelling photosynthetic dynamics using limited data constraints was pointed out by Wang et al. (2017), while Ali et al. (2016) have incorporated it into a global mechanistic model of photosynthetic capacity, based on the optimal nitrogen allocation model of Xu et al. (2012)."

**Comment 2.8**

P1 L21: "state of the art" is jargon and should be replaced by a statement with some clear scientific meaning.

**Response 2.8**

We have removed this phrase. The sentence now reads:

"These new developments enhance CABLE's capability for use within an Earth System Model, and in stand-alone applications to attribute trends and variability in the terrestrial carbon cycle to regions, processes and drivers."

**Comment 2.9**

P1 L25: I wasn't aware that we had any credible estimates of global GPP, let alone centennial trends therein.

**Response 2.9**

Please see our references to Campbell et al. 2017 (Section 5.6) for the COS-estimates of the trend in global GPP.

**Comment 2.10**

P3 L8: These two developments seems quite arbitrary and distinct from one another. Why are they the joint focus of this one paper?

**Response 2.10**

They are both important for global terrestrial carbon balance, which is the focus of the model evaluation. Please also see the sentence in Response 2.8 above.

**Comment 2.11**

P3 L10: Now there is a new list of developments. Why is this list different from the last list?

**Response 2.11**

Please see Response 2.4 clarifying model developments.

**Comment 2.12**.

P3 L33: "second generation" in what sense?

Response 2.12

We have replaced this descriptor with "demography-enabled"

**Comment 2.13**

P3 L36: Many current DGVM models using some sort of similarity clustering to deal with the problem of expanding numbers of disturbances classes to track. (see all implementations of ED...) This might be difficult, but it is nonetheless the 'state of the art', if we are going to use that sort of terminology.

**Response 2.13**

We stated that POP presents a simpler approach to dealing with this problem and removed 'state-of-the-art' phrases in the manuscript.

**Comment 2.14**

P4 L5: POP has some advantages in terms of computational time, but the rules used to disaggregate the big leaf fluxes into size classes of vegetation are necessarily arbitrary. There has not been, as far as I know, any attempt to investigate the uncertainty introduced by not resolving vertical light partitioning in POP. Thus, it is not clear to me that is all that useful of an idea.

**Response 2.14**

The latest version of POP (as described in Haverd et al. 2016) partitions stem biomass increment as already described in Section 3.1, and does now account for the vertical light partitioning:

"In the current implementation of POP, the annual stem biomass increment is partitioned among cohorts and patches in proportion to current net primary production of the given cohort. For this purpose, gross primary production and autotrophic respiration are passed from CABLE to POP, and each is partitioned amongst patches and cohorts. Gross resource uptake is partitioned amongst cohorts and patches in proportion to light interception, evaluated from vertical profiles of gap probabilities, computed using the CABLE maximum leaf area, distributed amongst patches and cohorts in proportion to sapwood area. Leaf, fine-root and sapwood respiration components are also partitioned amongst cohorts and patches, according to the size of each biomass component. Cohort-specific sapwood is prognosed by assuming sapwood conversion to heartwood at a rate 0.05 $y^{-1}$. Cohort-specific leaf and root carbon pools are estimated by partitioning the grid-cell values in proportion to leaf area index (LAI). Net resource uptake for each patch and cohort is evaluated as its gross primary production minus autotrophic respiration."

To be clear that this contrasts with the original algorithm, we modified the introductory paragraph to Section 3.1:

"To enable the extension of CABLE to simulate dynamic land use and implications for forest carbon uptake, we used the most recent version of POP's representation of growth partitioning amongst age/size classes (cohorts) of trees established in the same year that accounts for both cohort-dependent light interception and sapwood respiration. This contrasts with the original growth partitioning which assumed that individuals capture resources in varying proportion to their size."

To assess whether POP is "a useful idea", we have already noted that it has been:

"demonstrated to successfully replicate the effects of rainfall and fire disturbance gradients on vegetation structure along a rainfall gradient in Australian savannah – the Northern Australian Tropical Transect (Haverd et al., 2013c; Haverd et al., 2016b), and leaf-stem allometric relationships derived from global forest data, which may be argued to reflect the simultaneous development of trees in closed forest stands in terms of structural and functional (productivity) attributes (Haverd et al., 2014)."

We further evaluate POP's predictions of age effects on biomass accumulation for boreal, temperate and tropical forests in the current work (Section 5.2).

**Comment 2.15**

P4 L 12: As above, this idea was also proposal by Xu et al. (2012) and it's global implementation presented by Ali et al. (2016)

**Response 2.15**

Please see Response 2.7, additional references and clarification has been added.

**Comment 2.16**

P5 L11: Define what is meant by 'offline' in this context?

**Response 2.16**

We have replaced "offline" with "using prescribed meteorology".

**Comment 2.17**

P5 L27: This sentence is really just hype and doesn't add anything of scientific value to the paper.

**Response 2.17**

We disagree: the sentence "POP is designed to be modular, deterministic, computationally efficient, and based on defensible ecological principles." summarises the design principles of the POP module.

**Comment 2.18**

P5 L30: If the timestep is one year, how is the growth of leaf tissue, and disturbance events from individual fires resolved with appropriate fidelity?

**Response 2.18**

The time-steps for all the processes is much clearer now with our new Figure 1. Growth of leaf tissue is resolved daily. Fire is not implemented in this version of the model.

**Comment 2.19**

P5 L31: Input variables to POP, not CABLE, I assume.

**Response 2.19**

We now clarify "input variables to POP".

**Comment 2.20**

P5 L35: Surely neglecting partial disturbance from fires and mortality will introduce a large bias? How can this decision be justified?

**Response 2.20**

We do not neglect mortatily. Fire is not explicit, but implicit in the catastrophic disturbance that is imposed. We are still working on implementation of fire, as flagged in the final paragraph of the paper.

**Comment 2.21**

P5 L37: Need to define the nature of a 'cohort' here. Are they all of similar height, age, dbh? Similarly, are the 'patches' spatially explicit or implicit? To what does the term 'neighborhoods' refer?

**Response 2.21**

We have removed 'neigbourhoods' (interchangeable with 'patches'). This paragraph has been modified as:
"State variables are the density of tree stems partitioned among cohorts of trees and representative patches of different age-since-last-disturbance across a simulated landscape or grid-cell. Each patch has a number of cohorts. Trees in each cohort are the same age and size because they are established simultaneously and share the same growth rate. Patches are not spatially explicit. Their areal representation in the landscape is given by the patch age distribution."

**Comment 2.22**

P6 L1: Does this just mean that stem biomass is a fixed fraction of NPP?

**Response 2.22**

No. Stem biomass is the outcome of growth and mortality processes, aggregated over cohorts and patches. Mortality is described in the following paragraph.

**Comment 2.23**

P6 L2: GPP and Ra at the grid-scale level?

**Response 2.23**

Yes, amended to "gross primary production and autotrophic respiration for each woody tile".

**Comment 2.24**

P6 L3-6: This is not an adequate description of the disaggregation process, which is the most important assumption in this POP system. How are gap probabilities evaluated? How is light interception of the different cohorts and patches evaluated? With what set of assumptions? I think the authors would do well, if they genuinely wish this approach to become more broadly accepted, to apply some more critical thinking to this particular aspect of the model and to be much more transparent with the limitations and strengths of the assumptions here. Maybe it is defensible to assume that NPP is directly proportional to light interception, or maybe it isn't, but the absence of discussion and questioning of this topic is frustrating. I was a referee on the original POP paper too, and continue to find this to be a limiting aspect of this exercise.

**Response 2.24**

First, we are assuming GPP (not NPP) is proportional to light interception. This is stated clearly in the manuscript.

We agree that the use of gap probabilities and related light interception to partition GPP is too brief and have expanded as follows:

"In the current implementation of POP, the annual stem biomass increment is partitioned among cohorts and patches in proportion to current net primary production of the given cohort (Haverd et al., 2016b). For this purpose, gross primary production and autotrophic respiration for each woody tile are passed from CABLE to POP, and each is partitioned amongst patches and cohorts. Gross resource uptake is partitioned amongst cohorts and patches in proportion to light interception, which is evaluated for each cohort as the difference between downward-looking gap probabilities above and below each cohort. Gap probabilities are calculated using the geometric approach of Haverd et al. (2012). This requires estimates of cohort-specific crown cross-sectional area (related allometrically to DBH) and LAI, computed using the CABLE maximum leaf area, distributed amongst patches and cohorts in proportion to sapwood area. For autotrophic respiration: leaf, fine-root and sapwood respiration components are also partitioned amongst cohorts and patches, according to the size of each biomass component. Cohort-specific sapwood is prognosed by assuming sapwood conversion to heartwood at a rate $0.05 \ y^{-1}$. Cohort-specific leaf and root carbon pools are estimated by partitioning the aggregate values for each woody tile in proportion to leaf area index (LAI). Net resource uptake for each patch and cohort is evaluated as its gross primary production minus autotrophic respiration. "

**Comment 2.25**

P6 L11: I thought there was only catastrophic disturbance?

**Response 2.25**

We have removed "according to disturbance intensity", since partial disturbance is not considered in this work.

**Comment 2.26**

P6 L13: There should, at the very least, be a reference to the place where one can find an actual description of this mortality function. Growth efficiency is often also defined as NPP/LAI (in LPJ, for example). Hence this needs more careful definition. There is no description at all of how the crowding mortality works.

**Response 2.26**

References for growth efficiency and crowding mortality have been inserted.

**Comment 2.27**

P6 L 17: In what sense are the patches 'replicates'?

**Response 2.27**

We agree this is confusing. "replicate" has been removed.

**Comment 2.28**

P6 L24: How are the state variables interpolated? This sentence doesn't make sense to me, nor does the one that follows. Is this a new feature of POP? In which case, it needs much, much clearer documentation.

**Response 2.28**

This interpolation is not a new feature of POP. We have modified the text to be clearer:

"To account for disturbances and the resulting landscape structure, state variables of patches of different ages are linearly interpolated between ages, and weighted by probability intervals from the negative exponential distribution. The resultant weighted average of, for example, total stem biomass or annual stem biomass turnover, is taken to be representative for the grid-cell as a whole."

**Comment 2.29**

P6 L30: "The resulting tree biomass turnover" : resulting from what? The combination of the mortality rates discussed above?

**Response 2.29**

This sentence has been clarified as:

"The POP biomass lost by mortality is applied as an annual decrease in the CASA-CNP tree biomass pool, and replaces the default fixed biomass turnover rate."

**Comment 2.30**

P6 L32: Thus far the distinction of how CASA-CNP and CABLE interact has not been made clear.

**Response 2.30**

This is now clear in Figures 1 and A1.

**Comment 2.31**

P6 L35: Is this a feature of POP, or of CASA-CNP?

**Response 2.31**

Clarified as:

"Sapwood replaces stem-wood biomass in the CASA-CNP calculation of stem respiration."

**Comment 2.32**

P6 L36: I thought NPP, GPP and Ra were all calculated at the grid scale level, so how can NPP thus be depdendant on stand age?

**Response 2.32**

This line is only true if the woody tile has a uniform age distribution. We have modified the text to read:

"These feedbacks of POP structural variables on leaf area and autotrophic respiration result in net primary production that reflect the area-average sapwood area and mass of each woody tile."

**Comment 2.33**

P7 L 12: What is a biome in this context? Is each grid cell really only populated by one or two PFTs?

**Response 2.33**

10    We have clarified the definition:

"Biomes (combinations of dominant plant types (Prentice et al., 1992)) are mapped…"

Yes, as stated, the biomes (one per grid-cell) are each mapped to one or two CABLE pfts.

**Comment 2.34**

15    P8 L5: How many age classes are there? Is this dynamic or fixed?

**Response 2.34**

We have inserted the following text: "POPLUC represents integral secondary forest ages classes from 0 to 1000 y old inclusive. This is fixed, although many ages may have a weight of zero. The frequency distribution is fully dynamic. In contrast POP represents 60 patches in each woody tile, spanning a distribution of ages from 0 to 1000. "

**Comment 2.35**

P8 L10: I don't think the description in section 3.1 was sufficient to let the reader understand how this interacts with the age structure tracking in secondary forests.

**Response 2.35**

25    Agreed. This is confusing. As now detailed in Figure 1, the POPLUC module supplies POP with the secondary forest patch age distribution. We have amended the text as follows:

"The POPLUC code provides the secondary forest patch age distribution to POP. POP tracks biomass in each of a set of patches with different ages,, based on patch-dependent growth and turnover. It then interpolates biomass in the simulated patches to give biomass in each integral age class represented by the secondary forest tile patch age distribution."

**Comment 2.36**

P8 L14: typo in 'pools'

**Response 2.36**

Fixed.

**Comment 2.37**

P9 L5: Again, this seems like a huge shift in focus from land use and demography to fast timescale photosynthesis. Further, a very similar method was suggested by the studies of Xu, and Ali.

**Response 2.37**

Please see Response 2.7.

**Comment 2.38**

P11 L23: So, shouldn't the BGC model be CASA-CN, not CASA-CNP?

**Response 2.38**

Although the P-cycle was disabled, CASA-CNP is still the name of the BGC model.

**Comment 2.39**

P11 L31: Which version of CRU-NCEP are you using here?

**Response 2.39**

V7: now noted.

**Comment 2.40**

P12 L9: None of these scenarios explores the impact of the photosynthetic optimization approach that you just documented in considerable detail? This happens later, but that is mixing of methods and results and is confusing.

**Response 2.40**

We now pre-empt this:
"In addition to the above scenarios, we also explored the impact on global GPP of dynamically optimizing $b_{JV}=J_{max,0}/V_{cmax,0}$. Simulations were performed under assumptions of dynamically optimized and fixed $b_{JV}$ (values of 1.6, 1.7, 1.8). For these simulations, static 1860 land-cover was assumed and for computational efficiency, simulations were based on a sample of 1000 randomly distributed grid-cells across the global ice-free land-surface."

**Comment 2.41**

P15 L4: From where are these successional data taken? Surely there is massive geographical/climatic variance in these rates? Is the model sampled to make sure that it has the same climatic regimes as the dataset?

**Response 2.41**

As stated in the text, the sampling is only approximate:

"CABLE regrowth rates of secondary forests in the Tropical Rainforest, Tropical Seasonal Forest and Tropical Dry Forest/Savanna biomes (Figure 1) in South America…. observation-based estimates by Poorter et al. (2016) from 1500 forest plots at 45 sites spanning the major environmental gradients across the Neotropics (Figure 4)."

We plot the observed and modeled 20-y biomass accumulation against mean annual precip, because Poorter et al. established this as the strongest environmental predictor. We add this clarification:

"… Neotropics, where mean annual rainfall is the strongest environmental predictor of biomass accumulation after 20 y (Poorter et al., 2016)."

**Comment 2.42**

P16 L27 : Papua New Guinea??

**Response 2.42**

Fixed.

**Comment 2.43**

P26 L10: Arguably, if one is going to say 'state of the art', this model should already include fire effects on vegetation, croplands and dynamic biogeography and PFT interactions already, since those are things that are included in many other models. Saying a model is the 'state of the art' is a bold statement, given both the complexity and the wide range of approaches within this field. Further, it is not really necessary for the purpose of model documentation. All LSMs have strengths and weaknesses in different areas. Progress can only be made be careful and objective analysis of the uncertainties inherent in different types of structural assumption, parameters and boundary conditions. I found this paper somewhat lacking in any thoughtful discussion of these things, whether the model is 'state of the art' or not.

**Response 2.43**

Both uses of 'state of the art' (in the Abstract and Conclusion) have been removed.

[revised manuscript text omitted]

* * *
Margin comments:

Microsoft Office User 29/3/18 1:44 PM

Microsoft Office User 29/3/18 1:47 PM

Microsoft Office User 29/3/18 1:47 PM

Microsoft Office User 29/3/18 1:48 PM

Microsoft Office User 29/3/18 1:50 PM
Comment [1]: You could add all the years it has contributed eg, Le Quere et al 2016, 2017, 2018.

Matthias Cuntz 29/3/18 5:44 PM

Matthias Cuntz 29/3/18 9:39 PM

Microsoft Office User 29/3/18 1:55 PM

Vanessa Haverd 28/3/18 11:32 PM

al., 2013b). The key simplification in the POP approach, compared with other demography-enabled DVMs, is to compute physiological processes such as photosynthesis at the scale of a land-cover tile ("grid-scale"), but to partition the grid-scale biomass increment amongst sub grid-scale patches, each subject to its own dynamics, and distinguished by time since last disturbance. This makes tracking biomass in a large number of patch ages (as arise through both natural disturbance and human land-cover change) easy, and circumvents the computational difficulties of tracking land-cover classes in DVMs.

**Coordination of Photosynthesis**

Almost all global LSMs use the photosynthesis model of Farquhar et al. (1980), or a related scheme derived from this model. Different implementations result in divergent estimates of the response of photosynthesis to environmental drivers in large scale models (e.g. Friend et al., 2014). One reason for this may be that global LSMs have mostly neglected the constraint imposed by the evolutionary-ecological assumption that plants optimise productivity in their environment through relative investment in electron transport and Rubisco-limited steps in the photosynthesis chain, that adjust seasonally and across biomes to be co-limiting. This so-called Co-ordination Hypothesis was originally proposed by Chen et al. (1993) and has been verified experimentally by Maire et al. (2012). Its advantages as an approach to modelling photosynthetic dynamics using limited data constraints was pointed out by Wang et al. (2017), while Ali et al. (2016) have incorporated it into a global mechanistic model of photosynthetic capacity, based on the optimal nitrogen allocation model of Xu et al. (2012). In this work, we will show that the assumption of a temporally invariant ratio of Rubisco and electron-transport capacities (at standard temperature), adopted in Prior CABLE and typically in other LSMs, is not only inconsistent with the Co-ordination Hypothesis, but introduces large uncertainty in simulated sensitivity of GPP to atmospheric $CO_2$ concentration. We solve this problem by developing an algorithm for dynamic optimisation of this ratio, such that co-ordination is achieved as an outcome of fitness maximisation.

**Paper Structure**

The paper is structured as follows. In Section 2 we review the basic structure of CABLE. In Section 3 we describe the model developments that are the focus of this work: firstly, updates to the POP module for woody demography and disturbance; secondly, the new land-use and land-cover change module; thirdly, the dynamic optimisation of plant photosynthesis. In Section 4, we describe the modelling protocol that is used to deliver simulations for evaluating the new model version, and assessing terrestrial carbon-cycle implications of changing climate, $CO_2$, land-use and land-cover over the historical period (1860-2016). In Section 5, we present results of these simulations. Section 5.1 evaluates predictions of present-day spatial distributions of evapotranspiration, gross primary production, biomass and soil carbon. Section 5.2 evaluates predictions of biomass accumulation rates in re-growing forests. Section 5.3 illustrates the capability and behaviour of the land use implementation, showing examples of land-atmosphere carbon exchange at four locations with contrasting LUC histories. Section 5.4 shows the implications of $CO_2$, climate and LUC on historical global and regional land-atmosphere exchange. Sections 5.5 and 5.6 address the implications of simulated photosynthesis co-ordination for the sensitivity of photosynthesis to $CO_2$ and for the $CO_2$ fertilisation of global photosynthesis. Section 5.7 evaluates the new model's prediction of the annual time series of the net land carbon sink by comparison with the equivalent quantity derived from atmospheric mass balance (atmospheric growth rate + ocean sink – fossil fuel emissions). Priorities for future development are summarised in Section 6.
* * *
**Comments (margin):**

Vanessa Haverd 28/3/18 11:33 PM

Microsoft Office User 29/3/18 5:32 PM

Vanessa Haverd 28/3/18 10:33 PM

Vanessa Haverd 28/3/18 10:34 PM

[Figure]

Figure 1: Sub-models of CABLE and their interactions.

[revised manuscript text omitted]

Vanessa Haverd 28/3/18 11:27 AM

Vanessa Haverd 28/3/18 11:27 AM

Vanessa Haverd 28/3/18 11:27 AM

Vanessa Haverd 28/3/18 11:27 AM

Vanessa Haverd 28/3/18 11:31 AM

Vanessa Haverd 28/3/18 11:31 AM

Vanessa Haverd 28/3/18 11:31 AM

Vanessa Haverd 28/3/18 11:31 AM

Vanessa Haverd 28/3/18 11:31 AM

Vanessa Haverd 28/3/18 11:31 AM

Vanessa Haverd 29/3/18 2:59 AM

Vanessa Haverd 29/3/18 2:59 AM

Vanessa Haverd 29/3/18 3:00 AM

Vanessa Haverd 29/3/18 2:52 AM

Vanessa Haverd 28/3/18 2:46 PM

Vanessa Haverd 28/3/18 2:46 PM

Vanessa Haverd 28/3/18 2:46 PM

Vanessa Haverd 28/3/18 2:46 PM

Vanessa Haverd 28/3/18 2:46 PM

Vanessa Haverd 28/3/18 2:46 PM

Vanessa Haverd 28/3/18 2:46 PM

Vanessa Haverd 28/3/18 3:21 PM

Vanessa Haverd 28/3/18 3:46 PM

Vanessa Haverd 28/3/18 3:47 PM

Vanessa Haverd 28/3/18 3:47 PM

[revised manuscript text omitted]

**CABLE Biophysics**

**State Variables:** Soil moisture and temperature in 6 vertical layers; snow water equivalent (up to 3 layers); canopy interception store.

Initialise parameter and state variables

**Main time step loop (sub-diurnal)**
- Read subdiurnal meteorology ← Meteorology data
- Compute surface roughness characteristics
- Compute albedo of canopy and back-ground
- Canopy radiation transfer: Compute canopy extinction coefficients for beam and diffuse radiation; canopy reflectances; fractions beam and diffuse incoming radiation; short-wave radiation absorption by shaded and sunlit leaves and background (soil or snow); iso-thermal longwave radiation absorption by background and vegetation
- Update canopy water storage and fraction wet canopy and compute throughfall

**Loop over Monin-Obukov atmospheric stability parameter**
- Compute aerodynamic properties: friction velocity and turbulent resistances required to compute the dispersion matrix (Localised Near Field Theory)
- Compute forced convection boundary layer conductance at leaf surface

**Loop over (dry) leaf temperature** (solves coupled leaf energy balance, stomatal conductance, net photosynthesis)
- Compute free convection boundary layer conductance at leaf surface
- Compute T-dependent $V_{cmax}$ and $J_{max}$ for shaded and sunlit leaves, accounting for extinction through canopy (leaf-to-canopy scaling).
- Compute T-dependent Michaelis Menten constants for Rubisco
- Compute leaf respiration
  - Fixed fraction of $V_{cmax}$ (default)
  - Alternative: temperature acclimation function multiplied by instantaneous T-response
  - Option: modify for photo-inhibition
- Solve coupled equations for net photosynthesis and stomatal conductance
- Compute root-water extraction and update soil-moisture modifier to stomatal conductance

**Check for convergence**

- Update dry leaf surface energy balance
- Compute leaf wet leaf energy balance, including wet leaf temperature
- Update canopy energy balance
- Compute soil surface energy balance (long-wave component depends on canopy energy balance above)
- Compute dispersion matrix, and update in-canopy temperature and humidity
- Recompute Monin-Obukhov stability parameter

**Next stability iteration**

- Soil physics: update vertical distribution of heat and water content in soil and snow and compute surface runoff and deep soil drainage
- Update climate history variables as required for phenology, acclimation of respiration, optimization of $J_{max}/V_{cmax}$
- Update daily aggregates of GPP, soil temperature and moisture for use in biogeochemistry
- If end of day: Call driver for CASA-CNP Biochemistry
- If end of year: Call drivers for POPLUC (land-use change) and POP (woody demography)

**Next sub-diurnal time step**

[Figure]

| Daily aggregates of GPP, soil temperature, soil moisture | Updated LAI, Vcmax,0 and Jmax,0 | Woody vegetation height | Updated tile areas |
|---|---|---|---|

**CASA-CNP Biogeochemistry** | **POP** | **POPLUC**

[Figure]

**CASA-CNP Biochemistry**

**State variables:** C, N, P pools in each of 3 plant compartments (leaves, fine roots, wood); 3 litter compartments (metabolic litter, fine structural litter, coarse woody debris); 3 soil compartments differing by turnover time (fast, slow, passive); soil mineral N and P pools; soil occluded P pool; labile C pool.

**CABLE Biophysics**

Daily aggregates of GPP, soil temperature, soil moisture

Updated LAI, $V_{cmax,0}$ and $J_{max,0}$

**Main time step loop (daily)**
- Get leaf phenology phase for deciduous pfts based on remote-sensing climatology or climate history
- Construct root-weighted soil temperature and moisture variables from vertical profiles.
- Evaluate autotrophic growth and maintenance respiration fluxes for leaves, stems (sapwood only) and fine-roots based on tissue nitrogen content. Assumed Lloyd and Taylor (1994) T-dependence. Option for acclimation based on temperature of warmest quarter, similar to acclimation of leaf respiration.
- Compute modifier to leaf base turnover rate based on cold and/or drought stress. For deciduous pfts, reduce or accelerate leaf turnover based on phenological phase.
- Calculate turnover rates of plant pools and fraction of plant turnover entering litter pool. For woody pfts, wood turnover rate is inherited from POP demography module.
- Check if soil nutrient supply can meet the plant uptake demand: otherwise reduce NPP
- Set allocation coefficients to partition NPP between leaves fine roots and wood. For woody pfts, relative leaf and woody allocation coefficients are based on leaf-area to sapwood-area ratio, with sapwood area inherited from POP demography module.
- Compute temperature- and moisture-modifiers to base turnover rates of soil and litter carbon. New options to use Trudinger et al. (2016) moisture response and Lloyd and Taylor (1994) temperature response.
- Calculate turnover rates of plant, soil and litter carbon pools and the transfer coefficients between different pools
- Computing the reduction in litter and SOM decomposition when decomposition rate is N-limiting
- Compute N and P uptake by plants and allocation of each to plant compartments
- Update C, N and P stores according to turnover rates, NPP, allocation coefficients and transfer coefficients computed above.
- Augment annual aggregates of carbon allocated to stems; maximum LAI, mean fine-root and leaf carbon pools for use in POP.
- Compute LAI (from leaf carbon store) and Vcmax,0 from leaf N and P stores. Option to use global synthesis (Walker et al. 2014) to relate $V_{cmax,0}$ to leaf N and P. $J_{max,0}$ set to constant (1.7) times $V_{cmax,0}$.
- Adjust prior $V_{cmax,0}$ and $J_{max,0}$ using **OptJV algorithm** to minimize nitrogen cost of net photosynthesis, based on conditions for the last 5 days.
- Return updated LAI, $V_{cmax,0}$ and $J_{max,0}$ to CABLE biophysics

**Next daily time step**

Annual, for woody vegetation tiles: total mortality, sapwood mass & area

Annual, for woody vegetation tiles: Stem NPP, max LAI, mean fine-root & leaf carbon pools

Annually updated C,N,P pools

**OptJV algorithm for optimizing ratio $V_{cmax,0}$/$J_{max,0}$**

- Define leaf nitrogen available for re-distribution, based on prior estimates of $V_{cmax,0}$ and $b_{JV}=J_{max,0}/V_{cmax,0}$.
- Find the value of $b_{JV}$ that minimizes leaf nitrogen cost per unit net photosynthesis (aggregated over the last 5 days) for each of sunlit and shaded leaves.
- Return to CABLE biophysics the next day's $V_{cmax,0}$ and $J_{max,0}$ for sunlit and shaded leaves, based on updated value of $b_{jv}$.

**POP**

**POPLUC**

Figure A1b: CASA-CNP Bogeochemistry

[Figure]

CASA-CNP Biogeochemistry

CABLE Biophysics

Annual, for woody vegetation tiles: Stem NPP, max LAI, mean fine-root & leaf carbon pools

Updated C,N,P pools

Gross land-use transitions & wood harvest data

Updated tile areas

**POPLUC**
**Land-use Change & Land Management**

**State variables:** State variables: tile area fractions of primary vegetation, secondary woody vegetation, open land; Crop- and pasture-fractions of open land; age distribution of secondary woody vegetation; wood harvest and clearance pools, each with 3 turnover times (1 y, 10 y , 100y); combined harvest and grazing product pool (turnover time 1y).

**Main time step loop (yearly)**
- Update land-use area fractions, subject to land availability.
- In secondary forest tiles, update the areal fraction of each integral age class (0-400 y), as influenced by secondary forest expansion, harvest, clearing and natural disturbance.
- Redistribute C, N, P associated with land-use transitions and wood harvest.
- Updated tile areas are returned to CABLE biophysics. Updated C,N,P pools returned to CASA-CNP. Updated secondary forest age distribution returned to POP.
- Direct C emissions from decay of wood harvest and clearance pools and crop-grazing pool are deducted from grid-cell Net Biospheric Production.

**Next yearly time step**

Updated secondary forest age distribution

Total grid-cell mortality, sapwood mass/area

Woody veg height

**POP**
**Woody Demography & Landscape Heterogeneity**

**State variables:** density of tree stems partitioned among cohorts of trees and representative neighbourhoods (patches) of different age-since-last-disturbance in each woody vegetation tile.

**Main time step loop (yearly)**
- Partition stem growth amongst patches (distinguished by time since last disturbance) within the landscape and cohorts within each patch.
- Augment biomass, sapwood and heartwood in patches and cohorts by stem growth, accounting for sapwood-heartwood conversion.
- Compute resource-limitation and crowding mortalities and reduce cohort stem densities accordingly. Remove cohorts in which stem densities are reduced to near-zero.
- Recruit new cohorts
- Calculate annually-resolved patch age frequency distribution (exponential distribution for unmanaged forests), or inherit distribution from LUC code (secondary forests)
- Interpolate key patch variables (biomass; growth; sapwood area and volume; crowding and resource-limitation mortality) to annually-resolved patch age.
  - Integrate these variables, weighted by patch frequency, to obtain grid-cell-average variables.
  - Construct grid-cell disturbance mortality as the residual: growth minus crowding mortality minus resource-limitation mortality minus Δbiomass.
- Total grid-cell mortality, sapwood mass and sapwood area are returned to CASA-CNP
- Woody vegetation height returned to CABLE biophysics.

**Next yearly time step**

[revised manuscript text omitted]

---

## Author Response (AR2)

Response to Review by editor (22 May 2018)

Dear Editor,

Thank-you for the constructive feed-back. We have addressed your minor points as follows.

**Comment 1.1** Overall the label and reference to tables and equations are incorrects (not properly indexed)

**Response 1.1**
We have checked these, and corrected, particularly Tables4-5.

**Comment 1.2** The added figure 1 is valuable but the caption could be slightly detailed;

**Response 1.2** The figure caption has been expanded as follows:

"Figure 1: Major sub-models of CABLE (revision 4546), showing forcing data, characteristic time steps, and information flows between modules, which include fluxes, store updates, and changes to vegetation characteristics and their spatial extent (tile areas) within grid cells. Data from faster modules are aggregated before passing to slower modules. Faster modules are updated with data from slower modules at the rate of the slower time step. "

**Comment 1.3** In the model description, section "Re-distribution of carbon stocks following land-use-change": Equation 2 is not easy to follow with for example one index in some symbols ("0") that is not fully explicited. You could help the reader with few additional clarifications;

**Response 1.3**

"We have extended the explanation of Eqs 1-2 as follows:
Changes in pool sizes of biomass, soil and litter carbon in the biogeochemical module are updated to reflect the areal changes from gross land-use transitions. Analogous updates occur for nitrogen pools. The mass balance equation for the carbon density $c_j$ [g m$^{-2}$] in each land-use tile $L$, with area $A_L$ [m$^2$] that accounts for the possibility of more than one gross receiver ($r$) or donor ($d$) transition to or from the tile, is:

$$c_{j,L,0}A_{L,0} - c_{j,L,0}\Delta A_{L,d} + F_{j,L,r}^{transfer} = c_{j,L}\left(A_{L,0} + \Delta A_L\right) \qquad (1)$$

Here $j$=1-9 (referring to carbon in leaf, wood, fine roots, 3 litter pools and 3 soil pools) and $L$ = 1-3 (referring to primary woody, secondary woody, open land-use tiles); subscript $0$ refers to the value of the tile area or carbon density prior to the transitions; $\Delta A_L$ refers to the total (net) change in land-area of the $L_{th}$ tile; $\Delta A_{L,d}$ refers to the absolute change in land area due to donor transitions. In Eq **Error! Reference source not found.**, the first term on the LHS is the carbon stock prior to land-use perturbations; the second term is the carbon lost from the tile due to donor transitions (transitions from the $Lth$ tile) and the third

term is the carbon gained by receiver transitions (transitions to the *Lth* tile).
The term on the RHS is the carbon stock following the perturbations (i.e. the
product of the new carbon density and the new tile area).
The carbon gained by receiver transitions is generally:

$$F_{j,L,r}^{transfer} = \sum_{k=1}^{n_{trans}} \Delta A_k c_{j,k}$$

where the total transfer of carbon is summed over all possible gross transitions
($n_{trans}$ = 4), and each transition contributes carbon to the receiver pool that is
equal to the product of the transition area $\Delta A_k$ multiplied by the carbon density
of the donor pool $c_{j,k}$."

**Comment 1.4** In the model description, section "Dynamic optimization of Bjv":
please also check the coherence of all symbols.

**Response 1.4**

We have checked for symbol consistency, and clarified the distinction between
super-script and sub-script zeros:
"where superscript *0* denotes prior estimate; subscript *0* denotes standard
temperature"

**Comment 1.5** first sentence of section 5.7 seems incomplete.

**Response 1.5**

We have modified as followed:
"Key functions of global terrestrial biosphere models such as CABLE are the
attribution and projection of the global net land carbon sink."

**Comment 1.6** Finally, as reviewer 1 was asking, a little "critical appraisal" of the
simplifications/choices/hypothesis made in the POP model would be beneficial
in the conclusion section: your view with respect to other approaches.

**Response 1.6** We have inserted the following text in Section 3.1:

[revised manuscript text omitted]

Unknown
Field Code Changed

Vanessa Haverd 1/6/18 5:59 PM

Vanessa Haverd 1/6/18 5:59 PM

Vanessa Haverd 1/6/18 5:59 PM

Vanessa Haverd 1/6/18 5:59 PM

Vanessa Haverd 1/6/18 5:59 PM
Vanessa Haverd 1/6/18 4:27 PM
Vanessa Haverd 1/6/18 4:28 PM
Vanessa Haverd 1/6/18 4:27 PM
Vanessa Haverd 1/6/18 4:28 PM
Vanessa Haverd 1/6/18 4:28 PM
Vanessa Haverd 1/6/18 4:27 PM
Vanessa Haverd 1/6/18 5:59 PM
Vanessa Haverd 25/5/18 2:49 PM
Vanessa Haverd 1/6/18 5:59 PM
Vanessa Haverd 1/6/18 5:59 PM
Vanessa Haverd 1/6/18 5:59 PM
Vanessa Haverd 1/6/18 5:59 PM
Vanessa Haverd 1/6/18 5:59 PM
Vanessa Haverd 1/6/18 5:59 PM

[revised manuscript text omitted]

**CABLE Biophysics**

**State Variables:** Soil moisture and temperature in 6 vertical layers; snow water equivalent (up to 3 layers); canopy interception store.

Initialise parameter and state variables

**Main time step loop (sub-diurnal)**
- Read subdiurnal meteorology ← Meteorology data
- Compute surface roughness characteristics
- Compute albedo of canopy and back-ground
- Canopy radiation transfer: Compute canopy extinction coefficients for beam and diffuse radiation; canopy reflectances; fractions beam and diffuse incoming radiation; short-wave radiation absorption by shaded and sunlit leaves and background (soil or snow); iso-thermal longwave radiation absorption by background and vegetation
- Update canopy water storage and fraction wet canopy and compute throughfall

**Loop over Monin-Obukov atmospheric stability parameter**
- Compute aerodynamic properties: friction velocity and turbulent resistances required to compute the dispersion matrix (Localised Near Field Theory)
- Compute forced convection boundary layer conductance at leaf surface

**Loop over (dry) leaf temperature** (solves coupled leaf energy balance, stomatal conductance, net photosynthesis)
- Compute free convection boundary layer conductance at leaf surface
- Compute T-dependent $V_{cmax}$ and $J_{max}$ for shaded and sunlit leaves, accounting for extinction through canopy (leaf-to-canopy scaling).
- Compute T-dependent Michaelis Menten constants for Rubisco
- Compute leaf respiration
    - Fixed fraction of $V_{cmax}$ (default)
    - Alternative: temperature acclimation function multiplied by instantaneous T-response
    - Option: modify for photo-inhibition
- Solve coupled equations for net photosynthesis and stomatal conductance
- Compute root-water extraction and update soil-moisture modifier to stomatal conductance

**Check for convergence**

- Update dry leaf surface energy balance
- Compute leaf wet leaf energy balance, including wet leaf temperature
- Update canopy energy balance
- Compute soil surface energy balance (long-wave component depends on canopy energy balance above)
- Compute dispersion matrix, and update in-canopy temperature and humidity
- Recompute Monin-Obukhov stability parameter

**Next stability iteration**

- Soil physics: update vertical distribution of heat and water content in soil and snow and compute surface runoff and deep soil drainage
- Update climate history variables as required for phenology, acclimation of respiration, optimization of $J_{max}$/$V_{cmax}$
- Update daily aggregates of GPP, soil temperature and moisture for use in biogeochemistry
- If end of day: Call driver for CASA-CNP Biochemistry
- If end of year: Call drivers for POPLUC (land-use change) and POP (woody demography)

**Next sub-diurnal time step**

[Figure]

| Daily aggregates of GPP, soil temperature, soil moisture | Updated LAI, Vcmax,0 and Jmax,0 | Woody vegetation height | Updated tile areas |
|---|---|---|---|

| CASA-CNP Biogeochemistry | POP | POPLUC |
|---|---|---|

Figure A1a: CABLE Biophysics

[Figure]

**CABLE Biophysics**

**Daily aggregates of GPP, soil temperature, soil moisture**

**Updated LAI, $V_{cmax,0}$ and $J_{max,0}$**

**CASA-CNP Biochemistry**

**State variables:** C, N, P pools in each of 3 plant compartments (leaves, fine roots, wood); 3 litter compartments (metabolic litter, fine structural litter, coarse woody debris); 3 soil compartments differing by turnover time (fast, slow, passive); soil mineral N and P pools; soil occluded P pool; labile C pool.

**Main time step loop (daily)**
- Get leaf phenology phase for deciduous pfts based on remote-sensing climatology or climate history
- Construct root-weighted soil temperature and moisture variables from vertical profiles.
- Evaluate autotrophic growth and maintenance respiration fluxes for leaves, stems (sapwood only) and fine-roots based on tissue nitrogen content. Assumed Lloyd and Taylor (1994) T-dependence. Option for acclimation based on temperature of warmest quarter, similar to acclimation of leaf respiration.
- Compute modifier to leaf base turnover rate based on cold and/or drought stress. For deciduous pfts, reduce or accelerate leaf turnover based on phenological phase.
- Calculate turnover rates of plant pools and fraction of plant turnover entering litter pool. For woody pfts, wood turnover rate is inherited from POP demography module.
- Check if soil nutrient supply can meet the plant uptake demand: otherwise reduce NPP
- Set allocation coefficients to partition NPP between leaves fine roots and wood. For woody pfts, relative leaf and woody allocation coefficients are based on leaf-area to sapwood-area ratio, with sapwood area inherited from POP demography module.
- Compute temperature- and moisture-modifiers to base turnover rates of soil and litter carbon. New options to use Trudinger et al. (2016) moisture response and Lloyd and Taylor (1994) temperature response.
- Calculate turnover rates of plant, soil and litter carbon pools and the transfer coefficients between different pools
- Computing the reduction in litter and SOM decomposition when decomposition rate is N-limiting
- Compute N and P uptake by plants and allocation of each to plant compartments
- Update C, N and P stores according to turnover rates, NPP, allocation coefficients and transfer coefficients computed above.
- Augment annual aggregates of carbon allocated to stems; maximum LAI, mean fine-root and leaf carbon pools for use in POP.
- Compute LAI (from leaf carbon store) and Vcmax,0 from leaf N and P stores. Option to use global synthesis (Walker et al. 2014) to relate $V_{cmax,0}$ to leaf N and P. $J_{max,0}$ set to constant (1.7) times $V_{cmax,0}$.
- Adjust prior $V_{cmax,0}$ and $J_{max,0}$ using **OptJV algorithm** to minimize nitrogen cost of net photosynthesis, based on conditions for the last 5 days.
- Return updated LAI, $V_{cmax,0}$ and $J_{max,0}$ to CABLE biophysics

**Next daily time step**

[Figure]

**Annual, for woody vegetation tiles: total mortality, sapwood mass & area**

**Annual, for woody vegetation tiles: Stem NPP, max LAI, mean fine-root & leaf carbon pools**

**Annually updated C,N,P pools**

**OptJV algorithm for optimizing ratio $V_{cmax,0}/J_{max,0}$**

- Define leaf nitrogen available for re-distribution, based on prior estimates of $V_{cmax,0}$ and $b_{JV} = J_{max,0}/V_{cmax,0}$.
- Find the value of $b_{JV}$ that minimizes leaf nitrogen cost per unit net photosynthesis (aggregated over the last 5 days) for each of sunlit and shaded leaves.
- Return to CABLE biophysics the next day's $V_{cmax,0}$ and $J_{max,0}$ for sunlit and shaded leaves, based on updated value of $b_{jv}$.

**POP**

**POPLUC**

Figure A1b: CASA-CNP Bogeochemistry

[Figure]

**CASA-CNP Biogeochemistry**

**CABLE Biophysics**

Annual, for woody vegetation tiles: Stem NPP, max LAI, mean fine-root & leaf carbon pools

Updated C,N,P pools

Gross land-use transitions & wood harvest data

Updated tile areas

**POPLUC**
**Land-use Change & Land Management**

**State variables:** State variables: tile area fractions of primary vegetation, secondary woody vegetation, open land; Crop- and pasture-fractions of open land; age distribution of secondary woody vegetation; wood harvest and clearance pools, each with 3 turnover times (1 y, 10 y , 100y); combined harvest and grazing product pool (turnover time 1y).

**Main time step loop (yearly)**
- Update land-use area fractions, subject to land availability.
- In secondary forest tiles, update the areal fraction of each integral age class (0-400 y), as influenced by secondary forest expansion, harvest, clearing and natural disturbance.
- Redistribute C, N, P associated with land-use transitions and wood harvest.
- Updated tile areas are returned to CABLE biophysics. Updated C,N,P pools returned to CASA-CNP. Updated secondary forest age distribution returned to POP.
- Direct C emissions from decay of wood harvest and clearance pools and crop-grazing pool are deducted from grid-cell Net Biospheric Production.

**Next yearly time step**

Updated secondary forest age distribution

Total grid-cell mortality, sapwood mass/area

Woody veg height

**POP**
**Woody Demography & Landscape Heterogeneity**

**State variables:** density of tree stems partitioned among cohorts of trees and representative neighbourhoods (patches) of different age-since-last-disturbance in each woody vegetation tile.

**Main time step loop (yearly)**
- Partition stem growth amongst patches (distinguished by time since last disturbance) within the landscape and cohorts within each patch.
- Augment biomass, sapwood and heartwood in patches and cohorts by stem growth, accounting for sapwood-heartwood conversion.
- Compute resource-limitation and crowding mortalities and reduce cohort stem densities accordingly. Remove cohorts in which stem densities are reduced to near-zero.
- Recruit new cohorts
- Calculate annually-resolved patch age frequency distribution (exponential distribution for unmanaged forests), or inherit distribution from LUC code (secondary forests)
- Interpolate key patch variables (biomass; growth; sapwood area and volume; crowding and resource-limitation mortality) to annually-resolved patch age.
  - Integrate these variables, weighted by patch frequency, to obtain grid-cell-average variables.
  - Construct grid-cell disturbance mortality as the residual: growth minus crowding mortality minus resource-limitation mortality minus Δbiomass.
- Total grid-cell mortality, sapwood mass and sapwood area are returned to CASA-CNP
- Woody vegetation height returned to CABLE biophysics.

**Next yearly time step**

[revised manuscript text omitted]